# Estimation of annual runoff by exploiting long-term spatial patterns and short records within a geostatistical framework

Thea Roksvåg[1], Ingelin Steinsland[1], and Kolbjørn Engeland[2]

[1]Norwegian University of Science and Technology, NTNU, Department of Mathematical Sciences.
[2]The Norwegian Water Resources and Energy Directorate, NVE

**Correspondence:** Thea Roksvåg (thea.roksvag@ntnu.no)

**Abstract.** In this article, we present a Bayesian geostatistical framework that is particularly suitable for interpolation of hydrological data when the available dataset is sparse and includes both long and short records of runoff. A key feature of the proposed framework is that several years of runoff are modeled simultaneously with two spatial fields: One that is common for all years under study that represents the runoff generation due to long-term (climatic) conditions, and one that is year specific. The climatic spatial field captures how short records of runoff from partially gauged catchments vary relative to longer time series from other catchments, and transfers this information across years. To make the Bayesian model computationally feasible and fast, we use integrated nested Laplace approximations (INLA) and the stochastic partial differential equation (SPDE) approach to spatial modeling.

The geostatistical framework is demonstrated by filling in missing values of annual runoff and by predicting mean annual runoff for around 200 catchments in Norway. The predictive performance is compared to Top-Kriging (interpolation method) and simple linear regression (record augmentation method). The results show that if the runoff is driven by processes that are repeated over time (e.g. orographic precipitation patterns), the value of including short records in the suggested model is large. For partially gauged catchments the suggested framework perform better than comparable methods, and one annual observation from the target catchment can lead to a 50 % reduction in RMSE compared to when no observations are available from the target catchment. We also find that short records safely can be included in the framework regardless of the spatial characteristics of the underlying climate, and down to record lengths of one year.

## 1  Introduction

Characteristic values for streamflow are used for various purposes in water resources management. High flow indices or design flood estimates are needed for flood risk assessments and design of infrastructure and dams, low flow indices are needed for assessment of environmental flow and reliability assessment of water supply, while mean annual flow is an important basis for water resources management and a key for design of water supply systems and allocation of water resources between stakeholders. Mean annual flow can also be used as a predictor for low flow and high flow indices (Sælthun et al., 1997; Engeland and Hisdal, 2009).

At locations with measurements, the streamflow indices can be estimated based on observations. However, streamflow is only measured at a limited number of locations, and in many applications we need to predict the streamflow indices at ungauged locations. This is a central problem in hydrology and known as the Prediction in Ungauged Basins problem (Blöschl et al., 2013). Often it is of interest to estimate flow indices that represent the long-term average behavior in a catchment. If this is the case, using only a few years of data from the target catchment might lead to biased estimates. The reason is climate variability over short time scales combined with sample uncertainty. Often a minimum record length is recommended for estimation of long-term indices, but a substantial part of the available streamflow gauges in the world have too short records to provide reliable estimates. These short data series can, however, provide useful information if they are used together with longer time series from other catchments (Laaha and Blöschl, 2005). Motivated by this, we propose a framework for runoff interpolation particularly suitable for datasets including data series of this type, more specifically runoff datasets including a mix of fully gauged catchments (with data available from the whole study period) and partially gauged catchments (with data available from a subset of the study period). We suggest a framework for runoff interpolation that unifies two commonly used statistical approaches for runoff estimation: Geostatistical approaches and approaches for exploiting short records of data.

Within the geostatistical framework, Gaussian random fields (GRFs) are often used to model hydrological phenomena that are continuous in space and/or time. The hydrological variable of interest is a GRF if a vector containing a random sample of length $n$ from the process follows a Gaussian distribution with mean vector $\boldsymbol{\mu}$ and covariance matrix $\boldsymbol{\Sigma}$ (Cressie, 1993). The elements in the covariance matrix are typically determined by a covariance function that is specified based on the pairwise distances between the $n$ target locations. For most environmental variables it is straight forward to compute these distances. However, for runoff related variables the measure of distance is ambiguous because the observations are related to catchment areas, some of them nested, and not to point locations in space. Traditionally, this challenge has been solved by simply interpreting runoff as a point referenced process linked to the catchment centroids or stream outlets (see e.g. Merz and Blöschl (2005); Skøien et al. (2003); Adamowski and Bocci (2001)). The problem with these methods is that they can lead to a violation of basic conservation laws, and several alternatives approaches are suggested for making an interpolation scheme that takes the nested structure of catchments into account (Sauquet et al., 2000; Gottschalk, 1993; Skøien et al., 2006). In particular, the Top-Kriging approach suggested by Skøien et al. (2006) has shown promising results for interpolation of hydrological variables (Viglione et al., 2013). In the Top-Kriging approach, information from a subcatchment is weighted more than information from a nearby non-overlapping catchment when performing runoff predictions for an ungauged catchment.

In the literature, there exist several techniques to exploit short records of runoff, and these are known as record augmentation techniques. The first step in a record augmentation procedure is often to find one or several donor catchments with longer time series of runoff. The donor catchments are typically selected based on runoff correlation, catchment similarity, or proximity in space. By applying e.g. linear regression approaches and/or computing the correlation between time series, a relationship between the target catchment and the donor catchments is developed. Next, the longer time series from the donor catchment(s) are used to perform predictions for the target catchments for years/months/days without measurements (see e.g. Fiering (1963), Hirsch (1982), Matalas and Jacobs (1964), Vogel and Stedinger (1985) or Laaha and Blöschl (2005)). The regression and/or

correlation analysis is performed based on runoff observations that is of the same type as the target flow index, i.e. for annual runoff, short records of annual runoff are used (Blöschl et al., 2013).

In this paper, we suggest a geostatistical Bayesian framework that represents a new way of exploiting short records of data. The framework is constructed to exploit long-term spatial patterns stored in sparse datasets, i.e. hydrological datasets with several missing values. A key feature of the suggested framework is that it simultaneously models several years of runoff. This is done by using two statistical spatial components or GRFs in the hydrological model: The first GRF is common for all years under study and models the long-term spatial variability of runoff. We denote this the climatic GRF as it represents the spatial variability over time, or what we refer to as the climate in the study area. In this context the term *climate* also includes the runoff generation due to catchment characteristics that are static, like elevation and slope. The other GRF is year-specific and models the annual discrepancy from the climate, and we denote this the annual or year-specific GRF. If we have a study area for which the spatial variability of runoff is stable over time, the climatic GRF will capture this tendency. Hence, it will also capture how short records of runoff vary relative to longer data series from other catchments. On the other hand, if there are no strong long-term trends present in the data, the year-specific GRF will dominate over the climatic GRF. For this scenario, short records from the target catchment(s) will have less impact on the final results. By adjusting the two spatial fields relative to each other, our method represents a way for detecting long-term trends and uses this to exploit short records in the runoff interpolation.

The framework we suggest is flexible and can be used for any hydrological variable. However, its benefits are linked to exploiting long-term spatial trends in the data, and in order to work better than other interpolation methods, the hydrological variable of interest should be driven by processes that are repeated over time. For this reason, we develop our methodology for *annual* runoff. This is a flow index that often has a prominent spatial pattern over years, for example due to orographic precipitation and topography that creates weather divides. To describe study areas and/or variables like this, we hereby introduce the terms *hydrologically spatially stable* and *hydrological spatial stability*. For hydrologically spatially stable areas, the difference in runoff between two locations for a given year is close to the difference in runoff between these two locations any other year. Be aware that a hydrologically spatially stable area can both have large differences in annual runoff between two close locations, and have large variability in annual runoff over years for a given location. The key property is that the underlying spatial pattern is preserved over time.

While annual runoff represents a hydrologically spatially stable variable for many countries, the spatial pattern for monthly runoff is typically less stable. This is due to local weather patterns and the variability in the seasonality of snow accumulation and snow melt. To demonstrate our methodology for a variable with less hydrological spatial stability, we therefore fit the framework to annual time series of monthly runoff. These predictions allow us to discuss how the approach might work in different regions.

In the following presentation, we introduce two versions of our framework, i.e. two geostatistical models. The first model we propose is denoted the areal model and is particularly suitable for mass-conserved hydrological variables. It ensures that the water balance is preserved for the predicted runoff for any point in the landscape, and defines the average runoff in a catchment as the average point runoff integrated over (nested) catchment areas. This way, the nested structure of catchments is taken into

account, and the interpretation of covariance between two catchments is similar to the one of Top-Kriging. The areal model for annual runoff is already presented in Roksvåg et al. (2020) where its mass-conserving properties were demonstrated through an example from Voss in western Norway. The model's ability to exploit short data records was also indicated in Roksvåg et al. (2020), but the property was not tested for a larger dataset or compared to any existing methods. This is a key contribution of this article.

As an alternative to the areal model, we also propose a model that defines runoff as a point referenced process for which distances are measured between the catchment centroids. This model does not consider preservation of water balance, but on the other hand it can be used for any point referenced environmental variable, and it is computationally faster than the areal model. This model is more similar to models that have been used traditionally in hydrology, and we denote this the centroid model. Both the areal model and the centroid model have the ability to exploit hydrological spatial stability, but have different benefits, drawbacks and hence also area of use. These are discussed and highlighted throughout the article.

The main objective of this work is to present and evaluate the new geostatistical framework for exploiting short records and to compare its performance to Top-Kriging (interpolation method) and simple linear regression (record augmentation technique). In particular our goals are to:

1) Assess the two spatial models' ability to fill in missing annual observations of runoff for ungauged and partially gauged catchments.

2) Assess the two spatial models' ability to predict *mean* annual runoff for a longer time period for catchments with varying record lengths.

Through 1) and 2) we also aim to:

3) Demonstrate the potential added value of including short records in the modeling, compared to not using them or compared to using traditional methods.

The framework is evaluated by using annual and monthly runoff data from catchments in Norway. This dataset is presented in the section that follows (Section 2). Next, in Section 3, we briefly introduce relevant statistical background theory and notation. In Section 4 the suggested model for annual runoff is presented, before evaluation scores and experimental set-up are presented in Section 5. Here, we have one experimental set-up for annual predictions (Section 5.1) and one set-up for mean annual predictions (Section 5.2). In Section 6, the results are presented before they are discussed in Section 7. Finally, we conclude in Section 8.

## 2   Study area

The study is carried out by using a dataset from Norway provided by the Norwegian Water Resources and Energy Directorate (NVE). It originally consisted of *daily* runoff data from 1981-2010. To make the data suitable for an analysis, a data preparation procedure was performed to construct datasets for two purposes: For assessing the framework's ability to fill in missing annual data and for assessing the framework's ability to predict mean annual runoff.

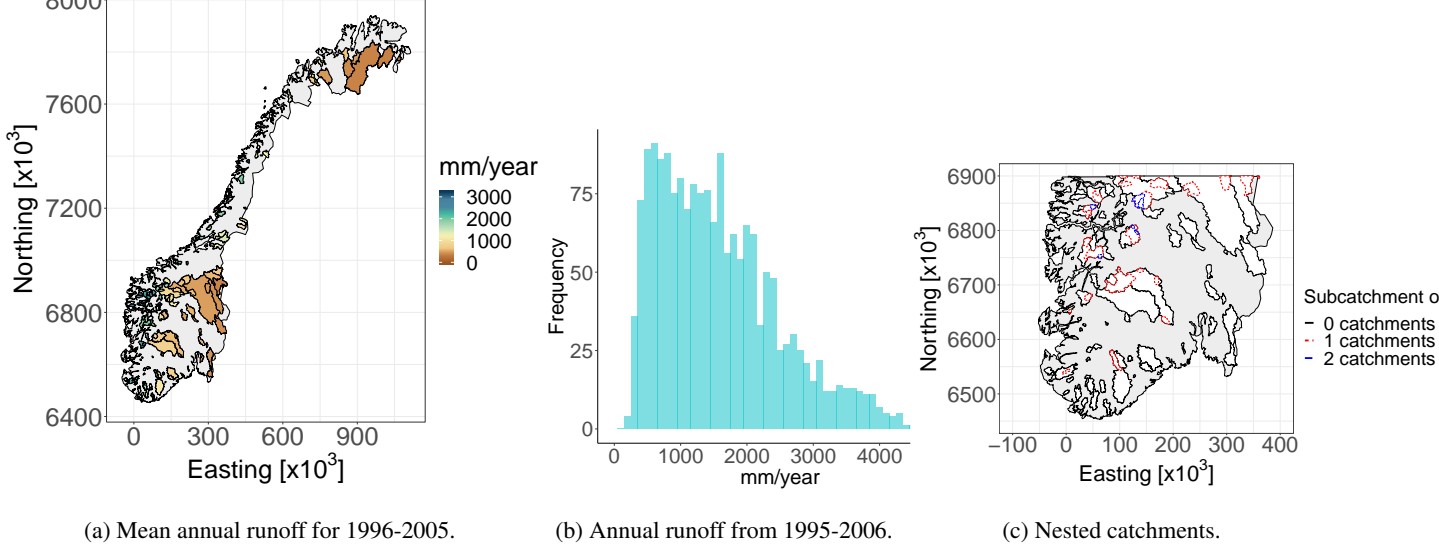

(a) Mean annual runoff for 1996-2005.     (b) Annual runoff from 1995-2006.     (c) Nested catchments.

**Figure 1.** Mean annual runoff (1996-2005) from 180 fully gauged catchments in Norway (1a) and annual runoff observations from all 180 catchments and years (1b). These data are used to evaluate the framework's ability to fill in missing values for individual years. 30 % of the involved catchments are nested and most of these are located in southern Norway as visualized in Figure 1c. In this figure, colored catchments are subcatchments of at least one larger catchment, while the black catchments are not subcatchments of any larger catchment (but might contain 1 or 2 smaller catchments). In the visualization in Figure 1a, subcatchments are plotted on top of larger catchments, and this is done throughout the article. The coordinate system used is EUREF89 - UTM33N (EPSG 25833). See Figure 7 for a closer image of the observed mean annual runoff in southern Norway (1996-2005).

To make a cross-validation dataset for the experiments related to infill of missing annual data, the daily runoff data were aggregated to annual runoff data for hydrological years that start September 1th and end August 31st. We chose to consider a study period from 1996-2005: For this period we had the maximum number of fully gauged catchments, i.e. 180 catchments. These 180 fully gauged catchments have areas ranging from 13 $km^2$ to 15500 $km^2$ and median elevations from 85 to 1562 m

5   a.s.l. Among these, none were significantly influenced by human activities in the time period of interest. Regulated catchments were removed from the original dataset.

Figure 1a and Figure 1b show two visualizations of the annual data from the 180 Norwegian target catchments. We see a large spatial variability of runoff. The annual runoff (for individual years) ranges from 170 mm/year to 5050 mm/year, whereas the mean annual runoff ranges from 350 mm/year to 4230 mm/year, with the highest values of runoff in western Norway and

10   more moderate values in east and north. In total 53 of the 180 catchments were nested with at least one other catchment, i.e. the degree of nestedness is 30 %. Most of these are located in southern Norway, and the nested structure here is shown in Figure 1c. The remaining 127 catchments did not overlap with any other catchment.

In the Norwegian annual data in Figure 1a we see an east-west pattern of runoff. This is mainly caused by orographic enhancement of frontal precipitation formed around extratropical cyclones. The orographic enhancement is driven by the steep

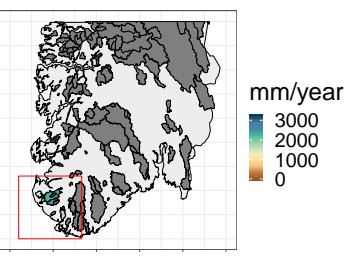 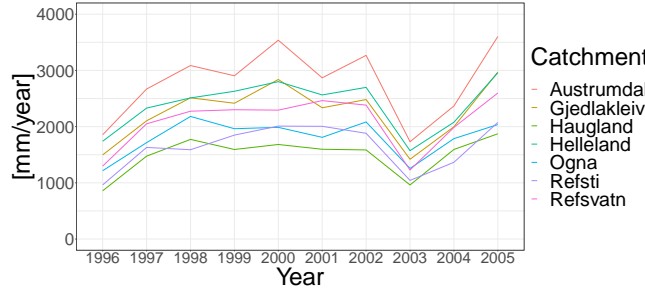

(a) 7 catchments.                          (b) Annual time series.

**Figure 2.** Time series of annual runoff from 7 selected catchments in western Norway. The 7 lines are almost parallel (and barely cross) indicating that most of the spatial variability can be explained by long-term spatial patterns. This represents a good example of what we mean by hydrological spatial stability.

mountains in western Norway that create a topographic barrier for the western wind belt, which transports moist air across the North Atlantic (Stohl et al., 2008). Due to the orographic enhancement, the maximum precipitation is observed at distances 30-70 km from the coast (Førland, 1979) and not necessarily at the highest elevations since the air dries out due to precipitation. The topography results in a spatial pattern of runoff that is stable over years, which means that there exist long-term spatial
patterns in the data that can be exploited.

Figure 2 shows time series of annual runoff from seven catchments in the south-western part of the country. We see a year to year variation for all catchments that is quite large. However, the seven time series are almost parallel (and almost never cross), indicating that the difference in annual runoff between stations is approximately constant over time. Hence, this is a good example of what we mean by hydrological spatial stability. The tendency we see in Figure 2 is typical for the annual
runoff in many of the areas in Norway.

To illustrate the framework's properties for study areas and/or variables that are driven by more unstable weather patterns or hydrological processes, we also aggregated the daily runoff data to monthly runoff for the 180 catchments in Figure 1a. From this we made annual time series of monthly runoff for 1996-2005 for three months: A winter month dominated by snow accumulation (January), a spring month with snow melting (April) and a summer month dominated by rain (June). The annual
observations of monthly runoff for the selected months are presented in Figure 3, and we see that January has the lowest average runoff whereas June has the highest. The variation in average monthly runoff describes a runoff regime, and in Norway the combination of snow accumulation, snow melt, and evapotranspiration processes control this regime (Gottschalk et al., 1979). Along the west coast, the winter weather is typically rainy with temperatures above the freezing point. In these regions the highest monthly runoff is observed in October - December. The colder areas are found in the interior of the country with
winters dominated by snow accumulation. In these regions the highest monthly runoff is observed for the snow melt season (May – June).

Annual time series of monthly runoff from the 7 selected catchments from Figure 2a are shown in Figure 4. We see that the spatial pattern is less stable on a monthly scale compared to the annual scale, particularly for January: The difference in monthly runoff between stations over time is not approximately constant for January, and the runoff in January hence represents a more hydrologically spatially unstable variable in Norway. For June however, the hydrological spatial stability is higher.

The cross-validation datasets described so far are used to assess the framework's ability to fill in missing annual observations for a 10 year period and to illustrate how the models behave for different hydrological settings. In addition, we also evaluate the framework's ability to predict *mean* annual runoff, which is a key hydrological signature. This is done for a 30 year period, from 1981 to 2010. As we consider a longer time period for this assessment, a different subset of the original dataset was used: More specifically annual data from 260 catchments located in southern Norway. These are shown in Figure 5. Each of the 260

catchments in Figure 5 have at least one observation of annual runoff between 1981-2010, but only 83 of them are fully gauged in the time period of interest (i.e. have annual observations for all 30 years). Among the partially gauged catchments, the mean record length is 15, while the median record length is 13. Furthermore, 20 of the involved catchments only have 1, 2 or 3 annual observations. As for the previously described datasets, we removed regulated catchments that were significantly influenced by human activity. Also note that we in this experiment only consider catchments from southern Norway. This is done to reduce

the computational complexity of fitting 30 years of runoff simultaneously in a cross-validation setting.

When using the data in Figure 5 to predict mean annual runoff, we do predictions by cross-validation for the 83 fully gauged catchments. However, data from both partially gauged and fully gauged catchments are included in the observation sample (see Section 5.2). For the 83 fully gauged catchments in Figure 5, 53 % of the catchments were nested with a fully gauged or a partially gauged catchment.

**3    Statistical methodology**

In Section 4 we present two Bayesian geostatistical models for runoff interpolation particularly suitable for sparse datasets containing several missing values. First, some statistical background is necessary.

**3.1    Bayesian statistics and hierarchal modeling**

The goal in hydrology is to learn about processes related to hydrological variables like daily rainfall, annual runoff or the 5th

percentile flow. To gain knowledge about the different hydrological processes, relevant data are collected. There are always uncertainties related to the data that must be accounted for in an analysis, and which make a statistical analysis appropriate.

Assuming $\boldsymbol{x} = (x_1, ... x_n)$ is a vector consisting of hydrological variables of interest, e.g. the annual runoff at several locations for a specific year, the observation likelihood $\pi(\boldsymbol{y}|\boldsymbol{x})$ expresses how the data $\boldsymbol{y} = (y_1, ..., y_m)$ are connected to the truth $\boldsymbol{x}$. In the classical frequentist stastistical approach, the variables in $\boldsymbol{x}$ are considered as unknown, but fixed. In the Bayesian

approach however, $\boldsymbol{x}$ is considered to be a quantity whose variation can be described by a probability distribution (see e.g. Casella and Berger (1990)). Prior to the analysis, this probability distribution is expressed through what is called a prior distribution $\pi(\boldsymbol{x})$. This is constructed based on expert knowledge about the variable(s) of interest. The goal of the Bayesian analysis

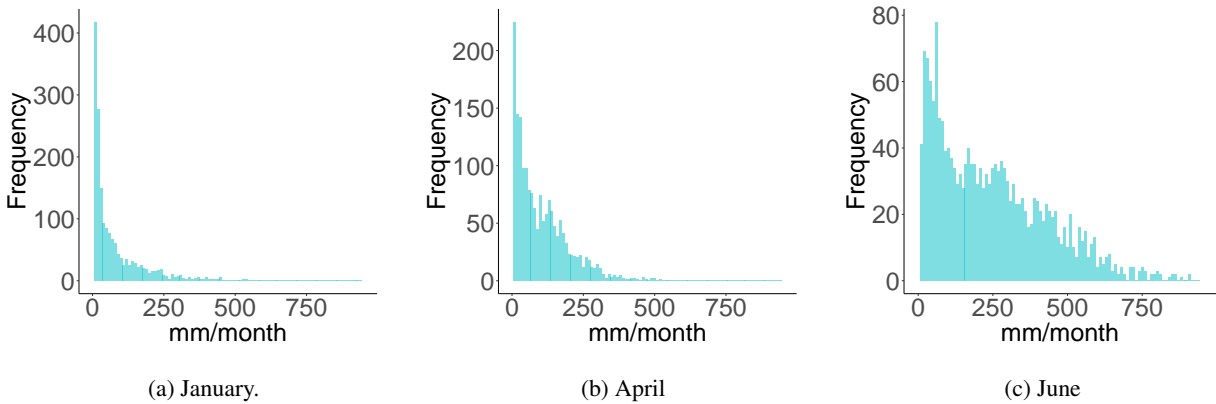

(a) January.          (b) April          (c) June

**Figure 3.** Monthly runoff data (1996-2005) from 180 catchments in Norway for January, April and June. These are used to evaluate the framework's ability to fill in missing values for hydrological variables and/or study areas that are driven by more unstable weather patterns.

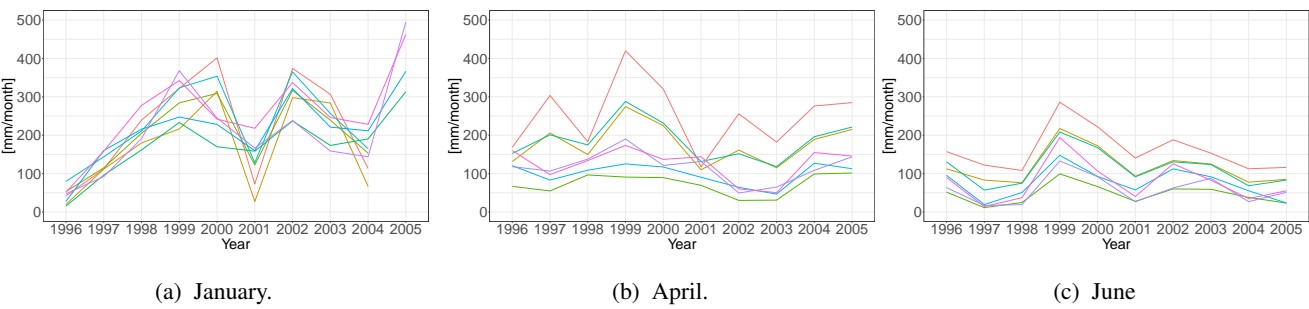

(a) January.          (b) April.          (c) June

**Figure 4.** Annual series of monthly runoff for January, April and June for the 7 catchments in Figure 2a. The time series for January and April are less parallel compared to the time series for June and for the annual runoff (Figure 2b). This suggests that the datasets from January and April represent a more hydrologically spatially unstable setting.

is to update the prior distribution by using data. Through Bayes' formula, the so-called posterior distribution of $x$ is obtained:

$$\pi(\boldsymbol{x}|\boldsymbol{y}) = \frac{\pi(\boldsymbol{x})\pi(\boldsymbol{y}|\boldsymbol{x})}{\pi(\boldsymbol{y})} \propto \pi(\boldsymbol{x})\pi(\boldsymbol{y}|\boldsymbol{x}). \tag{1}$$

Next, the marginal distribution $\pi(x_i|\boldsymbol{y})$ for $x_i \in \boldsymbol{x}$ can be integrated out, and a prediction of $x_i$ can be summarized through e.g. the mean, median or the mode of the posterior distribution $\pi(x_i|\boldsymbol{y})$.

5      If a complex process is under study, it is sometimes easier to model it by thinking of its mechanisms in a hierarchy of underlying processes or distributions (Banerjee et al., 2014). The annual runoff $x$ can e.g. be thought of as a process that depends on some parameters $\boldsymbol{\theta}$ that express the spatial correlation between locations. Here, both $x$ and $\boldsymbol{\theta}$ are stochastic variables with prior (and posterior) distributions. A Bayesian model of this type is typically expressed as a three-staged hierarchical model where the first stage consists of the observation likelihood $\pi(\boldsymbol{y}|\boldsymbol{x}, \boldsymbol{\theta})$, the second stage is the prior distribution $\pi(\boldsymbol{x}|\boldsymbol{\theta})$,

10     often referred to as the latent model or process model, while the third stage is the prior distribution of the model parameters

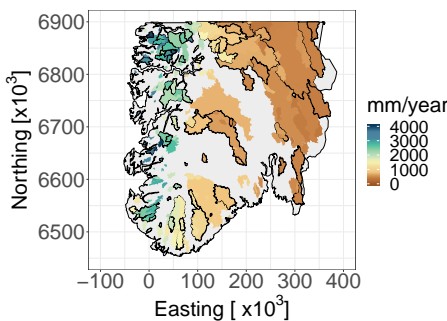

**Figure 5.** Mean annual runoff for 1981-2010 for 260 catchments in southern Norway where only 83 of them are fully gauged (i.e. have annual data for each year in the study period). The 83 fully gauged catchments have black borders in the above plot. In addition, there are data available from 177 so-called partially gauged catchments. These have at least one annual observation between 1981-2010 and are visible as catchments without borders in the above figure. Among the 83 fully gauged catchments, 44 catchments are nested (53 %) while 39 catchments don't overlap with any other catchment in the dataset. Data from the catchments in this figure are used to evaluate the framework's ability to estimate *mean annual runoff*.

$\pi(\boldsymbol{\theta})$. As before, Bayes' formula can be used to make inference about the variables of interest $\boldsymbol{x}$, but also about the model parameters $\boldsymbol{\theta}$ given the set of observations $\boldsymbol{y}$. In this study we use a three-staged hierarchical Bayesian model to model annual runoff.

### 3.2 Gaussian random fields

Gaussian random fields (GRFs) are commonly used to model environmental variables like precipitation, runoff and temperature or other phenomena that are continuous in space and/or time. In this analysis, the second stage of the Bayesian hierarchical model consists of GRFs that model the spatial dependency of runoff between catchments. A continuous field $\{x(\boldsymbol{u}); \boldsymbol{u} \in \mathcal{D}\}$ defined on a spatial domain $\mathcal{D} \in \mathcal{R}^2$ is a GRF if for any collection of locations $\boldsymbol{u}_1, ..., \boldsymbol{u}_n \in \mathcal{D}$ the vector $(x(\boldsymbol{u}_1), ..., x(\boldsymbol{u}_n))^{\mathrm{T}}$ follows a multivariate normal distribution (Cressie, 1993), i.e. $(x(\boldsymbol{u}_1), ..., x(\boldsymbol{u}_n))^{\mathrm{T}} \sim \mathcal{N}(\boldsymbol{\mu}, \boldsymbol{\Sigma})$ where $\boldsymbol{\mu}$ is a vector of expected

values and $\boldsymbol{\Sigma}$ is the covariance matrix. The covariance matrix $\boldsymbol{\Sigma}$ defines the dependency structure in the spatial domain, and element $(i, j)$ is typically constructed from a covariance function $C(\boldsymbol{u}_i, \boldsymbol{u}_j)$. The dependency structure for a spatial process is often characterized by two parameters: The marginal variance $\sigma^2$ and the range $\rho$. The marginal variance provides information about the spatial variability of the process of interest, while the range gives information about how the covariance between the process at two locations decays with distance. The range is defined as the distance at which the correlation between two

locations in space has dropped to almost 0. If the range and the marginal variance are constant over the spatial domain, we have a stationary GRF.

In this study, the involved GRFs have their dependency structure specified by a stationary Matérn covariance function that is given by

$$C(\boldsymbol{u}_i, \boldsymbol{u}_j) = \frac{\sigma^2}{2^{\nu-1}\Gamma(\nu)}(\kappa||\boldsymbol{u}_j - \boldsymbol{u}_i||)^\nu K_\nu(\kappa||\boldsymbol{u}_j - \boldsymbol{u}_i||). \tag{2}$$

Here, $||\boldsymbol{u}_j - \boldsymbol{u}_i||$ is the Euclidean distance between two locations $\boldsymbol{u}_i, \boldsymbol{u}_j \in \mathcal{R}^d$, $K_\nu$ is the modified Bessel function of the second kind and order $\nu > 0$, $\Gamma(\cdot)$ is the gamma function and $\sigma^2$ is the marginal variance that controls the spatial variability (Guttorp and Gneiting, 2006). The parameter $\kappa$ is the scale parameter, and it can be shown empirically that the spatial range can be expressed as $\rho = \sqrt{8\nu}/\kappa$, where $\rho$ is defined as the distance at which the correlation between two locations has dropped to 0.1. Using a Matérn GRF is convenient for computational reasons because it makes it possible to use the SPDE approach to spatial modeling from Lindgren et al. (2011) which is briefly described in Section 4.3.

## 3.3 Kriging and Top-Kriging

Within the geostatistical framework, Kriging approaches have shown promising results for interpolation of hydrological variables (see e.g. Gottschalk (1993), Sauquet et al. (2000) or Merz and Blöschl (2005)). In Kriging methods, the target variable is represented as a random field, typically a Gaussian random field $x(\boldsymbol{u})$ defined through a covariance structure and some unknown parameters. The process of interest is observed at $n$ locations $\boldsymbol{u}_1, ..., \boldsymbol{u}_n$, and any unknown parameters can be estimated based on e.g. maximum likelihood procedures. Furthermore, to estimate the value of the variable $\hat{x}(\boldsymbol{u}_0)$ at an unobserved location $\boldsymbol{u}_0$ a weighted average of the observations is used, i.e.

$$\hat{x}(\boldsymbol{u_0}) = \sum_{i=1}^{n} \lambda_i x(\boldsymbol{u}_i), \tag{3}$$

where $\lambda_i$ are interpolation weights and $x(\boldsymbol{u}_i)$ for $i = 1, ..n$ are observed values. The interpolation weights are computed by assuming that $\hat{x}(\boldsymbol{u}_0)$ is the Best Linear Unbiased Estimator (BLUE) of $x(\boldsymbol{u}_0)$. That is, we determine $\hat{x}(\boldsymbol{u}_0)$ by finding the weights that both minimize the mean squared error and that give zero mean expected error (Cressie, 1993). Mark that the consequence of the latter, is that the Kriging weights are restricted to sum to 1, i.e. $\sum_{i=1}^{n} \lambda_i = 1$ if we assume that the process is homogeneous in space.

Further, to minimize the mean squared error of the Kriging-predictor in Equation (3), the covariance function (or variogram) must be estimated and evaluated. The covariance function typically depends on the distance between the observations and the target locations, such that observations measured close to the target location $\boldsymbol{u}_0$ are weighted more than observations further away. In many hydrological applications, the centroids of the catchments are used to compute the catchment distances (Merz and Blöschl, 2005; Skøien et al., 2003), but as mentioned in the introduction this can lead to a violation of basic mass conservation laws. The reason is that streamflow variables are connected to (catchment) areas, not single point locations. Catchments are also organized into subcatchments, and this should be considered when computing the Kriging weights.

The Top-Kriging approach suggested by Skøien et al. (2006) is an example of a method that takes the nested structure of catchments into account. In this method, the streamflow observations are interpreted as areal referenced, and the covariance

is computed based on the pairwise distances between all grid nodes in a discretization of the involved catchments. This way, observations from a subcatchment can be weighted more than observations from nearby, non-overlapping catchments. Top-Kriging is currently one of the leading methods for interpolation of hydrological variables (Viglione et al., 2013) and is therefore chosen as a benchmark when we evaluate our new interpolation approach.

## 3.4 Methods for exploiting short records (record augmentation techniques)

The framework we suggest is both a framework for spatial interpolation and a framework for record augmentation. There exist several approaches for record augmentation for which many of them are based on developing a linear relationship between the target catchment and one or several catchments with longer time series of runoff (Fiering, 1963; Laaha and Blöschl, 2005; Matalas and Jacobs, 1964). One class of approaches is the maintenance of variance extension (MOVE) methods. MOVE methods are based on developing a linear relationship between the target catchment and the donor(s) catchment(s) by assuming that the sample mean and sample variance of runoff are maintained over time for the target catchment (Hirsch, 1982). There are different ways the sample mean and sample variance can be estimated, giving different estimators for the predicted runoff. Another way to develop a linear relationship between a donor and a target catchment, is to use simple linear regression (Hirsch, 1982). In this article, we use simple linear regression as a benchmark method, in addition to Top-Kriging.

Assume annual runoff is observed for year $1, ..., n$ in the target catchment and that there exist annual runoff data from some other catchments for year $1, ..., n + m$. Simple linear regression is performed by first finding a so-called donor catchment for the catchment of interest. This can be e.g. the closest catchment in space or a catchment with similar catchment characteristics (elevation, annual precipitation, vegetation). Next, it is assumed that there is a linear relationship between the annual runoff in the target catchment and the donor catchment, $y_i = \beta_0 + \beta_1 x_i + \epsilon_i$ for $i = 1...n$, where $y_i$ is the the annual runoff in the target catchment, $x_i$ is the annual runoff in the donor catchment, $\epsilon_i$ is normal distributed measurement error $\mathcal{N}(0, \sigma^2)$ with fixed (but typically unknown) variance $\sigma^2$, and $\beta_0$ and $\beta_1$ are coefficients that must be estimated. The linear relationship between the two catchments is developed by estimating $\beta_0$ and $\beta_1$ by minimizing the sum of least squares, $\sum_{i=1}^{n}(y_i - (\beta_0 + \beta x_i))^2$. Next, the linear relationship can be used to estimate the runoff at the target catchment $y_{n+1}, ..., y_{n+m}$ based on $x_{n+1}, ..., x_{n+m}$ with corresponding uncertainty estimates.

## 4 A geostatistical framework for exploiting long-term averages and short records

In this section we present the suggested Bayesian geostatistical framework for runoff interpolation. We start by developing a three staged hierarchical model for annual runoff consisting of a process model, an observation likelihood and prior distributions as described in Section 3.1. Next, we highlight two model properties that make the suggested framework different from most other methods used for interpolation in hydrology (Section 4.2) and explain how the framework is made computationally feasible (Section 4.3).

### 4.1 Hierarchical model for annual runoff

#### 4.1.1 True annual runoff (process models)

Let the spatial process $\{q_j(\boldsymbol{u}) : \boldsymbol{u} \in \mathcal{D}\}$ denote the runoff generating process at a point location $\boldsymbol{u}$ in the spatial domain $\mathcal{D} \in \mathcal{R}^2$ in year $j$. The true annual runoff generated at point location $\boldsymbol{u}$ in year $j$ is modeled as

$$q_j(\boldsymbol{u}) = \beta_c + c(\boldsymbol{u}) + \beta_j + x_j(\boldsymbol{u}) \qquad j = 1,..,r, \tag{4}$$

$$\pi(\beta_c) \sim \mathcal{N}(0, (10000 \text{ mm/year })^2);$$

$$\pi(\beta_j | \sigma_\beta) \sim \mathcal{N}(0, \sigma_\beta^2)$$

$$\pi(c(\boldsymbol{u}) | \rho_c, \sigma_c) \sim \text{GRF}(\rho_c, \sigma_c)$$

$$\pi(x_j(\boldsymbol{u}) | \rho_x, \sigma_x) \sim \text{GRF}(\rho_x, \sigma_x)$$

where $\beta_c$ is an intercept common for all years $j = 1,..r$ that models the average runoff in the study area over time, while $\beta_j$ is a year specific intercept that models the annual discrepancy from the long-term average runoff. Likewise is $c(\boldsymbol{u})$ a spatial effect that models the long-term spatial variability of runoff that is caused by climatic conditions in the study area, while $x_j(\boldsymbol{u})$ is a year specific spatial effect that models the spatial variability due to annual discrepancy from the climate. We emphasize that in this context, climate is for simplicity used as a collective term that describes both runoff generation caused by long-term weather-patterns *and* the runoff generation due to catchment characteristics like e.g. elevation and slope. The two spatial effects are modeled as Gaussian random fields (GRFs) with zero mean and stationary Matérn covariance functions with $\nu = 1$, given a range and a marginal variance parameter; $c(\boldsymbol{u})$ with range parameter $\rho_c$ and marginal variance $\sigma_c^2$, and $x_j(\boldsymbol{u})$ with range parameter $\rho_x$ and marginal variance $\sigma_x^2$. Furthermore, the spatial fields $x_j(\boldsymbol{u})$ for $j = 1,..,r$ are assumed to be independent realizations, or replicates, of the same underlying field to increase the identifiability of the model parameters (Ingebrigtsen et al., 2015). The same applies for the year-dependent intercepts $\beta_j$ that are all assigned a Gaussian prior $\mathcal{N}(0, \sigma_\beta^2)$ given the variance parameter $\sigma_\beta^2$. The intercept $\beta_c$ is assigned the weakly informative wide Gaussian prior $\mathcal{N}(0, (10000 \text{ mm/year })^2)$.

So far, runoff has been defined for point locations in space. However, runoff observations are linked to catchment areas, and we need to define the true average annual runoff generated inside a catchment $\mathcal{A}$. We suggest two alternative models: The first model is denoted **the areal model**. For the areal model, the true annual runoff in catchment $\mathcal{A}$ in year $j$ is given by the average point runoff over the catchment area, i.e.

$$Q_j(\mathcal{A}) = \frac{1}{|\mathcal{A}|} \int_{\boldsymbol{u} \in \mathcal{A}} q_j(\boldsymbol{u}) d\boldsymbol{u}, \tag{5}$$

where $|\mathcal{A}|$ is the catchment area and $q_j(\boldsymbol{u})$ is the point runoff from Equation (4). Interpreting annual runoff as an integral of point runoff ensures that the water balance is approximately preserved for the posterior mean runoff for any point in the landscape. Thus, the areal model is a model for mass-conserved hydrological variables. It also gives a realistic representation of distances and hence also the correlation between the catchments under study (see Equation (2)).

The second model for the annual runoff generated inside a catchment area is denoted **the centroid model**. For the centroid model, the true average annual runoff inside a catchment $\mathcal{A}$ in year $j$ is given by

$$Q_j(\mathcal{A}) = q_j(\boldsymbol{u}_\mathcal{A}), \tag{6}$$

where $q_j(\boldsymbol{u}_\mathcal{A})$ is the point runoff from Equation (4), and $\boldsymbol{u}_\mathcal{A}$ is the centroid of catchment $\mathcal{A}$. This alternative does not provide a preservation of the water balance for the posterior mean predicted runoff and can be used for any point referenced environmental variable. Distances are measured between catchment centroids, such that this method is more similar to the traditional Kriging-methods described in Section 3.3.

### 4.1.2   Observation likelihood

The true annual runoff from Section 4.1.1 is observed with uncertainty through streamflow data from $n$ catchments which we denote $\mathcal{A}_1, ..., \mathcal{A}_n$. We use the following model for the observed runoff $y_{ij}$ in catchment $\mathcal{A}_i$ in year $j$

$$y_{ij} = Q_j(\mathcal{A}_i) + \epsilon_{ij}; \quad i = 1, ..n, \quad j = 1, .., r. \tag{7}$$
$$\pi(y_{ij} | \sigma_y) \sim \mathcal{N}(Q_j(\mathcal{A}_i), s_{ij}\sigma_y^2).$$

Here, $Q_j(\mathcal{A}_i)$ is the true runoff from Equation (5) if we use the areal model, or the true runoff from Equation (6) if we use the centroid model. The error terms $\epsilon_{ij}$ are identically, independently distributed as $\mathcal{N}(0, s_{ij}\sigma_y^2)$ given the parameter $\sigma_y^2$, and we assume that each observation has its own uncertainty by scaling the variance parameter $\sigma_y^2$ with a fixed factor $s_{ij}$ that is further specified in Section 4.1.3.

Through the observation likelihood and the areal formulation of annual runoff from Equation (5), the areal model puts (soft) constraints on the annual runoff over the catchment areas of the gauged catchments. This way the areal model is able to influence the model to distribute the observed annual runoff within the catchment areas and not only at certain gauging points which is what the centroid model does. This represents a potential benefit for the areal model compared to the centroid model when modeling runoff. However, imposing constraints on areas also comes with a computational cost.

### 4.1.3   Prior models

According to the model specification in Section 4.1.1 and 4.1.2, there are 6 model parameters in the suggested hierarchical model for annual runoff, i.e. $(\sigma_y, \rho_c, \sigma_c, \rho_x, \sigma_x, \sigma_\beta)$. As we apply the Bayesian framework, these have to be given prior distributions, and we use knowledge based priors for most parameters. Note that since the priors are based on expert opinions about the study area, they are specific for the Norwegian dataset and should be modified before further use for other countries or environmental variables.

In the observation model for runoff in Equation (7), each observation is allowed to have its own measurement uncertainty by scaling the variance parameter $\sigma_y^2$, with a fixed scale $s_{ij}$. This makes sense because the spatial variability of mean annual runoff in Norway is large, with values ranging from around 400 mm/year to 4000 mm/year, and heteroscedastic errors can be

expected (Petersen-Øverleir, 2004). In the specification of the prior standard deviation $\sqrt{\sigma_y^2 s_{ij}}$, we assume that the measurement uncertainty for runoff increases with the magnitude of the observed value $y_{ij}$. Based on this we suggest the following scaling factors:

$$s_{ij} = (0.025 \cdot y_{ij}/1000)^2, \tag{8}$$

where $y_{ij}$ is the observed runoff in catchment $i$ in year $j$ in mm/year. The scaling factors are chosen based on what the data provider NVE believes are realistic standard deviations for the observed values, around 2.5% of the observed runoff. They are scaled down by 1000 to achieve appropriate values for $s_{ij}\sigma_y^2$. For the variance parameter $\sigma_y^2$, we use the penalized complexity prior (PC prior) suggested by Simpson et al. (2017). The PC prior is a prior constructed for the precision, i.e. the inverse of the variance, and the PC prior for the precision $\tau$ of a Gaussian effect $\mathcal{N}(0, \tau^{-1})$ has density

$$\pi(\tau) = \frac{\lambda}{2}\tau^{-3/2}\exp(-\lambda\tau^{-1/2}), \qquad \tau > 0, \quad \lambda > 0, \tag{9}$$

where $\lambda$ is a parameter that determines the penalty of deviating from a simpler base model. The parameter $\lambda$ can be specified through a quantile $u$ and a probability $\alpha$ by $\mathrm{Prob}(\sigma > u) = \alpha$, where $u > 0$, $0 < \alpha < 1$ and $\lambda = -\ln(\alpha)/u$. Here, $\sigma = 1/\sqrt{\tau}$ is the standard deviation of this Gaussian distribution. In our case, we specify the PC prior for $\sigma_y$ as

$$\mathrm{Prob}(\sigma_y > 1500 \text{ mm/year}) = 0.1. \tag{10}$$

Recall that $\sigma_y$ is scaled with $s_{ij}$ in the final uncertainty model such that a prior 95 % credible interval for the standard deviation $\sqrt{(\sigma_y^2 s_{ij})}$ for the observed runoff in catchment $\mathcal{A}_i$ year $j$ becomes $(0.04, 6)\%$ of the observed value $y_{ij}$. This is a quite strict prior that is chosen in order to influence the posterior observation uncertainty to be as low as possible. The reason behind this modeling choice is further described in Section 4.2. However, an observation uncertainty of 0.04-6 % of the observed value also corresponds quite well to what NVE knows about the measurement uncertainty for runoff in the study area. Percentages around 2.5% are as mentioned realistic.

For the spatial ranges $\rho_x$ and $\rho_c$ and the marginal variances $\sigma_x^2$ and $\sigma_c^2$ for the Gaussian random fields $x_j(\boldsymbol{u})$ and $c(\boldsymbol{u})$, we use the joint informative PC prior suggested in Fuglstad et al. (2019). It is specified through the following probabilities and quantiles:

$$\mathrm{Prob}(\rho_x < 20 \text{ km}) = 0.1, \qquad \mathrm{Prob}(\sigma_x > 2000 \text{ mm/year}) = 0.1,$$
$$\mathrm{Prob}(\rho_c < 20 \text{ km}) = 0.1, \qquad \mathrm{Prob}(\sigma_c > 2000 \text{ mm/year}) = 0.1.$$

The percentages and quantiles are chosen based on expert knowledge about the spatial variability in the area of interest. It is reasonable to assume that locations that are less than 20 km apart are correlated when it comes to runoff generation. In Norway the annual runoff varies from around 300 mm/year - 6000 mm/year such that a marginal standard deviation that is below 2000 mm/year is reasonable. The parameters of the climatic GRF $c(\boldsymbol{u})$ and the year dependent GRF $x_j(\boldsymbol{u})$ are given the same prior as it is difficult to identify if the spatial variability mainly comes from climatic processes or from annual variations. We also want the data to decide which of the two effects that dominates in the study area, and in this way detect hydrological spatial

stability or instability. Recall that the phrase *hydrological spatial stability* here is used to describe a variable and/or a study area that is characterized by an underlying spatial pattern that is repeated over time.

As specified in Section 4.1.1, the year specific intercepts $\beta_j$ for $j = 1, .., r$ are all assigned the same Gaussian prior $\mathcal{N}(0, \sigma_\beta^2)$ given the standard deviation parameter $\sigma_\beta$. The standard deviation $\sigma_\beta$ is given the PC prior from Equation (9) specified by the wide prior $P(\sigma_\beta > 10 \ \text{m/year}) = 0.2$. With this prior, the prior $95\%$ credible interval is approximately $(0.002, 40.5)$ m/year for the standard deviation $\sigma_\beta$ of $\beta_j$.

### 4.1.4 Feasible computation of catchment runoff for the areal model

In the areal model in Equation (5), the true runoff is modeled as the integral of point runoff over a catchment. To make the areal model computationally feasible, the integral is calculated by a finite sum over a discretization of the target catchment. More specifically, if $\mathcal{L}_i$ denote the discretization of catchment $\mathcal{A}_i$, the annual runoff in catchment $\mathcal{A}_i$ in year $j$ is calculated as

$$Q_j(\mathcal{A}_i) = \frac{1}{N_i} \sum_{\boldsymbol{u} \in \mathcal{L}_i} q_j(\boldsymbol{u}), \tag{11}$$

where $N_i$ is the number of grid nodes in the discretization $\mathcal{L}_i$. In the discretization of the catchments it is important that a subcatchment shares grid nodes with its overlapping catchment(s) such that the water balance can be preserved. In our analysis, we use a regular grid with 4 km spacing. It is also important that the discretization of the study area is fine enough to capture the rapid changes of annual runoff in the study area. Otherwise, non-realistic results such as negative runoff can occur.

### 4.1.5 Full model specification

Assuming that we observe runoff at $n$ stream gauges for $j = 1, .. r$ years and that $\mathcal{L}_\mathcal{D}$ contains all grid nodes in the discretization of the catchments $\mathcal{L}_{\mathcal{A}_i}$ for $i = 1, ..., n$, the areal model in Section 4.1.1 - 4.1.4 can be summarized as the following hierarchical geostatistical model:

$$\pi(\boldsymbol{y}|\boldsymbol{x}, \sigma_y) \sim \prod_{j=1}^{r} \prod_{i=1}^{n} (I\{\text{Observation} \ y_{ij} \ \text{is available}\} \cdot \mathcal{N}(Q_j(\mathcal{A}_i), s_{ij}\sigma_y^2)$$

$$+ 1 \cdot I\{\text{Observation} \ y_{ij} \ \text{is missing}\}) \quad [\text{Observation likelihood}]$$

$$\pi(\boldsymbol{x}|\boldsymbol{\theta}) = \pi\left(c(\boldsymbol{u_1}), ..., c(\boldsymbol{u_m})|\rho_c, \sigma_c\right) \cdot \pi(\beta_c) \tag{12}$$

$$\cdot \prod_{j=1}^{r} [\pi\left(x_j(\boldsymbol{u_1}), ..., x_j(\boldsymbol{u_m})|\rho_x, \sigma_x\right) \cdot \pi(\beta_j|\sigma_\beta)] \ [\text{Latent Model}]$$

$$\pi(\sigma_y, \boldsymbol{\theta}) = \pi(\rho_x, \sigma_x) \cdot \pi(\rho_c, \sigma_c) \cdot \pi(\sigma_\beta) \cdot \pi(\sigma_y) \quad [\text{Prior}]$$

where $\boldsymbol{y}$ is a vector containing all runoff observations $y_{ij}$ from all catchments $i$ and years $j$, $\boldsymbol{x}$ is a vector containing all latent variables, i.e. the intercepts $\beta_c$, $\beta_j$ and the GRFs $c(\boldsymbol{u}.)$ and $x_j(\boldsymbol{u}.)$ for all combinations of grid nodes $\boldsymbol{u_1}, ..., \boldsymbol{u_m} \in \mathcal{L}_\mathcal{D}$ and

years $j=1,..,r$. Furthermore, $Q_j(\mathcal{A}_i)$ is the true annual runoff that is modeled as a function of the latent field $\boldsymbol{x}$, while $I(\cdot)$ is an indicator function that is equal to 1 if its argument is true, and 0 otherwise allowing for missing data and short records of runoff. Finally, $\boldsymbol{\theta} = (\rho_x, \sigma_x, \rho_c, \sigma_c, \sigma_\beta)$. Together with $\sigma_y$ it contains all model parameters.

The centroid model is summarized as a hierarchical model similarly, except that the true annual runoff $Q_j(\mathcal{A}_i)$ is given by Equation (6) instead of Equation (11). This also means that the grid nodes $\boldsymbol{u_1}, ..., \boldsymbol{u_m}$ in the above hierarchical model must be replaced by $\boldsymbol{u}_{\mathcal{A}_1}, ..., \boldsymbol{u}_{\mathcal{A}_n}$, i.e. the locations of the centroids of the $n$ catchments under study.

The purpose of Bayesian inference is to estimate the posterior distributions of the latent variables $\boldsymbol{x}$ and the parameters $\boldsymbol{\theta}$ based on the observations $\boldsymbol{y}$ as described in Section 3.1. In this study, the resulting distributions are used to quantify the variable of interest, the catchment runoff $Q_j(\mathcal{A})$. By Equation (6) and Equation (11) we see that the catchment runoff is determined by the point runoff $q_j(\boldsymbol{u_1}), .., q_j(\boldsymbol{u_m})$ which is again determined by the latent field $\boldsymbol{x}$ through Equation (4). This means that in the process of estimating the catchment runoff $Q_j(\mathcal{A})$ we always estimate the point runoff $q_j(\boldsymbol{u})$ and the latent field $\boldsymbol{x}$ first. To clarify this process, consider Figure 10 that is presented later in the article. This shows the posterior mean runoff $q_j(\boldsymbol{u})$, or $\pi(\boldsymbol{x}|\boldsymbol{y})$ implicitly, for all points in the study area. From these point estimates, predictions for the *areal* model $Q_j(\mathcal{A})$ are obtained by taking the average of $q_j(\boldsymbol{u})$ over relevant grid nodes according to Equation (11). For the centroid model, a catchment areal prediction $Q_j(\mathcal{A})$ is obtained by simply extracting the value of $q_j(\boldsymbol{u}_\mathcal{A})$ at the catchment centroid $\boldsymbol{u}_\mathcal{A}$ according to Equation (6). From the point referenced predictions in Figure 10 we this way obtain catchment predictions like the ones presented later in e.g. Figure 7.

From the hierarchical formulation in (12) we also note that the framework takes the time dimension into account through multiplying the likelihood for annual runoff over different years $j = 1, ..r$. These years don't need to be consecutive, which allows for e.g. combining old measurements from closed stations with more recent data. Different years of data are connected through the constant climatic component $(c(\boldsymbol{u}) + \beta_c)$. Apart from this, there is no temporal dependency in the model that assumes correlation over time, and routing is not taken into account. This makes sense for our suggested application, as there is no prominent time dependency for annual runoff in Norway (see e.g. Figure 2b). Routing effects can typically be neglected for time-aggregated runoff variables for longer time scales. For shorter time scales for which routing has an impact, other spatio-temporal models should be considered, for example the one in Skøien and Blöschl (2007).

## 4.2 Two model properties and contributions

In this section we highlight and describe two of the model properties that make the suggested framework different from Top-Kriging and geostatistical interpolation methods that are typically used for hydrological applications.

### 4.2.1 Exploiting short records

The first property we highlight is how the model is particularly suitable for exploiting short records of runoff, and this holds for both the areal model and the centroid model. This property is already briefly addressed in the introduction, and is enabled because we simultaneously model several years of data with a spatial component $c(\boldsymbol{u})$ that is common for all years under study. The GRF $c(\boldsymbol{u})$ represents the long-term spatial variability of runoff. If most of the spatial variability can be explained by long-

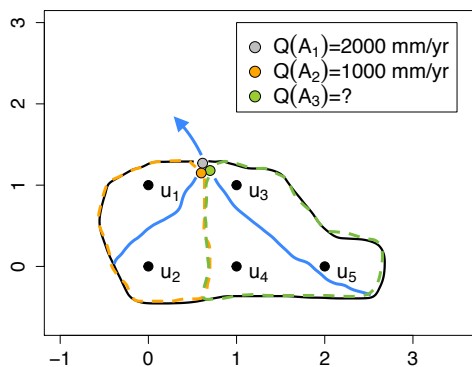

**Figure 6.** Conceptual, simplified figure of a river network, for which the involved catchments are discretized by 5 grid nodes $\boldsymbol{u}_1, .., \boldsymbol{u}_5$, and each grid node represent one areal unit. Catchment $\mathcal{A}_1$ contains all grid nodes $\boldsymbol{u}_1, .., \boldsymbol{u}_5$, Catchment $\mathcal{A}_2$ consists of grid nodes $\boldsymbol{u}_1$ and $\boldsymbol{u}_2$, while Catchment $\mathcal{A}_3$ consists of grid nodes $\boldsymbol{u}_3, \boldsymbol{u}_4$ and $\boldsymbol{u}_5$. Hence, this is a system of nested catchments where $\mathcal{A}_1$ covers $\mathcal{A}_2$ and $\mathcal{A}_3$. Assume that there are observations of annual runoff at the outlet of catchment $\mathcal{A}_1$ and catchment $\mathcal{A}_2$: $Q(\mathcal{A}_1)$=2000 mm/year and $Q(\mathcal{A}_2)$=1000 mm/year. Catchment $\mathcal{A}_3$ is ungauged. In order to fulfill water balance constraints of the areal model from Equation (11), imposed by the likelihood in Equation (7), the predicted mean annual runoff in catchment $\mathcal{A}_3$ must be around 2667 mm/year if we assume a low observation uncertainty.

term patterns, the marginal variance parameter $\sigma_c^2$ will dominate over the marginal variance parameter $\sigma_x^2$ of the annual GRF $x_j(\boldsymbol{u})$ (and the other model variances), i.e. $\sigma_c \gg \sigma_x$. Thus, a short record of runoff from an otherwise ungauged catchment will have a large impact also for predictions in years without data through $c(\boldsymbol{u})$. On the other hand, if most of the annual runoff is explained by year specific effects, $x_j(\boldsymbol{u})$ will dominate over $c(\boldsymbol{u})$ and short records will not have a large impact on the final
model. Hence, it is safe to include short records in the model regardless of the weather-patterns in the study area.

   Existing methods for exploiting short records are typically based on linear regression or computing the correlation between the runoff in the target catchment and one or several donor catchments, and in order to perform these procedures the short record must be of length larger than one (Fiering, 1963; Laaha and Blöschl, 2005). In the method we suggest, it is possible to include a short record of length one, and it is already shown for a smaller case study that this often is enough to see a large
improvement in the predictability of (annual) runoff for certain climates (Roksvåg et al., 2020).

### 4.2.2   The water balance constraints of the areal model

The second property we highlight only holds for the areal model, and is related to its mass-conserving properties and its ability to do more than smoothing: Runoff is in Equation (11) defined as an weighted sum of point runoff. Through Equation (11) and the likelihood defined in Equation (7), a (soft) constraint is put on the predicted annual runoff for the catchments for which we
have observations. This also has the consequence that the suggested model allows us to predict values that are larger than any of the observed values in the area of interest. As a conceptual example, consider the river network in Figure 6, where each black

node represents one areal unit in the discretization of the catchments. The observed runoff is 1000 mm/year in the subcatchment and 2000 mm/year in the surrounding larger catchment. That means that the constraints imposed by the observation likelihood and Equation (11) are the following:

$$1000 \text{ mm/year} = (q(\boldsymbol{u}_1) + q(\boldsymbol{u}_2))/2 + \text{uncertainty} \tag{13}$$

5    $2000 \text{ mm /year} = (q(\boldsymbol{u}_1) + q(\boldsymbol{u}_2) + q(\boldsymbol{u}_3) + q(\boldsymbol{u}_4) + q(\boldsymbol{u}_5))/5 + \text{uncertainty}.$

As described in Section 4.1.3 we use a quite strict prior on the observation uncertainty. This is done to try to force the above soft constraints to be stronger. By solving this system of equations, it can be shown that the predicted value in the ungauged catchment $\mathcal{A}_3$ must be

10    $\hat{Q}(\mathcal{A}_3) = \dfrac{q(\boldsymbol{u}_3) + q(\boldsymbol{u}_4) + q(\boldsymbol{u}_5)}{3} + \text{uncertainty} = 2667 \text{ mm/year} + \text{uncertainty}$

where the above uncertainty term is determined by the observation uncertainties for a fixed set of ranges and marginal variances. Hence, as long as the observation uncertainty is sufficiently low, the predicted runoff in the unobserved area $\mathcal{A}_3$ in Figure 6 is close to 2667 mm/year which is larger than any of the observed values. This example illustrates that the areal model is able to go beyond smoothing through its runoff constraints. This makes the model different from many other interpolation methods, that only rely on spatial smoothing. For the above example, such methods would typically produce a prediction between 1000 15    mm/year and 2000 mm/year.

The full areal model is of course slightly more complicated than the simple example above, as prior distributions, covariance calculations and spatial ranges must be taken into account. However the simple example illustrates the general idea of how the observation likelihood interprets the areal observations and constrains runoff. That the full areal model actually is able to 20    conserve mass in practice, is demonstrated for a real case example from Norway in Roksvåg et al. (2020).

The constraints in Equation (13) also show how the areal model ensures consistent predictions over nested catchments: As the predicted runoff in the main catchment $\mathcal{A}_1$ can be expressed as a weighted sum of the predicted runoff in all its subcatchments depending on catchment areas, i.e. as $\hat{Q}(\mathcal{A}_1) = \frac{2}{5}\hat{Q}(\mathcal{A}_2) + \frac{3}{5}\hat{Q}(\mathcal{A}_3)$, the water balance can not be violated for the predicted runoff for any of the catchments in Figure 6. This means that the equations in (13) correspond to water balance constraints.

25    Compared to Top-Kriging, both Top-Kriging and the proposed method assume that the underlying variable is linearly aggregated and mass-conserved when performing covariance calculations. Top-Kriging is also able to predict larger values than any of the observed values by allowing negative Kriging weights. However, Top-Kriging does not use constraints to ensure that the mass balance is preserved over nested catchments, as in the above example. Consequently, the Top-Kriging predictions can more easily violate the water balance, which can have both benefits and drawbacks depending on the target variable and 30    the problem we are trying to solve. Another hydrological model that uses water balance constraints, not unlike the proposed method, is the Kriging approach in Sauquet et al. (2000) where mass balance constraints are introduced as additional constraints in the Kriging system of equations.

## 4.3 Inference

In order to make the framework described in Section 4 computationally feasible, some simplifications of the suggested models are necessary. In general, statistical inference on models including GRFs is slow when the number of target locations is large because matrix operations on dense covariance matrices are required. The computational complexity is particularly large for the areal model, because each grid node in the discretization of the catchments can be regarded as a new target location, and because it includes soft constraints. To solve the computational issues for the centroid and areal model, we utilize that a GRF with a Matérn covariance function can be expressed as the solution of a specific Stochastic partial differential equation (SPDE) (Lindgren et al., 2011). This SPDE can be solved by using the finite element method (see e.g. Brenner and Scott (2008)), and the result is a Gaussian Markov random field (GMRF). Working with GMRFs is convenient because GMRFs have precision matrices (inverse covariance matrices) that typically are sparse with more zero elements, and efficient algorithms are available for sparse matrix operations (see e.g Rue and Held (2005)). In this work, both GRFs $x_j(\boldsymbol{u})$ and $c(\boldsymbol{u})$ are approximated by GMRFs.

Another challenge with the suggested models, is that we suggest Bayesian models that include a large number of parameters for which the marginal distributions must be estimated. Traditionally, Bayesian inference is done by using Markov chain Monte Carlo-methods (MCMC), but inference can be slow when the dimension of the problem is large (Gamerman and Lopes, 2006). These challenges are met by modeling runoff as a latent Gaussian model (LGM). That is, the latent part $\boldsymbol{x}$ of the hierarchical model in 4.1.5 consists of only Gaussian distributions. More specifically, the prior distributions for $c(\boldsymbol{u})$ and $x_j(\boldsymbol{u})$ are modeled as GRFs, and the prior distributions for $\beta_j$ and $\beta_c$ are Gaussian given the model parameters (see the equations in (4)). This is convenient, because it allows us to use integrated nested Laplace approximations (INLA) to make inference and predictions. INLA is a tool for making Bayesian inference for LGMs (Rue et al., 2009) and represents a fast and approximate alternative to MCMC algoithms. The INLA approach is based on approximating the marginal distributions by using Laplace or other analytic approximations, and on numerical integration schemes. The main computational tool is the sparse matrix calculations described in Rue and Held (2005), such that in order to work fast, the latent field of the LGM should be a GMRF with a sparse precision matrix. This requirement is fulfilled through the SPDE approach as already outlined.

INLA in general provides approximations of very high accuracy for most models (Rue et al., 2009; Martino et al., 2011; Eidsvik et al., 2012; Huang et al., 2017), but has faced problems for some (more extreme) models with binomial or Poisson data (Fong et al., 2009; Ferkingstad and Rue, 2015). For Gaussian likelihoods however, INLA is exact up to numerical integration error. As we use Gaussian likelihoods in this work, we can thus expect INLA to give reliable approximations. The SPDE approach also provides accurate approximations (Lindgren et al., 2011; Huang et al., 2017), but it is important that the mesh involved in the finite element method computations is sufficiently dense relative to the spatial variability and range in the study area.

Because of the high computational speed and accuracy, the INLA and SPDE framework has become quite common to use within different fields of science. See for example Khan and Warner (2018); Opitz et al. (2018); Yuan et al. (2017); Guillot et al. (2014); Ingebrigtsen et al. (2014); Bakka et al. (2018). We refer to the R-package `r-inla` for a user-friendly interface

for applying INLA and the SPDE approach to spatial modeling. In particular, Moraga et al. (2017) is recommended for a description of how a model with (catchment) areal data can be implemented in `r-inla`. Furthermore, we have made code for the centroid model available on *http://www.github.com/tjroksva/runoffinterpolation* (doi: 10.5281/zenodo.3630348) with example data from the catchments in Figure 1a.

## 5   Model evaluation

The main objectives of this article are to (1) evaluate the new framework's ability to fill in missing annual runoff observations and to (2) predict mean annual runoff for catchments with varying record lengths. By this we also want to (3) demonstrate the potential added value of including short runoff records in the modeling compared to not using them. In this section we present the experimental set-up and the evaluation criteria used to address our research questions.

### 5.1   Experimental set-up for infill of missing annual observations (1996-2005)

To assess the framework's ability to fill in missing values of annual runoff, we do interpolation of runoff for the 10 hydrological years 1996-2005 for the 180 fully gauged catchments shown in Figure 1a. This is done both for series of annual runoff, and for the annual series of monthly runoff for January, April and June described in Section 2.

The annual time series of monthly runoff are included in the analysis in order to demonstrate the framework's properties for hydrological variables or areas that are driven by more unstable hydrological processes. For the annual series of monthly runoff, the models from Section 4 are specified as before: Considering predictions for January, $Q_j(\mathcal{A}_i)$ in Equation (5) represents the true runoff in January for catchment $\mathcal{A}_i$, year $j$, such that the GRF $c(\boldsymbol{u})$ represents the long-term spatial variability in January. Likewise, the GRF $x_j(\boldsymbol{u})$ represents the annual discrepancy from the climate in January, and $y_{ij}$ is the observed runoff in January for catchment $\mathcal{A}_i$ year $j$. The models for June and April are specified similarly, and for simplicity we use the same prior distributions for all experiments.

In our assessment of the framework's predictive performance for infill of missing annual observations, the three following methods are compared:

**Top-Kriging:** Spatial interpolation with Top-Kriging. For Top-Kriging each year (1996-2005) is interpolated independently from other years. Short records on an annual (or monthly) scale don't have an impact on years without data. The default covariance function (or variogram) in the R package `rtop` was fitted as this gave the most accurate results. This is a multiplication of a modified exponential and fractal variogram model, the same model as used in Skøien et al. (2006).

**Areal model:** Spatial interpolation with the model defined in Section 4 with true annual runoff given by the areal model from Equation (11). That is, the annual runoff in a catchment is interpreted as the average point runoff over the catchment area. All years are modeled simultaneously (1996-2005) such that short records of data can influence years without data.

**Centroid model:** Spatial interpolation with the model defined in Section 4 with true annual runoff given by the centroid model from Equation (6). That is, annual runoff is interpreted as a process linked to point locations in space (the catchment centroids), and not to catchment areas. All years are modeled simultaneously (1996-2005) such that short records of data can

influence years without data.

The predictive performance of the three methods is evaluated by cross-validation: The 180 catchments in Norway were divided into 20 groups or folds, each containing 9 catchments. In turn each group was left out, and annual or monthly runoff predictions were performed for these so-called target catchments by using observations from the catchments in the other groups. That is, we predict runoff for 1996-2005 for 9 target catchments at once by using data from the remaining 171 fully gauged catchments, and repeat the process for all 20 cross-validation folds. To evaluate and compare the three methods described above, we do the following two tests:

**UG (ungauged):** Assess the methods' ability to fill in missing values for ungauged catchments (denoted UG). That is, the target catchments are treated as totally ungauged, and all their observations are left out of the dataset when the predictions for 1996-2005 are performed.

**PG (partially gauged):** Assess the methods' ability to fill in missing values for partially gauged catchments (denoted PG). Each of the 9 target catchments in the cross-validation group is allowed to have one annual observation of runoff. That is, a short record of length one from the target catchment is included in the observation likelihood in addition to the full data series of runoff from the catchments in the other cross-validation folds. The short record is drawn randomly from the ten years of observations available for each target catchment. We perform predictions for 9 partially gauged target catchments at once, for all 10 study years (for which one of them is observed for each catchment), and repeat the process for all 20 cross-validation folds.

To make the results comparable, we use the same cross-validation groups for both experiments (UG and PG) and methods (Top-Kriging, areal model and centroid model), and remove the same set of annual observations for PG across methods. For the PG-case, we also compare our models to a method for exploiting short records from the target catchment. The method we choose for comparison is simple linear regression, and we perform linear regression for the PG-case as follows:

**Linear regression:** The closest catchment in terms of catchment centroid is used as a donor catchment and only catchments outside the target catchment's cross-validation group can be considered. Two annual observations between 1996 and 2005 are randomly drawn from the target catchment, and data from the donor catchment and target catchment are used to fit a linear regression model on the form $y_i = \beta_1 x_i + \epsilon_i$. Next, the model is fitted as described in Section 3.4, and used to predict runoff for the target catchment for 1996-2005 (where two of the years are observed). The reason for using a short record of length two instead of one, is that at least two observations are required to fit a linear regression model with uncertainty. Also mark that we have omitted the intercept $\beta_0$ in the regression model, such that we only have two unknown variables ($\beta_1$ and $\sigma^2$). We emphasize that estimating the variables statistically based on only two pair of observations might not give the most reliable estimates, particularly when considering $\sigma^2$. However, the linear regression results can provide an intuition about the target variable and how correlated the observation locations are over time. A good performance for linear regression in this study, suggests that the spatial pattern for the target variable is very stable and that a fair prediction can be obtained by simply using the ratio of runoff between a target catchment and a donor catchment for any chosen year to develop a linear relationship between the two catchments. Hence, a short record can be very valuable. Motivated by this, linear regression is treated as

an indicator for when our geostatistical method can be expected to perform particularly well, rather than as a recommended method for record augmentation when having only two observations.

## 5.2 Experimental set-up for predictions of mean annual runoff (1981-2010)

To assess the framework's ability to estimate mean annual runoff, we use annual data from 1981-2010 from the 260 catchments in Figure 5. Recall that these catchments have at least one observation of mean annual runoff between 1981 and 2010, but only 83 of them are fully gauged. This was different from the experiments described in Section 5.1, where all the test catchments were fully gauged before the cross-validation was performed.

For this experiment, we again compare the performances of Top-Kriging, the areal and the centroid model. The areal model and the centroid model are fitted for several years of annual runoff simultaneously, as before. As a predictor for the mean annual runoff, we use the posterior distribution of the climatic part of the model. This is given by $c(\boldsymbol{u}_{\mathcal{A}}) + \beta_c$ for the centroid model, where $\boldsymbol{u}_{\mathcal{A}}$ is the centroid of the catchment $\mathcal{A}$ of interest. For the areal model it is given by the average $c(\boldsymbol{u}_i) + \beta_c$ over the grid nodes $\boldsymbol{u}_i$ in the discretization of the target catchment. Note that the climatic part of the model must be re-estimated for each experiment or cross-validation fold.

In order to interpolate mean annual runoff by using Top-Kriging, we have to compute the mean annual runoff based on the annual observations for all catchments before running the analysis. For catchments with less than 30 annual observations we use the average of the 1-29 available observations as an approximation for the mean annual runoff for 1981-2010. Next, the mean annual runoff is interpolated by using Top-Kriging where the uncertainty of the observations is specified as a function of record length. This is the suggested approach from Skøien et al. (2006) for including short records in the Top-Kriging framework. We set the observation variance for a catchment with record length $m$ to $\hat{\sigma}^2/m$, where $\hat{\sigma}$ is the average empirical standard deviation for the observed annual runoff taken over the 83 fully gauged catchments in our dataset, in this case $\hat{\sigma} = 336$ mm/year. For the Top-Kriging experiments, we fit the same covariance model as in Section 5.1.

The areal and centroid model and Top-Kriging are again evaluated by cross-validation. The 83 fully gauged catchments from Figure 5 were divided into 4 folds containing 20, 20, 20 and 23 catchments respectively, and in turn observations from each fold were removed and predicted. This was done for varying record lengths for the target catchments, more specifically when 0, 1, 3, 5 or 10 randomly drawn annual observations from the target catchments were included in the likelihood. We denote these settings UG, PG1, PG3, PG5 and PG10. Note that while we only are able to assess the predictive performance for the 83 fully gauged catchments in Figure 5, data from the remaining 177 partially gauged catchments in Figure 5 are used in the observation sample. This is in addition to the data from the fully gauged catchments from the other folds.

## 5.3 Evaluation scores

To evaluate the predictions we use the root mean squared error (RMSE) and the continuous rank probability score (CRPS). Having $m$ pairs of observations and predictions, the RMSE is computed as

$$\text{RMSE} = \sqrt{\frac{1}{m}\sum_{j=1}^{m}(y_j^* - \hat{y}_j^*)^2},$$

where $y_j^*$ is the observed value and $\hat{y}_j^*$ is the corresponding predicted value. In our analysis, the posterior mean is used as a the predicted value for the areal and centroid model.

The CRPS is defined as

$$\text{CRPS}(F, y^*) = \int_{-\infty}^{\infty}(F(s) - 1\{y^* \leq s\})^2 ds,$$

where $F()$ is the predictive cumulative distribution and $y^*$ is the actual observation (Gneiting and Raftery, 2007). For the
10 methods we test (areal, centroid, Top-Kriging and linear regression), $F()$ is a Gaussian distribution with mean equal to the predicted value and standard deviation equal to the standard deviation of the prediction.

For the experiments related to infill of individual years, the CRPS and RMSE are first computed for each of the 180 catchments in the dataset based on 10 pairs of predictions and observations. The average RMSE and CRPS over all catchments are used as a summary scores. For the experiments related to predictions of mean annual runoff, there is only one (mean annual)
prediction for each catchment, and the RMSE and CRPS over all catchments are reported. Both the CRPS and the RMSE are negatively oriented such that low scores mean better predictions.

To be able to compare the RMSE and CRPS across methods we use a paired Wilcoxon Signed-Rank Test (Siegel, 1956). This is a non-parametric test that does not require normal distributed data. The null hypothesis of the test is that the median difference between pairs of data (in this case pairs of RMSE or CRPS values) follows a symmetric distribution around zero.
The alternative hypothesis is that the difference between the data pairs does not follow a symmetric distribution around zero. If the null hypothesis is rejected, it indicates that one of the methods gives a significantly smaller RMSE or CRPS than another method.

In addition to the RMSE and the CRPS, we report the 95 % coverage of the experiments. The 95 % coverage is computed by calculating the amount of the actually observed runoff values that are within the corresponding 95 % posterior prediction
intervals. Here, we make posterior prediction intervals for Top-Kriging and linear regression by assuming that the predictions are Gaussian. A 95 % coverage close to 0.95 is optimal and indicates that the model provides an accurate representation of the underlying uncertainty.

We also want to compare our mean annual runoff results with other studies of mean annual runoff, more specifically the studies collected in Blöschl et al. (2013). In Blöschl et al. (2013), the absolute normalized error (ANE) and the squared
correlation coefficient ($r^2$) are used as evaluation scores. The ANE is computed as

$$\text{ANE} = \frac{|\hat{y}^* - y^*|}{y^*}, \tag{14}$$

where $y^*$ and $\hat{y}^*$ are the observed and predicted value as before. The ANE divides the absolute difference between the actual observation $y^*$ and corresponding prediction $\hat{y}^*$ by the observed runoff, and is therefore scale independent. An ANE close to zero corresponds to an accurate prediction.

Finally, the squared correlation coefficient between $m$ pairs of observations and predictions is computed as

$$r^2 = (\text{Cor}\{(y_1, ..., y_m), (\hat{y}_1^*, ..., \hat{y}_m^*)\})^2, \tag{15}$$

where $\text{Cor}(\cdot, \cdot)$ denotes the Pearson correlation. An $r^2$ close to 1 indicates a high correlation between the predicted and observed values.

## 6 Results

### 6.1 Predictions for individual years (1996-2005)

We now present the results related to the framework's ability to predict runoff for individual, missing years for the annual time series of annual and monthly runoff for a 10 year period (1996-2005). First, we present the results for the ungauged catchments (UG), before we proceed to the partially gauged catchments (PG) that have short records of length one.

#### 6.1.1 Infill for ungauged catchments (UG)

For the ungauged case (UG), the target catchments are treated as totally ungauged for the ten study years 1996-2005, and missing values are predicted both for annual and monthly runoff. In Figure 7 the resulting average predicted *annual* runoff in southern Norway is presented for Top-Kriging, the areal model and the centroid model. The three methods give similar results for the posterior mean, and all are able to reproduce the true spatial pattern of annual runoff. Furthermore, the RMSE plots in Figure 7 show that the three methods succeed and fail for many of the same catchments. Here, we should keep in mind that the RMSE is scale dependent and might not give the best impression of the relative performance across the study area. However, we note that many of the catchments with high RMSE values typically are small catchments located in western Norway. We will come back to these catchments in Section 6.1.2 to see how the predictions here were affected when including a short record.

Considering the posterior standard deviation in Figure 7, we notice that Top-Kriging and the areal model provide a similar quantification of the predictive uncertainty. Top-Kriging and the areal model take the nestedness of catchments into account by interpreting the runoff data as areal referenced, providing a predictive standard deviation of runoff that varies with the size of the target catchment: Figure 7 shows that smaller catchments typically have a larger predictive uncertainty, which is reasonable. For the centroid model, runoff observations are point referenced and weighted independently of catchment size. Consequently, the predictive uncertainty only depends on how the centroids of the observed catchments are distributed in space, and decreases in areas where there are clusters of data. The predictive uncertainties provided by Top-Kriging and the areal method are thus more intuitive and realistic considering the process we are studying. The latter is also reflected in the coverage percentages

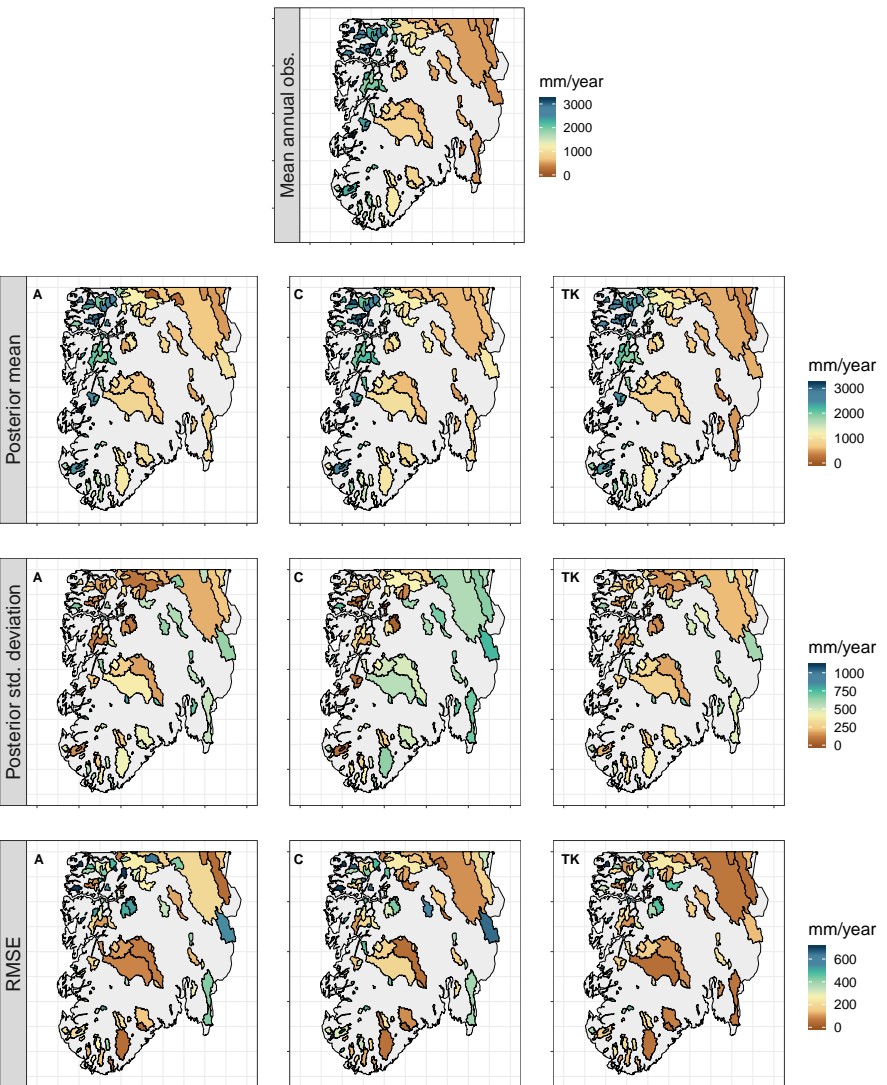

**Figure 7.** Average posterior mean for $Q_j(\mathcal{A})$, average posterior standard deviation for $Q_j(\mathcal{A})$ and average RMSE for each catchment for predictions of missing annual observations in southern Norway for $j = 1,..10$ for the areal model (A,left), the centroid model (C, middle) and Top-Kriging (TK, right) when the target catchments are treated as ungauged (UG). The observed mean annual runoff is also included as a reference (first plot).

presented in Table 1. The coverages show the amount of the actual observations that were captured by the corresponding 95 % prediction intervals, and these are slightly closer to 0.95 for Top-Kriging and the areal model compared to the centroid model.

Table 1 also presents the summary scores for the predictive performance for infill of missing values for ungauged catchments for all methods. According to the RMSE and CRPS, Top-Kriging is a better interpolation method than our two suggested

**Table 1.** Predictive performance for predictions of missing annual values in ungauged catchments (UG) and partially gauged catchments (PG) for the areal model, centroid model, Top-Kriging (TK) and simple linear regression (LR). The best performance in each row is marked in bold. The RMSE and CRPS were compared across methods by using a one sided paired Wilcoxon Signed-Rank test for assessing the significance of the results. Results that were significantly better than other results are marked with stars.

| | | RMSE [mm/year] | | | | CRPS [mm/year] | | | | Coverage 95 % | | | |
|------|---------|--------|--------|--------|--------|--------|--------|--------|------|-------|--------|------|------|
| Case | Dataset | Areal | Centr. | TK | LR | Areal | Centr. | TK | LR | Areal | Centr. | TK | LR |
| UG | Annual | 337 | 343 | **310** * | - | 242 | 249 | **225*** | - | 0.92 | 0.91 | **0.94** | - |
| UG | January | 39 | 37 | **36** * | - | 26 | 25 | **24*** | - | 0.92 | 0.89 | **0.93** | - |
| UG | April | 38 | 38 | **37** | - | 25 | 25 | **24** | - | 0.89 | 0.85 | **0.93** | - |
| UG | June | 87 | 96 | **82** * | - | 59 | 67 | **56** * | - | **0.91** | 0.84 | **0.91** | - |
| PG | Annual | **171** ** | 184 ** | 290 | 178 ** | **105*** | 113 ** | 201 | 240 | **0.95** | 0.94 | **0.95** | 0.96 |
| PG | January | **30*** | **30*** | 33 | 61 | **19*** | 20** | 21 | 88 | 0.91 | 0.89 | 0.91 | **0.95** |
| PG | April | **31*** | 33 ** | 35 | 50 | **20** ** | 21 ** | 22 | 94 | 0.86 | 0.84 | **0.94** | 0.96 |
| PG | June | **55** ** | 63 ** | 78 | 95 | **35** ** | 42 ** | 50 | 136 | 0.90 | 0.84 | 0.93 | **0.96** |

* The RMSE/CRPS is significantly lower than the RMSE/CRPS of the *areal and the centroid model* on a 5 % significance level.

** The RMSE/CRPS is significantly lower than the RMSE/CRPS of *Top-Kriging* on a 5 % significance level.

methods for ungauged catchments. However, the boxplots in Figure 8 illustrate the distribution of RMSE for all catchments, and we see that on a monthly scale, the difference between Top-Kriging and the two other methods is quite low from a practical point of view. For January and April the differences are almost negligible.

### 6.1.2 Infill for partially gauged catchments (PG)

For the partially gauged (PG) case, each target catchment is allowed to have a short record of length one for Top-Kriging, the areal and centroid model, and length two for linear regression. Before we comment the results from the cross-validation in Table 1 and Figure 9, we consider the posterior estimates of the range parameters ($\rho_x$ and $\rho_c$) and the marginal variance parameters ($\sigma_x$ and $\sigma_c$) of the year-specific GRF $x_j(\boldsymbol{u})$ and the climatic GRF $c(\boldsymbol{u})$ for our four datasets. These are shown in Table 2 and indicate how much of the spatial variability that is captured by the climatic GRF relative to the annual GRF. In

particular, if $\sigma_c$ dominates over $\sigma_x$, it suggests hydrological spatial stability.

The estimates in Table 2 show that the hydrological spatial stability is largest for June and for annual runoff, as expected from the time series in Figure 2b and Figure 4. Here, the posterior mode for $\sigma_c$ is more than twice as large as the posterior mode for $\sigma_x$ for both the areal and the centroid model. Furthermore, we see in Table 2 that the climatic range $\rho_c$ is only around 12 % of the annual range $\rho_x$. In Figure 10 we have illustrated the spatial pattern these parameters give for annual predictions

in 1997 and 1998 for the whole study area. We see that the annual runoff for 1996 and 1997 have the same spatial pattern, and that this spatial pattern mostly originates from $c(\boldsymbol{u})$, i.e. climatic conditions including catchment characteristics. The trend we see in Figure 10 can also be seen for the remaining eight years in the dataset (1996,1999-2005), as well as for June. A spatial

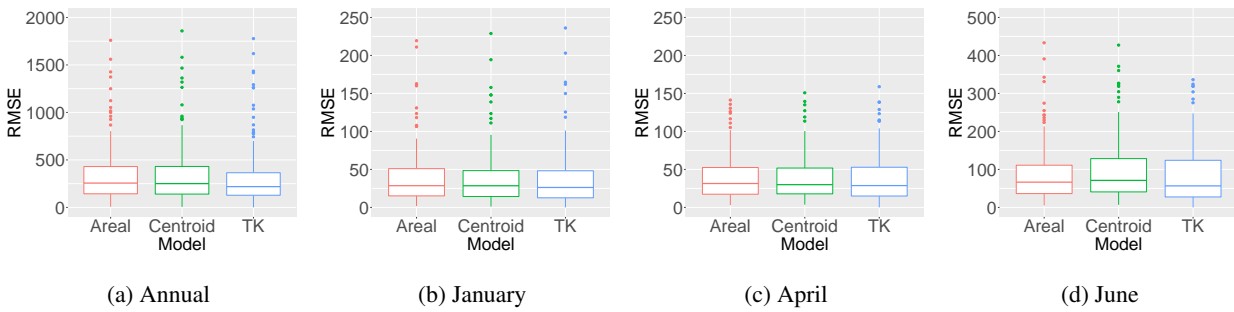

|  (a) Annual | (b) January | (c) April | (d) June |

**Figure 8.** Distribution of RMSE [mm/year] for infill of missing values for all catchments and years (1996-2005) when the target catchments are treated as ungauged (UG) in the cross-validation for the areal, centroid and Top-Kriging (TK) method. The lower and upper quartiles correspond to the first and third quartiles (the 25th and 75th percentiles), and the whiskers extend from the quartiles no further than 1.5· IQR, where IQR is the distance between the first and third quartile. The same applies for all boxplots presented in this paper.

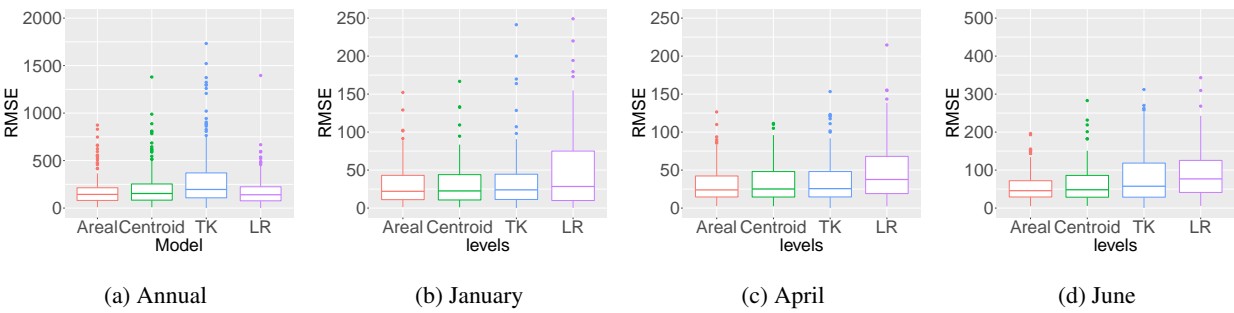

|  (a) Annual | (b) January | (c) April | (d) June |

**Figure 9.** Distribution of RMSE [mm/year] for infill of missing values for all catchments and years (1996-2005) for the areal model, centroid model and for Top-Kriging (TK) when the target catchments are treated as partially gauged (PG), i.e. a short record of length one from the target catchment is included in the observation likelihood in the cross-validation. Results for linear regression (LR) are also included here.

pattern like this, with $\sigma_c \gg \sigma_x$ and $\rho_c < \rho_x$, suggests that the information gain from neighboring catchments further away is low for an ungauged catchment, and that the potential information stored in short records is high.

For January however the situation is different: The posterior mode of $\sigma_x$ is larger than the posterior mode of $\sigma_c$ for both the areal and the centroid model. The parameters show that for January, year-specific effects explain a larger part of the spatial variability. This can be due to a more unstable hydrological setting with runoff driven by snow accumulation and snow melt. For April, we have that $\sigma_c > \sigma_x$, but $\sigma_c$ is less dominant than for June and for the annual data.

In the areal and centroid model, the inclusion of a short record changes the climatic spatial field $c(\boldsymbol{u})$, and hence the predictions can be considerably changed for the target catchment if the climatic effect is strong. The parameter values thus suggest that the gain of including short records is lower for April and January compared to the other two datasets. This is confirmed by comparing the RMSE and CRPS for the areal and the centroid model for the partially gauged case (PG), to the RMSE and CRPS obtained for the ungauged case (UG) in Table 1. For all datasets, the RMSE and CRPS for our two models

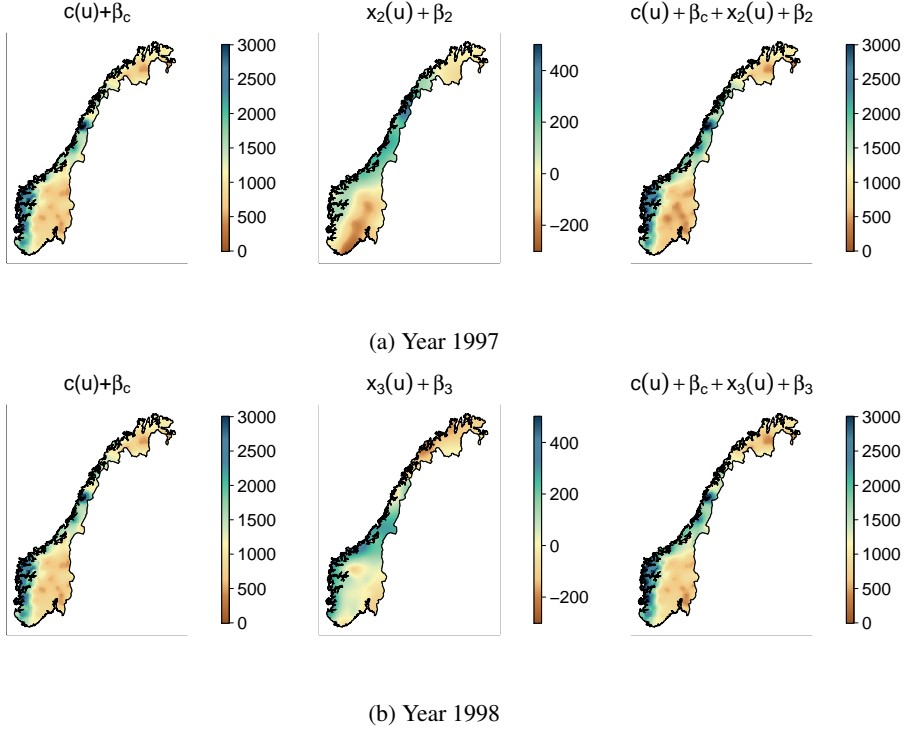

(a) Year 1997

(b) Year 1998

**Figure 10.** From left to right: The climatic part of the model (common for all years), the annual (year dependent) part of the model and the full model $q_j(\boldsymbol{u})$ for annual runoff in 1997 and 1998 [mm/year]. Note that the scales of the middle plots only cover 25 % of the scale of the other plots. We see that most of the spatial variability of annual runoff for 1997 and 1998 can be explained by climatic effects, and that the climatic range $\rho_c$ is considerably smaller than the year specific range $\rho_x$. The results above are produced by the centroid model, and plots similar to these are behind all results presented for the areal and centroid model in this article.

are reduced for PG compared to UG, but the reduction is lower for January and April than for June and the annual data. For the annual predictions, the RMSE and CRPS are reduced by more than $50\%$ when a short record of length one from the target catchment is included in the observation likelihood. The reduction for June is also remarkable (around 35-40 %), while the reduction for January and April is moderate (around 13-20 %). The results hold for both the areal and centroid model, but the

5    areal model seems to be somewhat better than the centroid model in terms of exploiting short records of data from the target catchment. This is again related to the parameter estimates in Table 2, where we see that $\sigma_c$ dominates more over $\sigma_x$ in the areal model than in the centroid model.

Considering the results for Top-Kriging in Table 1, we only obtain a small reduction in the RMSE and CRPS for the partially gauged case (PG) compared to the ungauged case (UG). This is because Top-Kriging treats each year of data independently

10   when considering infill of missing annual data. A reduction in RMSE and CRPS is only seen for the specific year with extra data. This is different from our framework where several years of data are modeled simultaneously. The evaluation scores in Table 1 and the boxplots in Figure 9 clearly show that our two suggested methods outperform Top-Kriging for the partially

**Table 2.** The posterior mode of the range parameters $\rho_c$ and $\rho_x$ and the marginal standard deviations $\sigma_c$ and $\sigma_x$ of the climatic and the annual GRFs $c(\boldsymbol{u})$ and $x_j(\boldsymbol{u})$ for the areal model (upper) and centroid model (lower). The posterior standard deviations of the parameters are shown in parenthesis as a measure of the uncertainty. The mode and standard deviations vary between the experiments and groups in the cross-validation, and the values given here are the mean over all folds and experiments (UG and PG). The spatial effect that dominates (annual or climatic) is marked in bold.

| Areal model | $\rho_c$ [km] | $\rho_x$ [km] | $\sigma_c$ [mm/year] | $\sigma_x$ [mm/year] |
|---|---|---|---|---|
| Annual | 58 (7) | 476 (65) | **880** (56) | 267 (23) |
| January | 31 (7) | 247 (22) | 72 (6) | **83** (4) |
| April | 77 (14) | 239 (32) | **75** (6) | 48 (3) |
| June | 43 (5) | 153 (22) | **181** (9) | 75 (3) |

| Centr. model | $\rho_c$ [km] | $\rho_x$ [km] | $\sigma_c$ [mm/year] | $\sigma_x$ [mm/year] |
|---|---|---|---|---|
| Annual | 89 (12) | 659 (77) | **750** (57) | 263 (22) |
| January | 82 (15) | 369 (44) | 60 (6) | **88** (7) |
| April | 118 (19) | 375 (51) | **66** (4) | 52 (5) |
| June | 69 (9) | 335 (47) | **161** (12) | 71 (6) |

(a) Annual predictions.

(b) Predictions for April

**Figure 11.** All observations for 1996-2005 compared to the corresponding predictions for the ungauged case (UG, left) and the partially gauged case (PG, right) for annual predictions (Figure 11a) and for April (Figure 11b). The predictions are performed by the areal model. The straight line represents a perfect correspondence between prediction and actual observation.

gauged case for annual predictions and monthly predictions in June, which were the two time-scales with most hydrological spatial stability ($\sigma_c \gg \sigma_x$). For January and April the three models are more similar in predictive performance.

For the PG case, we also compare the areal and the centroid model to simple linear regression. According to Table 1 and Figure 9 linear regression performs quite well for the annual data, which represent the most hydrologically spatially stable dataset. Linear regression actually provides the second lowest RMSE of all four methods for annual predictions. However, recall that a short record of length two from the target catchment is needed to use this method, while our areal model performs slightly better with a short record of length one (and observations from other neighboring catchments). For January, April and June, linear regression is outperformed by the three other methods in terms of RMSE and CRPS (Table 1).

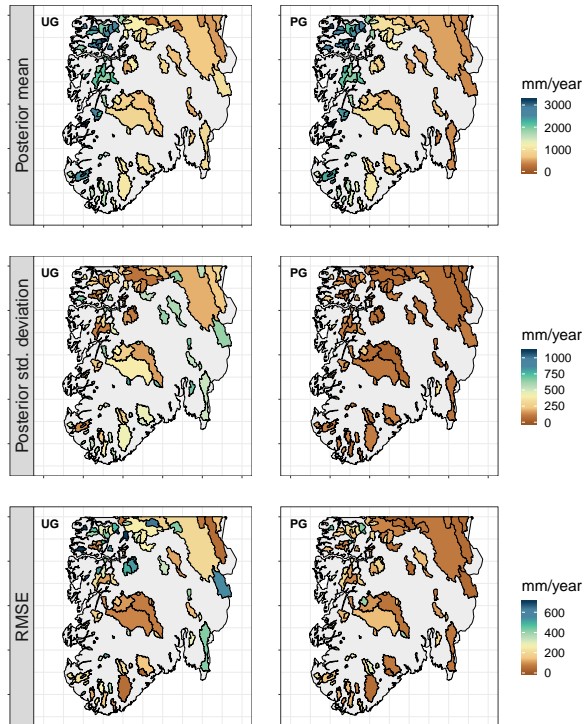

**Figure 12.** Average posterior mean $Q_j(\mathcal{A})$ (upper), average posterior standard deviation (middle) and RMSE (lower) for $j = 1,..10$ for predictions of missing annual observations for the areal model for the ungauged case (UG, left) and the partially gauged case (PG, right).

To illustrate the possible gain of including (very) short records of data from the target catchment, we present four scatter plots that compare the predicted values produced by the areal model to the actual observations of runoff (Figure 11). For the annual predictions in Figure 11a, the predictions for PG are considerably more concentrated around the straight line that indicates a perfect fit, than the predictions for UG. There are similar results for June, whereas the difference between the ungauged and partially gauged case is not that prominent for April (see Figure 11b) and January. Furthermore, the April scatter plots demonstrate that (very) short records don't lead to a poorer predictive performance, even if April is a month driven by more

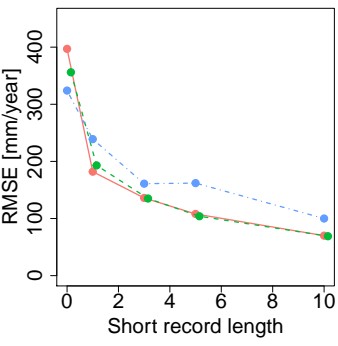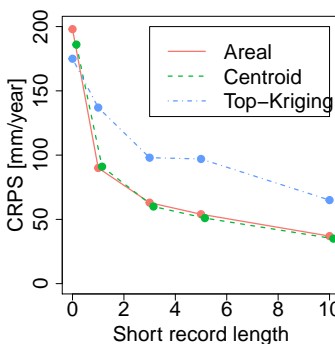

**Figure 13.** RMSE and CRPS as a function of record length (0, 1, 3, 5 and 10) for predictions of mean annual runoff for 1981-2010 for 83 fully gauged catchments in southern Norway.

unstable hydrological patterns. The predictions are simply not substantially affected by the new data points that are included in the likelihood, as we can see in Figure 11b. In our model, the risk of including very short records is low because climatic effects $c(\boldsymbol{u})$ are adjusted relative to year specific effects $x_j(\boldsymbol{u})$ by statistical inference. This way short records can safely be included in the modeling regardless of the underlying weather patterns and the degree of hydrological spatial stability.

In Figure 7 we saw that all three interpolation methods were able to reproduce the true spatial pattern of annual runoff when filling in missing annual values for ungauged catchments (UG). However, all three methods produced high RMSE values for some of the catchments. These were typically small catchments located on the western coast of Norway. Figure 12 shows the impact of including a short record of length one for these catchments. It compares the annual predictions from the ungauged case (UG) to the annual predictions from the partially gauged case (PG) for the areal model. We see a large reduction in the

RMSE for many of the catchments, and a (realistic) reduction of the posterior standard deviation. We also see that a few of the catchments obtain a decrease in predictive performance when short records are included, but the overall tendency is clear: The gain of including short records for annual predictions in Norway is high, and the suggested framework is able to exploit this property.

## 6.2    Predictions of mean annual runoff (1981-2010)

So far, we have presented an evaluation of the framework's ability to fill in missing annual observations of runoff for a 10 year period (1996-2005). We now present the evaluation of the framework's ability to predict mean annual runoff for a 30 year period as a whole (1981-2010), as described in Section 5.2.

Figure 13 shows the RMSE and CRPS for the predictions of mean annual runoff for Top-Kriging, the areal and the centroid model as a function of record length (0, 1, 3, 5 and 10). The record length is the number of annual runoff observations available from the target catchment in the cross-validation. We find that Top-Kriging again performs best for the ungauged case (short

record length 0), while the centroid model performs slightly better than the areal model for ungauged target catchments.

Furthermore, the RMSE and CRPS decrease with increasing record length for all three methods. However, Figure 13 shows that our areal and centroid models outperform Top-Kriging for record lengths larger than 0: The overall difference between our framework and Top-Kriging is around 30-60 mm/year in terms of RMSE, which is a considerable difference when the RMSE values are around 100-200 mm/year.

Furthermore, we notice the large increase in predictive performance when including a (very) short record of length one (PG1 in Figure 13). The reduction in RMSE and CRPS is 45-50 % from the UG to the PG1 case for the areal and centroid model. These results are thus comparable to the results we obtained for the experiments related to infill of missing annual values (Section 6.1).

To be able to compare our findings with other studies, we also included plots of the the absolute normalized error (ANE) and the squared correlation coefficient ($r^2$) for the experiments. These can be found in Figure 14 and 15, and are referred to in the discussion (Section 7.2). Also according to these scale independent evaluation criteria the overall results are that for ungauged catchments Top-Kriging performs best, while when there are short records available, our framework performs better.

## 7   Discussion

In this article we have presented a geostatistical framework particularly suitable for hydrological datasets that include short records of data. Here, we highlight four points for discussion: 1) the difference in performance across methods and study areas, 2) comparing the findings with other studies, 3) shortcomings of the suggested framework and 4) suggested areas of use.

### 7.1   Difference in performance across methods and study areas

In our work, we evaluated two versions of our suggested framework by predicting annual runoff and mean annual runoff for Norway. The results showed that our areal referenced method and our point (or centroid) referenced method gave very similar results in terms of posterior mean (see e.g. Figure 7 and Figure 13). We did not find a trend describing when one of the methods performed better than the others. In prior to the analysis, we would expect the areal model to perform better than the centroid model for ungauged, nested catchments since the areal model takes the water balance and the nested structure of catchments into account. However, these properties did not have a notable impact on the predicted posterior mean runoff for this particular dataset. This is not an extraordinary result as similar results have been obtained by other studies that have compared Top-Kriging (areal referenced approach) to ordinary Kriging (point referenced approach): The point referenced approaches often perform similarly as the areal referenced approaches (Farmer, 2016; Skøien et al., 2014).

A possible explanation for the similar performance of the centroid and areal model in this study, is that the proportion of nested catchments in our datasets was relatively low: Only 30 % of the catchments in Figure 1 were nested, while the percentage of nested catchments was 53 % among the fully gauged catchments in Figure 5. Furthermore, most of the nested catchments only have one overlapping catchment. The water balance constraints of the areal model might be more important for datasets where there is a higher percentage of nested catchments in an area with high spatial variability. One example is shown in Roksvåg et al. (2020).

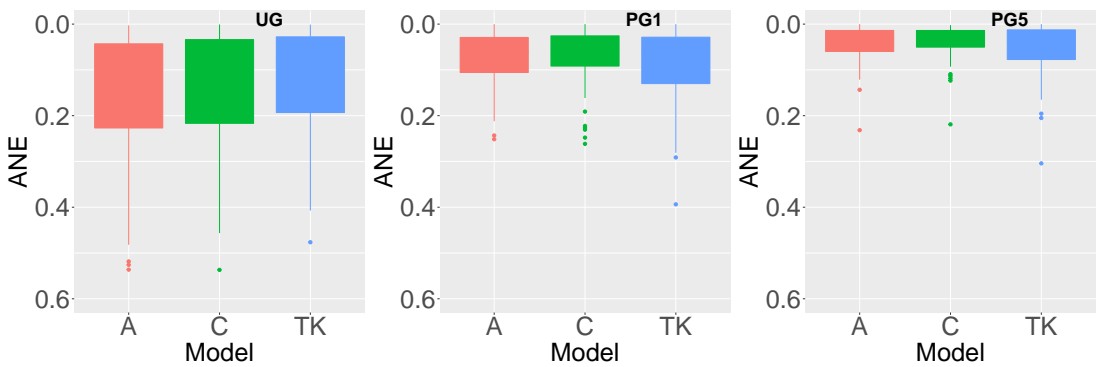

**Figure 14.** Absolute normalized error (ANE) for the areal model (A), centroid model (C) and Top-Kriging (TK) for predictions of mean annual runoff in ungauged catchments (UG, left) and in partially gauged catchments with short records of length one (PG1, middle) and length five (PG5 right).

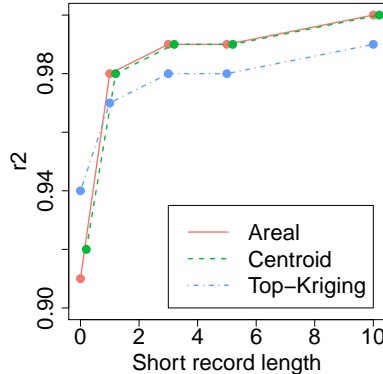

**Figure 15.** The squared correlation coefficient ($r^2$) for predictions of mean annual runoff for catchments in Norway with record length 0, 1, 3, 5 and 10.

It is also possible that the water balance constraints of the areal model have some drawbacks. One example is if there is poor data quality for a subcatchment in the dataset. Then we impose an inaccurate, but relatively strict constraint on the runoff in this catchment's drainage area. This will have an impact on the predictions for all overlapping catchments, and how the predicted runoff is distributed here. In this sense, the areal model is less flexible than the centroid model and requires better data quality.

5     The water balance constraints of the areal model also makes it computationally more expensive than the two other models. Top-Kriging used around 1 minute for the interpolation of mean annual runoff presented in Section 6.2 for one cross-validation fold. The centroid model used around 30-40 minutes for the interpolation, but provided results for both mean annual runoff and runoff for 30 individual years at the same time. Its run time is thus similar to the run time of Top-Kriging per year. The areal model on the other hand, used around 6-7 hours on the same computational server. Hence, from a practical point of view, the

10    centroid model might be the most convenient version of our suggested framework for many applications. However, note that

if the posterior uncertainty is important, the areal model gives a more realistic representation of uncertainty than the centroid model (see Figure 7). The centroid model also treats small and large catchments equally, which can be problematic for some applications and study areas.

When considering predictions for ungauged catchments, the results showed that Top-Kriging provided better results than our two suggested models. Figure 7 showed that the three methods failed and succeeded for many of the same catchments, but that our models failed slightly more than Top-Kriging on average. We also see an indication that our models fail more than Top-Kriging for ungauged catchments that are located further away from other catchments. See for example the catchment that is located south-east in Figure 7. For ungauged catchments located far away from other catchments (relative to the spatial range), the predicted value will go towards the intercept $\beta_c$ for our two Bayesian models. For Top-Kriging, the predicted value will always be a weighted sum of the observations from the neighboring catchments. This can explain the difference in performance here. Apart from this, we don't find a pattern for which catchments Top-Kriging performs better (mean elevation, location and the magnitude of the observed value were investigated).

While Top-Kriging performed best for ungauged catchments, our framework outperformed Top-Kriging when there were some available data from the target catchment. This was the case both when predicting mean annual runoff, and runoff for individual years. The results showed that the potential gain of including (very) short records in the modeling in Norway was large. An explanation is that the annual runoff in Norway is mainly controlled by orographic precipitation. Since the orographic precipitation is driven by topography and westerly winds are dominating, the precipitation patterns are repeated each year and we obtain hydrological spatial stability with $\sigma_c \gg \sigma_x$. The mountains in Norway also lead to rapid weather changes in space, here expressed through a low climatic spatial range $\rho_c$. Consequently, the information gain from neighboring catchments is often low for ungauged catchments, and information from the target catchment can be very valuable. It is also convenient that Norway has a humid climate where only around 10-20 % of the annual precipitation evaporates.

The evaluation study based on annual time series of monthly runoff gave us an indication of how the framework can be expected to behave for other climates and countries: For areas where the annual runoff is driven by unstable weather patterns and hydrological processes, short records can not be expected to contribute to as large improvements in the predictions as for the Norwegian annual data (see the predictions for April in Figure 11b). This might be the case for countries and areas where most of the runoff can be explained by convective precipitation, where the aridity index is large or for variables for which storage effects are significant. However, the monthly predictions for January and April also illustrated that we safely can include (very) short records in the model, even if year specific effects explain most of the spatial variability of runoff. By this we have demonstrated that our models represent a framework for safe use of short records regardless of record length and climate, and with the benefit that we don't need to consider the choice of donor catchment as in other comparable methods.

Norway is a country with a moderate gauging density. The framework has not been tested for a more dense gauging density. We suppose that there is less to gain from including short records if the gauging density is large relative to the spatial range: Here the information obtained from neighboring catchments could be sufficient. However, a high density of gauged catchments and a close distance to neighboring catchments does not always guarantee good predictability at an ungauged catchment (Patil and Stieglitz, 2011). It is for example often difficult to predict runoff in ungauged catchments that are very small and/or located

close to weather divides. We believe that for such catchments, our method for including short records can be useful regardless of gauging density (as long as the study area is characterized by repeated runoff patterns over time).

## 7.2   Comparing the findings of this study with other studies

There exist several other studies of mean annual runoff in the literature, and some of them are compared in terms of the absolute normalized error (ANE) in the chapter about annual runoff in Blöschl et al. (2013). According to Figure 5.27 in Blöschl et al. (2013), an ANE between 0.05 and 0.5 is a typical result for regions like Norway where the potential evapotranspiration is less than 40 % of the mean annual precipitation. Figure 14 showed that the median ANE obtained for our suggested models is around 0.12 for ungauged catchments, i.e. in the lower range of ANE values in Blöschl et al. (2013). When a short record og length one or five was available (PG1 and PG5), the median ANE was as low as 0.05 and 0.03 for our methods.

In Figure 5.30 in Blöschl et al. (2013) there is also a subplot showing the ANE for predictions of mean annual runoff for ungauged catchments in Austria. Here, geostatistical models (Top-Kriging) and process-based models (conceptual hydrological models) provided the best predictions according to the ANE, with a median ANE around 0.1. The results we obtain in Figure 14 for the ungauged catchments are thus comparable to the results from Austria. This is reasonable as the Austrian climate is humid, like the Norwegian, and the western part of the country is dominated by mountains (the Alps) and has similar climate characteristics as Norway.

Furthermore, Blöschl et al. (2013) reports an $r^2$ (squared correlation coefficient) between 0.60 and 0.99 for studies done by cross-validation of around 250 catchments, or for studies using models based on spatial proximity like our suggested framework (Figure 5.25 and Figure 5.26 in Blöschl et al. (2013)). The $r^2$ for our two models was shown in Figure 15, and we see that it lies between 0.91-0.99. This is in the higher range of values obtained by comparable studies.

## 7.3   Shortcomings

In this article, we proposed two models for runoff that are Gaussian. However, runoff is truncated at zero and typically not Gaussian distributed which we also can see from the histograms in Figure 1b and Figure 3. The consequence of the Gaussian assumptions is that there is nothing in the models that prevents them from predicting negative runoff. Negative values appear for both the areal and the centroid model due to the uncertainty given by $\sigma_y$, but this is also a problem for the Top-Kriging technique. Another source for negative values is that the climatic part of the model ($c(\boldsymbol{u}) + \beta_c$) can be negative in some areas. This is a fully valid result because the other model components could still ensure positive predictions for most catchments and years. However, it can become a problem if we are unlucky and the year specific GRF doesn't make up for the negative climatic GRF for one specific year. To avoid negative values, it is possible to log transform the data before performing an analysis. However, this is only valid for the centroid model, as the log transform is not compatible with the linear aggregation performed by the areal model (Equation (11)).

In the areal model, negative values also appear as a consequence of requiring preservation of water balance. If there are inconsistent or poor data over nested catchments, negative runoff in parts of a catchment can be the only option to fulfill the water balance requirements. To avoid negative runoff it is important that the discretization of the study area is fine enough

to capture rapid changes in runoff over nested catchments. Catchments that are significantly influenced by human activities should also be removed from the analysis as these can influence both the water balance and the significance of the climatic field $c(\boldsymbol{u})$ relative to the annual field $x_j(\boldsymbol{u})$.

In our study, some negative values were produced for the monthly predictions as we can see in Figure 11b. However, this is not common and happened for only 1.2 % of the predictions of missing monthly data, and for a few data points for the missing annual data for the areal and centroid model. For predictions of *mean* annual runoff, negative values almost never appear as such effects typically are averaged out. Note that unphysical results also appear for Top-Kriging and other interpolation methods, either in terms of violating the water balance or in terms of negative values. These model weaknesses should be remarked such that the modeler is able to choose what is most important in a real modeling setting. In this case it is a choice between 1) avoiding negative values by log transforming the data before using Top-Kriging or the centroid model or 2) to impose water balance constraints through the areal model.

## 7.4  Suggested areas of use

Finally, we want to highlight what we think are the main areas of use for our suggested framework. First, our results showed that our main benefit compared to Top-Kriging was connected to exploiting short records from the *target* catchment. For this reason, we think that our method is suitable as a pre-processing method for making inference about the (mean) annual runoff in partially gauged catchments before doing a further analysis with other statistical tools or process-based models. One possible approach for runoff estimation could for example be a two step procedure where we (i) use the centroid or areal model as a record augmentation technique to predict runoff for the partially gauged catchments in the dataset, and (ii) use Top-Kriging to predict runoff in *ungauged* catchments. Here, the results from step (i) can be used as observed values in Top-Kriging together with data from fully gauged catchments. Differences in observation uncertainty between fully gauged- and partially gauged catchments should here be taken into account.

Secondly, we see that the parameter values of the suggested model provides interesting information about the study area. More specifically, if the marginal variance of the climatic GRF $\sigma_c$ dominates over the marginal variance of the year specific GRF $\sigma_x$, it suggests that the spatial variability is stable over time, and that short records of runoff can have a large impact on the model, particularly if also $\rho_c < \rho_x$. This information can be used by decision-makers to e.g. motivate the installation of a new (possibly temporary) gauging station as this might improve the long-term estimates only a year after installation for this catchment. Likewise can the model and its parameters be used to assess whether a gauging station is redundant and can be shut down. However, to exactly quantify the importance of a gauging station, all model variances ($\sigma_x^2$, $\sigma_c^2$, $\sigma_\beta^2$, $\sigma_y^2$) and ranges ($\rho_x$, $\rho_c$) must be taken into account, as well as the distances between the donor catchments and the target catchment. Computing this gain is outside the scope of this article, but an interesting topic for further research that is related to the field of decision theory and the *value of information* (Eidsvik et al., 2015).

# 8 Conclusions

We have presented a geostatistical framework for estimating runoff by modeling several years of runoff data simultaneously by using one (climatic) spatial field that is common for all years under study, and one (annual) spatial field that is year specific. By this, we obtain a framework that is particularly suitable for runoff interpolation when the available data originate from a mixture of gauged and partially gauged catchments, and that can be used to estimate runoff at ungauged and partially gauged locations. We evaluated the framework by 1) its ability to fill in missing values of annual runoff and 2) its ability to predict mean annual runoff for ungauged and partially gauged catchments. The case study from Norway showed that the suggested framework performs better than Top-Kriging for catchments that have short records of data, both for predictions of mean annual runoff and when filling in missing annual values. For totally ungauged catchments, Top-Kriging performed best. We also 3) demonstrated the potential value of including short records in the modeling and found that the value of (very) short records was high in Norway: An average reduction of 50 % in the RMSE was reported when a short record of length one was available from the target catchment, compared to when no annual observations were available. The reason for the large reduction is that the annual runoff in Norway is mainly driven by hydrological processes that are repeated each year. For such areas, our methodology has its main benefits, and we can use it as a tool for motivating the installation of new gauging stations: The new gauging stations might improve the long-term estimates at the target catchments only a year after installation. Furthermore, the results also show that the framework represents safe use of short records down to record lengths of one year, regardless of the underlying climatic conditions in the area of interest.

*Author contributions.* Thea Roksvåg: Main author, main responsible for writing and wrote the majority of the paper. Came up with initial ideas for experimental design. Did the implementation, carried out the analysis and made figures.

Ingelin Steinsland: Contributed to discussion throughout the work, around ideas, analysis and discussion. Suggested ideas for experimental set-up and commented on the manuscript structure and content. Contributed to the writing of Section 6.

Kolbjørn Engeland: Provided the hydrological data. Contributed to discussion, particularly around the hydrological context and questions related to the data. Contributed to the writing of Section 1 and Section 2, and commented on the structure and content of the rest of the paper.

*Competing interests.* No competing interests are present.

*Code and data availability.* Example code for fitting the centroid model with example data is available on github.com/tjroksva/runoffinterpolation (doi: 10.5281/zenodo.3630348). The remaining data are available upon request.

*Acknowledgements.* The project is funded by The Research Council of Norway, grant number: 250362. We would also like to thank Dr. Gregor Laaha, Dr. Jon Olav Skøien, Mr. Joris Beemster and one anonymous referee for in-depth reviews and valuable comments. The review process improved the quality of the paper.

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
