# Peer review of "Estimation of annual runoff by exploiting long-term spatial patterns and short records within a geostatistical framework"

_Hydrology and Earth System Sciences, 2019_

## Referee Comment (RC1) · Gregor Laaha (Referee) · 11 Oct 2019

**General comments:**

This paper presents a method for estimating low flow indices based on short (one year) records. The novel method uses a Bayesian approach that treats annual flow index series of multiple gauges as Gaussian random field (GRF), that is linearly decomposed into one GRF representing the long-term average pattern, and one GRF representing the annual residual (deviation from the long-term average) pattern. Two approaches to

localize catchments in a geostatistical framework are presented, one called areal model and one called centroid model. The models are cross-validated and compared with the Top-kriging method for a larger Norwegian dataset, with some of the catchments nested, but more of them not nested. The evaluation is performed for the single-year prediction case, equivalent to in-filling single-year gaps in annual index series. The paper concludes that the proposed method is well suited for exploiting the information stored in short records of runoff data, which is seen a main benefit compared to Top-Kriging.

The paper addresses the problem how to perform optimal predictions of streamflow indices combining information of long and short records in the gauging network, which is an important science question within the scope of HESS. The objective is presenting and evaluating a novel estimation method. The paper is generally well written and easy to follow, but has some potential shortcomings that need to be amended before the paper can be considered for publication.

1. Scope of the paper

There is some inconsistency concerning the actual scope of the novel estimation method and how it is evaluated. The title suggest the method should be able to predict flow indices exploiting short and long records, what should include two cases: (A) estimating long-term average indices and interpolating annual values, and (B) models to fill in missing years in annual flow index series. However, in the paper only case B is evaluated. Filling in gaps in annual flow index series is indeed an important question, but long-term average indices may be more relevant, as they are basic requirements in a number of water management tasks. I would therefore see a greater value in assessing both cases, rather than restricting the approach to a method to fill-in missing values. For evaluating within the larger scope, the Top-kriging approach should be adapted by introducing observation weights representing the length of the record as proposed by Skoian (2006).

**2. Geostatistical methods**

The methods are generally well described and easy to follow for the reader. This is a general strength of this paper. The linear decomposition of space-time patterns and runoff into a long-term average and an annual residual pattern is a promising idea and the proposed Bayesian approach provides an elegant solution to perform joint estimation of both components. Most methods appear sound, but I have concerns about the actual value of the areal model. It is claimed to "ensure that the water-balance is close to preserved for any point in the landscape" and is presented as similar or equivalent to the Top-kriging approach with this respect. However, defining average runoff as the average point runoff in a catchment is not sufficient to ensure water balance in any sub-catchment. This requires that the "right" point runoff patterns are summed up and averaged, which cannot be observed and need to be estimated. But I cannot find in the methods how the disaggregation of point-runoff, underlying the hierarchical geostatistical model, is actually performed. In Geostatistics, this relates to the change-of-support problem which is resolved by using regularized variograms for various area, which also constitutes the core of the Top-kriging approach. No such approach is mentioned to be underlying the areal method.

A second concern is about the proposed extension to "monthly runoff". I think this term is misleading, as monthly runoff this is usually understood as a series of monthly values. In this paper, the focus is still on an annual runoff index series, but with a different index than the annual mean. One could rephrase this point, from an extension to monthly runoff, to an evaluation of the novel method for individual seasons. As processes change, the correlation length changes as well, and this may explain the different performance of methods.

**3. Evaluation method**

The evaluation so far is only focussing on the global, "overall" performance. It would be interesting now to add more specific assessments, that give insight how the methods

perform in which situation, and why.

Firstly, the underlying approach is a GRF decomposition by a linear model (Eq. 4). It would be interesting to see the importance (magnitude) of each effect. This will be informative about whether the yearly deviation from the annual pattern is rather constant, or has a spatial structure. This can be summarized in a table and in an additional plot of maps showing the spatial variability of the annual residual (range of xj(u)) as compared to the average spatial pattern c(u).

Secondly, it would be interesting to see how the proposed model performs in different estimation settings along the stream network, that define the river network estimation problem, such as small headwater catchments, interpolation between gauges, and catchments which are not nested (for example see my own attempts in (Laaha et al., 2013, 2014). This will enable the authors to show how well the areal model is able to incorporate the water-balance constraint in a useful way as stated in the introduction, and how far it is equivalent to Top-kriging in this respect. And more general, this would give an evidence which models perform well in which situation. The demonstration can be based on summary statistics stratified for the three estimation settings, and on regional examples (region with highest nestedness, and a region with very low nestedness).

Section 5.5 which gives a demonstration of an annual runoff map for southern Norway (also introducing a simpler centroid model) seems to deviate from the direct scope of the paper, appears rather uninformative and can be deleted. It would be more informative to see the components of the full model instead (see my previous comment).

4. Discussion

The Discussion section is merely a repetition of the results and findings stated before. This redundancy needs to be avoided. This can be partly obtained by a clearer separation of results and discussion section, where some parts of the results section may be shifted to the discussion (e.g. comparison with other studies, Section 5.4)

Additionally, the section should address how far the findings depend on the particular Norwegian setting and how far they can be generalized. One particularity stated in the introduction is that Norway has a very specific meteorological situation, with a pronounced east-west precipitation pattern related to orographic enhancement in front of the mountain range, that is common and prominent for each year of the data set. This suggests that taking an average pattern and shifting it, either by a constant or by a spatially corrected constant as used in this paper, will have much potential. There will likely be a lower value in other situations, when precipitation processes dominate that occur on a smaller space-time scale, such as convective events. Another particularity is the rather low gaging density, giving rise to a low number of sub-catchments along the river network. How far does this effect the performances of models should be discussed.

5. Conclusions

The conclusions (also in the Abstract) remain a bit general, and should be sharpened around the actual performance of the proposed method.

**Specific comments:**

Study area: "This leaves 195 catchments for testing with areas ranging from 7.5 km$^2$ to 18934 km$^2$." (p4 line 16). How many of them are nested?

Section 3.1.4: Please make clear whether such regression methods have been used for estimating annual discharge.

Section 3.3.1: monthly rainfall is not the scope of this paper, why not using annual runoff as an example?

Section 3.2.1: "Likewise is c(u)a spatial effect that models the long-term spatial average of runoff, or the spatial variability caused by climatic conditions in Norway..." (p10, line 30) - I think this interpretation is not sound, it is the combined effect of climate and catchment characteristics that lead to spatial variability of runoff. (This interpretation

occurs several times throughout the MS). "... while xj(u) is a year specific spatial effect that models the spatial variability due to annual discrepancy from the climate." (p10, line31f): This could also be formulated in a clearer, more meaningful way.

P11, line 19: Centroid model: "This alternative does not require preservation of water-balance and can be used for any environmental variable". Think the model "does not allow" for preservation of the water balance and is therefore not well suited for runoff and runoff-related variables, but can be applied for other environmental variables.

P14, line 16: "spatial variability" . . . is it rather space-time variability?

P14, line 30ff: Second property – preserving the water balance: Here the authors state what the model should be able to do. They need to show it is capable to do it. In contrast the example uses a constant point-runoff within the sub-catchments, which is not reasonable.

Several times: Hydrological stability is rather an abstract term that can be interpreted in different ways. Consider using low inter-annual variability instead.

Gregor Laaha, 11 October 2019

**References**

Laaha, G., Skøien, J. O., Nobilis, F. and Blöschl, G.: Spatial Prediction of Stream Temperatures Using Top-Kriging with an External Drift, Environ. Model. Assess., 18(6), 671–683, doi:10.1007/s10666-013-9373-3, 2013. Laaha, G., Skøien, J. O. and Blöschl, G.: Spatial prediction on river networks: comparison of top-kriging with regional regression, Hydrol. Process., 28(2), 315–324, doi:10.1002/hyp.9578, 2014.

---

## Author Comment (AC1) · 25 Oct 2019

**Response to the comments on the manuscript (HESSD-2019-415) "A geostatistical framework for estimating flow indices by exploiting short records and long-term spatial averages - Application to annual and monthly runoff"**

T. Roksvåg, I.Steinsland and K.Engeland

October 25, 2019

This is the author's response to the comments of referee Dr. Gregor Laaha on the manuscript (HESSD-2019-415) *"A geostatistical framework for estimating flow indices by exploiting short records and long-term spatial averages - Application to annual and monthly runoff"*. We first want to thank Dr. Gregor Laaha for his constructive suggestions and insightful comments. In this response we go through his comments on "1.Scope of the paper", "2.Geostatistical methods", "3.Evaluation method", "4.Discussion" and "5.Conclusions and Specific comments".

**1    Scope of the paper**

As the referee correctly state, we focus on filling gaps in annual flow index series in the article and don't provide an evaluation of the method's performance on predicting long-term average indices. Adding the latter to the final version is a good idea, and could replace Section 5.5. One suggestion is that we include an evaluation of the method's ability to estimate the mean annual runoff between 1996-2005 (for the 10 year period as a whole). For Top-Kriging, the approach has to be implemented with observation weights corresponding to the observation length as the referee suggests. This option will keep the presentation tidy and the results comparable to the other experiments that are done.

Another possible option is to design an entirely new experiment where we use the data in Figure 15a (all available data from 1981-2010), and do the evaluation for only some selected catchments and settings.

**2    Geostatistical methods**

About disaggregation, change of support and water-balance for the areal model:

- In our methodology the latent field (of point runoff) and the parameters (of the covariance model) are estimated simultaneously. A variogram is not explicitly estimated as in the Top-Kriging approach. Hence, we use a fully Bayesian approach as opposed to an empirical Bayesian or a frequentist approach. See Lindgren et al. (2011) for details about the inference procedure.

- We link the observed runoff (that represents catchment areas) to point runoff (averaged over the catchment areas) through the likelihood and Equation (11) for the areal model. In Equation (11), runoff is modeled as the average point runoff over a discretization of a catchment area $\mathcal{A}$. Furthermore, the likelihood and Equation (11) are used to put constraints over the gauged catchments. This means that the posterior mean runoff over the grid nodes that represent the area of a gauged catchment is constrained to the actual observed value with some (small) uncertainty through the likelihood. With one constraint for each gauged catchment, annual runoff is estimated (posterior mean) for each grid node in the discretization for the whole study area. See the illustration below for an example of how the posterior mean results can look like at point level. The posterior mean point runoff $q_j(\boldsymbol{s})$ is aggregated according to Equation (11) to produce figures like Figure 7, i.e catchment (areal) predictions. A figure

similar to this is behind all of the predictions presented in our article. See Moraga et al. (2017) for more technical details about the areal formulation.

- We should also mention that we have a quite strict prior on the uncertainty of the areal observations (page 12): This is to keep the esimated value in the gauged catchments as close as possible to the actual observed runoff (i.e we require the estimated runoff of a catchment to follow the observed runoff closely), and to avoid too much smoothing.

- Since two catchments that overlap share grid nodes, a catchment can not have a smaller posterior mean annual runoff runoff in m$^3$ /year than an overlapping sub-catchment for the areal model. (The posterior mean point runoff $q_j(\boldsymbol{u})$ from the illustration below is aggregated to catchment runoff for all nested catchments).

[Figure]

***Illustration:*** *From posterior mean at point level $q_j(\boldsymbol{u})$ for a selected year j (left) to posterior mean at catchment level $Q_j(\mathcal{A})$ (right). If we use the areal model, we go from the left plot to the right plot by aggregating point runoff according to Equation (11) for all catchments. If we use the centroid model, we simply take the posterior mean at the catchment centroid in the left plot to produce the catchment runoff in the right plot.*

The referee writes that *"most methods appear sound, but I have concerns about the actual value of the areal model"*. The areal model has two main benefits compared to the centroid model: 1) It gives a better representation of the posterior uncertainty and 2) its ability to fulfill the water-balance and distribute the annual runoff correctly over sub-catchments. In our article, only benefit 1 is demonstrated through a real case example (Figure 7 showing the posterior standard deviation of A, and Table 1 showing the coverage of A compared to C). We did not find a clear example in our dataset where property 2 represented a large benefit over the centroid model, which is also the answer to the referee's later comment: *"It would be interesting to see how the proposed model performs in different estimation settings [...] This will enable the authors to show how well the areal model is able to incorporate the water-balance constraint in a useful way as stated in the introduction, and how far it is equivalent to Top-kriging in this respect"*.

Benefit 2 of the areal model is probably easier to demonstrate through a simple case example. One example from Voss in Norway is already available on page 15-16 in Roksvåg et al. (2019), accessible at `https://arxiv.org/pdf/1904.02519`. In Roksvåg et al. (2019) we use the same model as in this article, except that also point referenced precipitation data are used in the analysis. Here, the areal representation of nested catchments allowed us to correctly predict larger values in Catchment 3 than any of the observed values (P+A in Figure 5 in Roksvåg et al. (2019)), which was our statement in Section 3.2.6 (page 14-15) in the article under discussion. The centroid model would not be able to do this. We can search for a similar simple case example for the final manuscript. However, since we did not see any clear trends in our original dataset, we think that the presentation will be clearest if we stick to the overall global performance and use the Norwegian dataset as it is.

Further, the referee has some comments about the monthly runoff extension and choice of words. This can be rephrased as the referee proposes. The main reason for including the analysis of the monthly data is to investigate and explain the method's performance for a different "climate" and parameter set ($\sigma_c$,$\sigma_x$, $\rho_c$, $\rho_x$).

**3   Evaluation method**

As already mentioned, we did not find a clear trend describing when the areal method performed differently from the centroid method. We did not find a trend for when Top-Kriging performed better/worse either. This is why we focus on the *"global overall, performance"* of the method rather than *"more specific assessments"*. See for example the RMSE plot for UG in Figure 7: The three methods typically fail for the same catchments, but on average the areal and centroid method fail a bit more than Top-Kriging for the ungauged case (UG) in terms of RMSE and CRPS. As stated in the discussion, we investigated if one of the methods was better for e.g high elevated catchments, location (nested catchments) or for wetter/dryer catchments, but we did not find a correspondence.

Furthermore, we did not find a general rule for when a short record has a large impact on the final results. See Figure 13 of the RMSE: Many of the catchments already have a low RMSE when treating them as ungauged (UG). The catchments that have a large RMSE for UG, typically gets a considerably lower RMSE when adding one observation of annual runoff (PG). However, from this figure it is difficult to see what kind of catchments that generally benefit from adding a short record. In a preliminary simulation study, we saw that short records have a larger impact on locations that are not nested and far away from the other catchments in terms of the spatial range. This is as expected. For catchments that are nested and/or have many surrounding observations, the increased value of adding one observation was low for simulated data. However, the nature is more complicated and never fits perfectly to any statistical model. In our case study, we have several examples of catchments where we get a large increase in predictive performance when adding a short record, even if the target catchment is nested or has several neighboring catchments (RMSE in Figure 13).

As no prominent patterns were found explaining when one of the models (A, C or TK) or settings (UG vs. PG) were beneficial for this specific dataset, we think that the presentation of the method is clearest when we restrict the main results to the overall global performance.

The referee also writes *"It would be interesting to see the importance (magnitude) of each effect. This will be informative about whether the yearly deviation from the annual pattern is rather constant, or has a spatial structure"* and suggests that we make a table. The authors are not quite sure about what is meant here, as the spatial range of $x_j(\boldsymbol{u})$ is constant for each $j = 1, ..r$ within a cross-validation fold. In Equation (4), the $x_j(\boldsymbol{u})$'s for $j = 1, ..r$ are modeled as independent realizations of the same underlying model, i.e each year has the same underlying range $\rho_x$. This range is already reported in Table 4 for the different time-scales. Here, we also report the magnitudes of $\sigma_c$ and $\sigma_x$ which gives a lot of information about *"the importance (magnitude) of each effect"*. However, we can easily add *"additional plot of maps showing the spatial variability of the annual residual xj(u) as compared to the average spatial pattern c(u)"*, as the referee suggests, i.e a figure that shows the spatial fields $x_j(\boldsymbol{u})$ and $c(\boldsymbol{u})$ with the full spatial point pattern $x_j(\boldsymbol{u}) + c(\boldsymbol{u})$ (as in the above illustration) for one or a few selected years $j$. Even if the range $\rho_x$ of $x_j(\boldsymbol{u})$ is constant for $j = 1, , .r$ within a fold, the picture produced for each $j$ can be very different from one year to another. A figure like this gives a good visualization of the magnitude of each spatial field, and the authors agree that this will be informative for the reader.

**4   Discussion**

The referee has several good suggestions here for improving the discussion part. The referee e.g writes that *"The section should address how far the findings depend on the particular Norwegian setting and how far they can be generalized"*. This can indeed be discussed more. One of the reasons for including monthly data, was to investigate if the results can be generalized to other runoff regimes. We saw that for June, short records have a great value, while for January and April they have a smaller value because the spatial pattern of runoff is more year dependent due to meteorological processes like snow melting/storage that are less stable from one year to another. This will also be the case for other areas and countries *"where precipitation processes dominate that occur on a smaller space-time scale, such as convective events"*. However, using the suggested models will not affect the results negatively if this is the case, because the model adjusts $\sigma_x$ relatively to $\sigma_c$: We showed that the predictions for January and April for ungauged catchments (UG, Table 1) are approximately equally as good as the predictions for partially gauged catchments (PG, Table 2).

In this context we can also add that the main author has done some quick experiments with a dataset consisting of 10 years of annual data from around 550 catchments in Austria. In a preliminary study we found that $\sigma_c > \sigma_x$ and $\rho_c < \rho_x$, i.e similar to the Norwegian annual data. This suggests that the framework could be useful in Austria too. The density of stream gauges is indeed larger in Austria than for the Norwegian dataset, but we saw in Figure 13 that even in areas with a higher gauging density, a short record can be valuable. Particularly in areas with rapid changes of runoff, and/or a prominent weather divide (like in e.g the Alps).

**5    Conclusion and specific comments**

The referee's suggestions will be taken into account in a final version of the manuscript.

**References**

F. Lindgren, H. Rue, and J. Lindström. An explicit link between Gaussian fields and Gaussian markov random fields: the stochastic partial differential equation approach. *Journal of the Royal Statistical Society: Series B (Statistical Methodology)*, 73:423–498, 2011.

P. Moraga, S. M. Cramb, K. L. Mengersen, and M. Pagano. A geostatistical model for combined analysis of point-level and area-level data using INLA and SPDE. *Spatial Statistics*, 21:27 – 41, 2017.

T. Roksvåg, I. Steinsland, and K. Engeland. A knowledge based spatial model for utilising point and nested areal observations: A case study of annual runoff predictions in the Voss area. *arXiv:1904.02519*, 2019.

---

## Short Comment (SC1) · 7 Nov 2019

**Review of manuscript:**
**"A geostatistical framework for estimating flow indices by exploiting short records and long-term spatial averages – Application to annual and monthly runoff" by Roksvåg et al. (2019a)**

Joris Beemster[1]

[1]Wageningen University and Research, Wageningen, The Netherlands

*This review was prepared as part of graduate program course work at Wageningen University, and has been produced under supervision of dr Ryan Teuling. The review has been posted because of its good quality, and likely usefulness to the authors and editor. This review was not solicited by the journal.*

**Recommendation:** Accept manuscript after minor revisions.

**Introduction**

The manuscript presents a new framework for the spatial interpolation of hydrological data, that includes a mix of fully and partially gauged catchments. It tackles the problem of combining short and long discharge records, when predicting streamflow indices. A key aspect of the novel method is that it models several years of runoff simultaneously using two Gaussian random fields (GRFs). One that is common for all years, representing the climatology and catchment characteristics, and one that is year specific, representing the deviation from the climatology. Based on this framework, the authors propose two models. One that satisfies the catchment mass balance (the areal model) and one that does not (the centroid model).

The models are tested on a dataset of 195 fully gauged Norwegian catchments and the results are compared to those produced by Top-Kriging (Skøien et al., 2006) and linear regression. The performance is evaluated by means of three tests, all based on cross-validation. In the first test, the models predict the runoff of ungauged catchments. During the second test, the ability to predict the runoff of a partially gauged catchment is evaluated. Lastly, the models predict the runoff of an ungauged catchment, of which the three nearest neighbors are partially gauged.

Top-Kriging outperforms the newly proposed method in the ungauged case. In the case of partially gauged catchments, the areal model generally performed best, indicating that the method is capable of exploiting short records, which seems to be the main advantage over existing methods. The presented method also outperforms Top-Kriging, when interpolating ungauged catchments with partially gauged neightbors, but differences are smaller.

As the presented methodology is capable of transferring information contained in short discharge records across different years, it significantly increases the value of large amounts of short records. Furthermore, it provides an easily interpretive way to indicate the relative importance of the climatic signal and catchment characteristics and the interannual variation. The novel method fits the scope of *Hydrology and Earth System*

*Sciences* well and the paper is generally understandable. The main weaknesses of the paper are the minimal explanation of mass conservation in the areal model, the missing discussion of the wider applicability of the method and the limited study area description. Therefor, I recommend to accept the manuscript after some minor revisions. My major, minor and textual suggestions are outlined below.

**General Comments**

The first main shortcoming of the manuscript is that it remains unclear how catchment discharge is disaggregated to point discharge. I assume that in catchments that do not contain any subcatchments, it is defined as the average point discharge and in catchments with nested gauged nested catchments, the nodes in the ungauged part are chosen such that the water balance is preserved. It is a simple and clever way to ensure mass conservation, but it should be explained more clearly in the methodology. If, in the results section, examples would be given of a system of nested catchments instead of only providing the global picture, the way nested catchments are incorporated into the method would become more transparent and advantages and disadvantages could be highlighted more clearly. An alternative would be to mention more clearly that the author has also applied the method to a more limited dataset, of only 5 catchments in the Voss area (Roksvåg et al., 2019b).

My second concern relates to the discussion. Although it thoroughly reflects on the strengths and weaknesses of the novel method applied to annual and monthly discharge of southern Norway, it does not address the applicability of the method to other flow indices and different geographical settings. A nice way to bridge between the two aspects would be to move the comparison with other studies (5.4) to the discussion. (More generally, I would recommend to reconsider which sections/sentences should be in the result and which in the discussion). This should be followed by a discussion whether or not the method is expected to perform particularly well for the Norwegian setting and how the spatial interpolation of annual and monthly average runoff compares to the interpolation of other flow indices.

Related to my previous comment, to give the reader a sense of how well this method would work in different geographical settings as well as explain the results, it would be helpful to further elaborate on the study area. There are over 300 large dams in Norway (Icold, 1998). The presence of large reservoirs in a catchment will undoubtedly have an impact on the performance of the method. In my opinion, it would strengthen the description of the study area, if an assessment of the degree to which the catchments are influenced by human activity was made. Furthermore, it would be valuable to mention the amount of nested catchments and degree of "nestedness". Currently, figure 1b gives an indication, but no reference to this figure is made in the study area description. Adding a couple of sentences mentioning the amount of nested catchments, as well as, a reference to figure 1b, will also improve the study area description. Lastly, it is unclear to me why records from all over Norway are used (figure 1a), but all maps in the results section only show the results for southern Norway. It seems more consistent to limit the analysis to southern Norway or to present the results for the entire country to increase transparency.

**Specific technical comments:**

- p1, line 17: Please provide more than one reference if you state that "Average annual flow is *often* used..."

- p3, lines 19-20: "A similar model has already shown promising results". Please mention how it differs.

- p9, lines 29-30: "However, it has been shown that Top-Kriging also performs well for variables that are not mass conserved, like e.g the specific 100-year flood". Please support this statement by one or more citations. Skøien et al. (2006) indeed show that Top-Kriging works well for the 100-year flood. Top-Kriging was also successfully applied to low flow data (the daily discharge that is exceeded 95% of the time) (Laaha et al., 2014) and to streamflow temperatures of Austrian rivers. (Laaha et al., 2013).

- p10, lines 30-31: "Likewise is c(u) a spatial effect that models the long-term spatial average of runoff, or the spatial variability caused by climatic conditions in Norway". The spatial variability is not only caused by climatic conditions, but also by the catchment characteristics.

- p12, lines 10-12: Petersen-Øverleir (2004) indeed shows that heteroscedasticity is a widespread problem of Norwegian gauging stations. However, he also shows that differences between gauging stations are large and that there is at least one example where the uncertainty decreases with increasing runoff. Please motivate why this value for the scaling factor was chosen.

- p12, lines 22-23: Please provide a source for the following statement: "This corresponds well to what we know about the measurement uncertainty for runoff in the study area."

- Inference (3.3): In this section several simplifications are mentioned aimed at reducing the computational complexity. Could you please comment on the effect these simplifications have on the expected outcome?

- p17, line 31: It would be interesting if the case of ungauged neighbors is also evaluated. Likely, large improvements will be seen in the **PG-N**, relative to the **UG-N** case for the new methods that are less apparent for Top-Kriging.

- Evaluation scores (4.2): The performance of the model is mainly evaluated in terms of RMSE and CRPS, two evaluation scores that are scale-dependent. In my opinion, adding a scale independent evaluation score, such as the Nash-Sutcliffe or the Kling-Gupta efficiency, would make the comparison between averaged annual and monthly runoff more straightforward. Furthermore, this would enable the evaluation in terms of the correlation, the conditional bias and the unconditional bias (Gupta et al., 2009).

- Figures 8-10: The units of the y-axis are not mentioned. Please add them. In my opinion these figures could be left out of the paper, because they are made redundant by table 1-3 and figure 7.

- p24, line 11-13: "However, recall that a short-record of length 2 from the target catchment is needed in order to use this method, while our areal model performs approximately equally well with a short-record of length 1 (and observations from other neighboring catchments)." If you would also test the areal and centroid method for partially gauged catchments with a record length of two, the comparison with linear regression would be more straightforward.

- Figures 11-12: It would be useful to add the scatter plots for the **PG-N** case, for completeness.

- p27 line 18-20: "The results from UG and PG-N in Table 1 and 3, indicate that exploiting long-term spatial averages in the interpolation scheme as in our methods, can be just as important as finding a process-based way of determining the Kriging weights which is the idea behind Top-Kriging." This sentence belongs to the discussion section.

- Mean annual runoff map for southern Norway (5.5). In my opinion, this section does not have much added value. Instead, as mentioned before, it would be interesting to highlight the results of specific catchments. Also, if you choose to keep this section, it would benefit from adding a map produced using the Top-Kriging method, for completeness.

**Minor textual suggestions**

- p1, line 17: "a predictor" instead of "an predictor"

- p9, line 30: "e.g." instead of "e.g"

- p11, lines 19-20: Consider to change "This alternative does not require preservation of the mass balance" to "This alternative does not assure preservation of the mass balance".

- p14, line 1: "is an indicator" instead of "is a indicator"

- p27 line 18: Please change "... annual scale in Table 3" to "annual scale (Table 3)"

**References**

Gupta, H. V., Kling, H., Yilmaz, K. K., and Martinez, G. F. (2009). Decomposition of the mean squared error and nse performance criteria: Implications for improving hydrological modelling. *Journal of hydrology*, 377(1-2):80–91.

Icold (1998). World register of dams.

Laaha, G., Skøien, J., and Blöschl, G. (2014). Spatial prediction on river networks: comparison of top-kriging with regional regression. *Hydrological Processes*, 28(2):315–324.

Laaha, G., Skøien, J. O., Nobilis, F., and Blöschl, G. (2013). Spatial prediction of stream temperatures using top-kriging with an external drift. *Environmental Modeling & Assessment*, 18(6):671–683.

Petersen-Øverleir, A. (2004). Accounting for heteroscedasticity in rating curve estimates. *Journal of Hydrology*, 292(1-4):173–181.

Roksvåg, T., Steinsland, I., and Engeland, K. (2019a). A geostatistical framework for estimating flow indices by exploiting short records and long-term spatial averages – application to annual and monthly runoff. *Hydrology and Earth System Sciences Discussions*, 2019:1–35.

Roksvåg, T., Steinsland, I., and Engeland, K. (2019b). A knowledge based spatial model for utilizing point and nested areal observations: A case study of annual runoff predictions in the voss area. *arXiv preprint arXiv:1904.02519*.

Skøien, J. O., Merz, R., and Blöschl, G. (2006). Top-kriging - geostatistics on stream networks. *Hydrology and Earth System Sciences*, 10(2):277–287.

---

## Author Comment (AC2) · 14 Nov 2019

**Response to the comments on the manuscript (HESSD-2019-415) "A geostatistical framework for estimating flow indices by exploiting short records and long-term spatial averages - Application to annual and monthly runoff"**

T. Roksvåg, I.Steinsland and K.Engeland

November 14, 2019

This is the authors' answer to the interactive comment posted by Joris Beemster. We are very thankful for Joris Beemster contribution. His review will for sure contribute to improving the paper.

**1    General comments**

First, Joris Beemster comments that *"it remains unclear how catchment discharge is disaggregated to point discharge"*. This comment is similar to the comment by referee Dr. Gregor Laaha. The authors conclude that this part of the methodology should be described in more detail.

Joris Beemster next comments that there should be more discussion regarding *"the applicability of the method to other flow indices and different geographical settings"*. We like his suggestions about moving the comparison with other studies (5.4) to the discussion part. We will also add some more discussion on how we expect the methodology to perform in other areas and for other indices.

*"Furthermore, it would be valuable to mention the amount of nested catchments and degree of nestedness". Currently, figure 1b gives an indication, but no reference to this figure is made in the study area description"*. Regarding this comment, we can add a reference to Figure 1b, and include a sentence about how many of the catchments that are nested and not nested. We can also add that only catchments where human activity has a minimal effect on the mean annual runoff are included in the analysis.

Joris Beemster lastly comments that *"it is unclear to me why records from all over Norway are used (figure 1a), but all maps in the results section only show the results for southern Norway. It seems more consistent to limit the analysis to southern Norway or to present the results for the entire country to increase transparency"*. The analysis and cross-validation is done for the whole country. However, we only show figures of southern Norway in the results section. This choice is made to make the presentation clearer and the figures as clean as possible. In Figure 1 we show the whole study area, and here it is a bit difficult to see the smaller catchments. We wanted to avoid this in the results section, and solved it by only plotting a "zoomed-in" version of Norway. Furthermore, each plot is included to illustrate a property of the model and these properties can be illustrated without including the whole study area. For example the point of Figure 7 is to show that the three methods all are able to reproduce the spatial pattern in Norway, and to show that they typically fail for the same catchments. As stated in the caption, *the methods give similar results for northern Norway*, i.e including northern Norway does not add any new information regarding the difference in performance of the three methods. The purpose of Figure 13 is to show that we are able to decrease the RMSE for some of the "problematic" catchments by including a short record of length 1. Again, this can be showed without including northern Norway, and is a choice made for making the illustrations clearer.

**2 Specific comments**

We will now reply to some of the specific comments by Joris Beemster. The specific comments that are not commented in this reply are mostly related to the references, choice of words and adding more explanations, and will be taken into account in a final version of the manuscript.

First, Joris Beemster writes that *"it would be interesting if the case of ungauged neighbors is also evaluated. Likely, large improvements will be seen in the PG-N, relative to the UG-N case for the new methods that are less apparent for Top-Kriging"*. This is a good suggestion. However, we think that we should prioritize making an assessment of the methods' predictive performance on mean annual runoff (for a period as a whole, not for individual years) as suggested by referee Dr. Gregor Laaha. As it is now, PG-N is included to show the difference between the three methods when having a sparser dataset with neighboring catchments that have a lot of missing data. This is informative also without adding UG-N.

Next, Joris Beemster writes that *"if you would also test the areal and centroid method for partially gauged catchments with a record length of two, the comparison with linear regression would be more straightforward"*. We chose to use a record length equal to one for our methods to emphasize that 1) our methods are able to exploit a short record of length one. Linear regression requires a record of length two or more. 2) We can provide predictions approximately equally as good or better than linear regression by using a shorter record length.

Regarding evaluation scores: We have chosen RMSE and CRPS which works well when the main interest is in comparing the predictive performance (and predictive uncertainty) between methods rather than between catchments or time scales. Furthermore, in Figure 14 we include the ANE and $r^2$ for the areal model for all settings. These two scores are scale independent and make it possible to compare the predictive performance on the annual scale to the predictive performance on the monthly scale.

Regarding Figures 8-10: These give some additional information compared to Table 1-3 because they show how large spread there is in the RMSE for each method and setting. The mean RMSE over all catchments can be equal for two methods (considering only Table 1-3), but the variability can be larger for one method compared to the other (which can be learned from Figures 8-10).

Regarding Figures 11-12: We chose to not include a scatter plot for PG-N here. The purpose of these scatter plots is to show the increase in the predictability of annual runoff when including a short record of length one, and that the impact on April/January is less apparent. The dataset used for PG-N is a lot sparser, and not directly comparable to the UG and PG case.

Similar to referee Dr. Gregor Laaha, Joris Beemster comments that the mean annual runoff map in section 5.5 does not have much added value. As stated in our previous reply, this section can be replaced with an assessment on the methods' predictive performance for mean annual runoff (for a longer period as a whole, not only for individual years as in the rest of the paper).

---

## Referee Comment (RC2) · Anonymous Referee #2 · 15 Nov 2019

The authors have presented a Bayesian geostatistical approach to filling in gaps in the record of annual streamflow. Two variants of the method are presented: One that attempts to conserve mass using areal weighting and a second that uses the more traditional centroid-referenced approach. Bayesian methods are introduced to allow for parameter uncertainty in the underlying processes. The manuscript is well-prepared, and the methods are clearly articulated. The impact of this work could be deepened by providing a more thorough exploration of the purported properties of these estimators and further quantifying the variability of performance.

[Figure]

My main concern is with the method that the authors are most excited about, the areal method. The manuscript leaves the reader with concerns about the benefits, performance and utility of this method.

On benefits of the areal method, the authors make several claims as to the superiority of the areal method (with some of these benefits extending to the centroid method, generally), but rarely is evidence provided to document these benefits. Around line 13 of page 14, the authors suggest that the areal and centroid methods are uniquely designed to take advantage of short and long-term records. This particular benefit is true of many kriging methods, as shown by various authors' suggestions of weight top-kriging and kriging estimates differently (see, e.g., Skoien, 2006; Farmer, 2016). In both reports, the authors show how building different variograms can increase the importance of longer records to build regionalize geostatistical approaches. This pervious work does not invalidate what is presented here, as this manuscript includes some unique Bayesian work, but does demonstrate that further evidence of this claim can be provided through experimentation. The second property proposed by the authors is that the areal method conserves mass. This is only demonstrated hypothetically. As a valuable claim, I think it important to document explicitly. Section 3.2.4. shows how runoff would be accumulated across the drainage area, but I am concerned that this would not conserve mass because, as one example, it does not account for routing. That is, summing all the grid cells, so to speak, on a day does not produce the outlet runoff on that same day. The runoff at the outlet cell is rarely a product of the contemporary runoff at a cell at the top of the drainage area. I'd like to see some further evidence of how this mass conservation is validated, especially in a sparse network.

On performance, the results of Tables 1, 2 and 3 do not convince the reader that the areal method is a meaningful improvement over the previous methods. Here, we consider only averaged performance across all basins. Even in this case, the top-kriging is superior in the ungauged cases, while linear regression and the centroid method are superior in the annual cases of partially gauged and partially gauged networks,

respectively. There is some benefit in the monthly cases. As a first order, I think it inappropriate to compare means and make definitive statements. As these RMSE and SRPS means are simplifications of data that is available, I strongly advise the use of significance testing to understand if the differences between these methods are meaningful. Given the variability Figures 8, 9 and 10, the differences may not be significant.

On utility, the authors acknowledge that the areal model is computationally prohibitive for any real-world application. For example, see line 11 on page 19 and section 3.3, where the author points out that the spatial discretization of the areal method means that several substantial assumptions must be made to simplify the areal method for application. While there is certainly value in presenting a hypothetical model for discussion, this leaves the reader feeling like the areal method is only a hypothesis that cannot be tested.

The biggest evidence of the weakness of the areal method and the centroid method, and the biggest undercut to the authors' claims of advance, is that, when it comes to application, even these authors do not use their proposed methods. See sections 4.3 and 5.5. where the authors present a new, untested method to reproduce annual values across Southern Norway. The reader is left interested in the hypothetical method, but surprised that it is not used.

In addition to this main concern, I will now move on to some other major concerns. Addressing these will, I hope, improve the manuscript.

I find the authors' simulation of short records somewhat concerning. At the bottom of page 1, the authors discuss the PUB initiative and its relevance to short records. First, I think it important to explicitly state that PUB is taken to apply to any ungauged point in space and time – that is, it included the completely ungauged and partially gauged cases. Line 23 claims that a few years of data could be useful for estimation. Indeed, there is a long history of such procedures, but I find it surprising that authors simulate a partially gauged site as one having on a single year of annual data (page 17, line

24). This is an extreme, and possibly unrealistic, case of partial gauging that will substantially affect the performance of the methods presented. While it is difficult to work with short records (e.g. 10 years of annual data), I would represent the ungauged case with three or more values. Indeed, on line 7 of page 18, linear regression is performed with only two data points. This is upsettingly problematic as linear regression is meaningless for two points – it's just a line connecting the points. A minimum of three points would be required for any meaningful regression. (Or, are the regression built across the entire region simultaneously resulting in a single regression for all sites? Even that is suspect.) Given that the use of one point for partial gauges and two points for regression (line 12, page 24), it would seem wise to use a consistent number of points to represent partial gauging. (An additional analysis that may be beyond the scope of this work could consider the sensitivity of these methods to partial record length.)

I suggest dropping the sections on monthly analysis. The simulation of monthly streamflow tends to imply that one is producing monthly sequences line Jan-Feb-Mar, but this work is looking at Jan-Jan-Jan (for example). This is akin to only predicting a new statistic of streamflow and is not a novel advance of the method. While it could be expanded to provide a more robust analysis, removing it might help streamline the manuscript.

I also suggest dropping the simulation of the mean annual runoff map for southern Norway. This provides no additional methodological advance and substantially undercuts this work. In tables 1, 2 and 3, the annual values are shown to be best reproduced by various methods (TK, LR and Centroid), none of which are used in this application. The narrative of the manuscript might be improved by removing this section.

Finally, I'd love to seem some additional analysis on the regional variability of performance of these methods. What seems to drive the varying levels of performance across Norway?

Some more minor comments:

Page 4, line 33: This figure does not show that runoff is lowest or highest at any single site, it only shows the relative distributions. Please correct this statement.

Page 6: This discussion focusses on whether or note the lines look parallel. This is highly subjective and should be quantified in some way. For example, if I changed the vertical axis to run from 0 to 100,000, all the lines would "look parallel". Please provide some quantification of correlation.

Page 11, line 2: Why would we expect the long-term spatial average runoff ($c(u)$) to have a zero mean?

Page 11, line 4: Why would expect a sequence of annual values to be independent? Is this true for monthly values?

Page 22: The coloring of figures like Figure 7 make it difficult to see the variability of performance. The results appear highly skewed, resulting in almost all RMSEs being brown; an alternative scale might distinguish performance better.

Throughout: The numbering of figures and tables is inconsistent. The figures and tables should be numbered according to the order of presentation in the prose.

Finally, thanks for a great read. I look forward towards revision and future discussion. Great work!

REFERENCES:

Farmer, W. H.: Ordinary kriging as a tool to estimate historical daily streamflow records, Hydrol. Earth Syst. Sci., 20, 2721–2735, https://doi.org/10.5194/hess-20-2721-2016, 2016.

J. O. Skøien, R. Merz, G. Blöschl. Top-kriging - geostatistics on stream networks. Hydrology and Earth System Sciences Discussions, European Geosciences Union, 2006, 10 (2), pp.277-287. ffhal-00304844f.

---

## Referee Comment (RC3) · Jon Olav Skøien (Referee) · 22 Nov 2019

This manuscript describes a methodology for spatial interpolation of time series, which can also take into account short records at the target location. The manuscript is well written and the methodology seems sound. I have some detailed comments below, which I think can all be answered within a minor revision.

Similarly to the other reviewers, and what the authors have written in their answers, I agree that it would be a good idea to replace the mean annual runoff map for Norway

[Figure]

with an assessment of the methods' performance for mean annual runoff.

The manuscript analyses a situation with only one observation for the partially gauged catchment. This is an extreme case, so extreme that an extra observation was introduced for the linear regression to be possible. Whereas this gave a convincing case study, I'd like to see at least some discussions around the effect of more observations also for the PG case. Also, the annual series are based on daily observations, so many observations are actually available for the regression. Could this give another result? Spatio-temporal kriging will most likely not do much better than the spatial interpolation, as it will use the model based covariance rather than the observed covariance also for the PG case, but I think it should be mentioned. It might be a good alternative for a time series with a few (maybe non-consecutive) missing observations.

P3L3 It is referred to how the model is developed for annual datasets, but might be used for indicators with other temporal supports (such as monthly). However, it is never explained why there is a difference between monthly or annual data, except for different correlation lengths etc. I guess this is related to the comments on P3L15, were it is referred to the water balance being "close to preserved . . . with some uncertainty". However, the "almost preserved" is never explained. The same description is used on P11L13. Could the method also be used for daily data? If not, why?

P25 – description of Figs 11-12. It is mentioned that UG has problems predicting large values of runoff, but I'd say it is just as difficult with small values. Additionally, the negative values should be mentioned here, not only in the conclusions. It seems there are no negative values for annual runoff for UG? I think this result is partly related to the fact that the data don't follow the assumption of being normally distributed, which should be discussed. A transformation method such as logtransform could avoid this problem, although log-transformed data on the other hand don't go so well with the linear aggregation assumption, see also Clark (1998).

The reference to Figure 3b comes before Figure 2. Figure 3a is only mentioned in the

reference to Figure 4. Figure 1b is not mentioned at all in Section 2, only in Section 3. Figure 1b is a bit difficult to understand, including the reference to 0-4 catchments. These types of figures are generally difficult to make nice, so I'm only asking the authors to test if other visualizations could work better. The lines of 4 catchments is difficult to see. Maybe it would be possible to add river network on top, to understand the catchment order?

Section 5.2. I think it could be mentioned that the areal model is substantially better than TK when $sigma\_c \gg sigma\_x$. Then later in the section, it says linear regression gives more unstable uncertainty estimates. Is unstable the correct word here?

P27L10 I do not think it is correct that observations from nearby catchments have a larger impact on the target catchment for the centroid model when $range\_c > range\_a$. It should be the opposite, large range means that stations further away will also get a weight in the interpolation. Then also the next sentence (L12-15) seems somewhat incorrect. If the values in Table 3 are divided by the similar values in Table 1, maybe the differences could easier be interpreted in light of the range?

The discussion at the end of P31 should also include some thoughts around gauging density. If the density is high, it is more likely that catchments are nested. Can the centroid model be expected to be as good as the areal model for non-nested observations? And when mentioning other environmental variables, it should maybe be stated that these are point values?

Edits

The use of commas should be checked. Several introductory clauses that need commas do not have them, whereas some commas are not needed. Examples:

P1L6 Another property, -> not needed

P4L18 In Western Norway, -> needed

P4L24 The processes explained above, creates -> not needed

P8L30 ... stationary, Matern ... -> not needed

Other edits:

P1L5 "The climatic GRF ..."- I think this could be rephrased. If I understand correct, the GRF learns the spatial pattern from a limited number of years, and can use this information to improve predictions for years without observations.

P2L5 "including data ... runoff datasets" – I think this part of the sentence can be deleted.

P2L32 Add "spatial" before "correlation"? Maybe the sentence could also be somewhat rephrased, at first reading it appears contradictory when referring to observation locations having both high correlation and high spatial variability.

P3L2 sparse datasetS

P3L10 "the climatic spatial field learns..." – I don't think a spatial field can learn, rephrase?

P4L7 "The study is carried out by utilizing" – a bit wordy, shorten?

P4L18 "The large values ...Norway are mainly ..." – could be shortened, maybe "This is mainly ..."

P4L13 "This leaves" – rephrase?

P4L14 The sentence referring to Figures 1a and 1c doesn't really fit in, try to rephrase paragraph.

P4L30-31 The sentence doesn't read well, try to rephrase.

P4L33-34 "The monthly runoff data from Norway" -> "The data from these months"? Caption Fig 1: "FREQUENCY OF annual runoff observations"? And maybe "...Figure 1a, subcatchments are plotted on top, and this ..."

P7L4 "and so on" is not a good expression in a scientific context. "In order" can be

removed.

P7L5-6 I don't think statistical analyses are only because of uncertainties?

P8L6 I'm not sure if it really clear what is meant by point prediction here.

P8L8 rephrase "things"

P9L4 Covariance or correlation?

P9L10 I think this sentence (known covariance structure with unknown parameters) is unclear.

P9L28 linear aggregation of what?

P10L7 The sentence is incomplete.

P10L15 It is not clear what is meant by "uncertainty based on . . ."

P10 Eq 4 I think the second equation needs units.

P10L30 "in Norway" – could maybe be generalized to something like "that models the average runoff over the study area (Norway)"? I think it is only some of the priors that are particular for Norway, the rest of the framework should be general.

P11L19 Isn't it rather that the alternative is for variables that do not require preservation of water balance?

P12L9 should it be 400 mM/year and 4000 mM/year (delete last s)

P12 Eq8 It is not clear where 0.025 comes from, and I also think the value should have a unit.

P16L2 computationalLY feasible

P16L3 remove "of" before matrix operations

P16L13 marginal distributionS?

P17L3 The default covariance function -> this model was also fitted?

P17L10 might influence or will influence?

P18L21 I like the idea of using CRPS for kriging predictions, but as this is (so far) rather uncommon, maybe also clarify here what the predictive cumulative distribution from kriging is in this context?

P20L6 is it the areal or the centroid model that gives larger uncertainty?

P21L19 I would say it's largest for June and for annual data. The sentence starting with "Particularly" is incomplete.

P31L28 I assume that the areal model might outperform the centroid model also for OTHER AREAL variables that are not mass-conserved?

References:

Clark, I.: Geostatistical estimation and the lognormal distribution. Geocongress. Pretoria, RSA., [online] Available from: http://kriging.com/publications/Geocongress1998.pdf, 1998.
* * *

---

## Author Comment (AC3) · 29 Nov 2019

**Response to the comments on the manuscript (HESSD-2019-415) "A geostatistical framework for estimating flow indices by exploiting short records and long-term spatial averages - Application to annual and monthly runoff"**

T. Roksvåg, I.Steinsland and K.Engeland

November 29, 2019

This is the authors' response to the Anonymous referee (referee #2). We would like to thank the referee for the review and for several useful suggestions that we can use to improve the manuscript. His/her comments have also helped us to see that our main contribution and aim of the work have become unclear. In this response, we will go through the review and provide some suggestions on how we can clarify and edit the manuscript in order to address the referee's main concerns. First, we start with a general comment.

**1 General comment**

The main contribution of this paper is the demonstration that a model with two spatial fields gives benefits for safe use of (very) short records. In the case of annual runoff the two spatial fields are a year specific spatial field and one spatial field that is common for all years, i.e. what we refer to as the climatic field. The combination of these spatial fields enables utilization of short records in a new way. The main goal of the article is to demonstrate this through examples, and we use different time scales (annual and monthly runoff) to show how different hydrological spatial patterns affect the predictive performance of the framework.

We introduce two versions of our two spatial field model: The areal and the centroid model. We compare these methods with the gold standard for spatial interpolation of catchment based data in hydrology; Top-Kriging. The presentation might have been clearer if we instead of comparing to Top-Kriging, compared our model to our own model with only one spatial field (only the year specific field). However, in the paper we prioritized to compare with one of the most recognized methods available for interpolation of streamflow variables.

We see that our main objective, to investigate the implications of including two spatial fields in a geostatistical model for runoff interpolation, can be made clearer in the manuscript. The article should include enough information to verify that the areal model works as we claim, and describe when our two models should be used (areal if we model mass-conserved variables, centroid if we don't care about mass conservation or if we have a point referenced variable like e.g. precipitation). However, comparing the areal and centroid model should not be the main topic of the analysis. The areal model is already documented in (Roksvåg et al., 2019) (but here for a smaller case study of annual runoff where also precipitation observations are included).

**2 Concerns specific for the areal model**

We now go through the referee's comments more specifically. First, the referee has concerns regarding the areal model. He/she writes that *"my main concern is with the method that the authors are most excited about, the areal method. The manuscript leaves the reader with concerns about the benefits, performance and utility of this method"*.

As stated in our reply to the other referee Dr. Gregor Laaha, the areal model has two main benefits compared to the centroid model: 1) It gives a better representation of the posterior uncertainty and 2) its ability to fulfill the

water-balance and distribute the annual runoff correctly over sub-catchments. In our article, we demonstrate benefit 1 through a real case example: Figure 7 shows that the posterior standard deviation of the areal model is different from the posterior standard deviation of the centroid model. Here, we also see that the areal model and Top-Kriging have a similar representation of uncertainty characterized by a larger posterior uncertainty for small catchments. Table 1 showing the coverage percentages, also gives an indication of the the areal model's benefits when it comes to the modeling of uncertainty.

However, we agree that benefit 2 is not demonstrated in this article, and as the referee writes *"the second property proposed by the authors is that the areal method conserves mass. This is only demonstrated hypothetically. As a valuable claim, I think it important to document explicitly."* The reason why it is not included is that we did not find a example where the areal model represented a clear benefit compared to the centroid model when it comes to posterior mean in our cross-validation experiment. As also stated in the response to Dr.Gregor Laaha, benefit 2 of the areal model is probably easier to demonstrate through a simple case example. One example from Voss in Norway is already available on page 15-16 in Roksvåg et al. (2019), accessible at `https://arxiv.org/pdf/1904.02519`. In Roksvåg et al. (2019) we use the same model as in this article, except that also point referenced precipitation data are used in the analysis. Here, the areal representation of nested catchments allowed us to correctly predict larger values in Catchment 3 than any of the observed values (P+A in Figure 5 in Roksvåg et al. (2019)), which was our statement in Section 3.2.6 (page 14-15) in the article under discussion. The centroid model would not be able to do this. As we think that the main point of this article should be to document the methods' ability to exploit short records, the concern of the referee regarding this might be resolved by referring to Roksvåg et al. (2019) and by writing the conceptual example in Section 3.2.6 with more mathematically notation. See our suggestions in Section 5 below.

Next, the referee writes: *"Section 3.2.4. shows how runoff would be accumulated across the drainage area, but I am concerned that this would not conserve mass because, as one example, it does not account for routing. That is, summing all the grid cells, so to speak, on a day does not produce the outlet runoff on that same day"*. The areal model is not suitable for modeling daily runoff for this particular reason. It should be used for variables that are approximately mass-conserved, like the annual runoff, since it does not account for routing.

**3    About the computational feasibility of the methods**

The referee has some concerns about the computational feasibility of the methods, in particular the areal method. We comment these concerns in this section.

The referee writes: *"On utility, the authors acknowledge that the areal model is computationally prohibitive for any real-world application. For example, see line 11 on page 19 and section 3.3, where the author points out that the spatial discretization of the areal method means that several substantial assumptions must be made to simplify the areal method for application. While there is certainly value in presenting a hypothetical model for discussion, this leaves the reader feeling like the areal method is only a hypothesis that cannot be tested"*.

The simplifications mentioned in Section 3.3 are mainly necessary because we are dealing with a full Bayesian model with two spatial fields that need to be estimated ($c(\boldsymbol{u})$ and $x_j(\boldsymbol{u})$), both with several target locations and $x_j(\boldsymbol{u})$ for several years, in addition to having 6 model parameters, all with a prior and posterior distributions. The computational challenges here are met by using the INLA and SPDE methdology, and are used not only for the areal model, but also for the centroid model. Thus, these approximations are not something that is introduced only for the areal model. This can be emphasized in line 5 in Section 3.3, as we see that it can be read this way.

A bit more regarding the INLA methodology: It represents an approximate alternative to MCMC. The approximation is in general accurate and fast, see Rue et al. (2009) for more. INLA has become quite common to use within different fields of science, and has made "unfeasible" Bayesian models computational feasible. See for example Khan and Warner (2018); Opitz et al. (2018); Yuan et al. (2017); Guillot et al. (2014); Ingebrigtsen et al. (2015) for other papers that use SPDE and/or INLA to fit complex Bayesian models.

The areal model is indeed slower than the centroid model due to more target locations, but is not computational infeasible as shown by fitting the areal model $20 \times 4 \times 3$ times: For 20 cross-validation folds, 4 different datasets (annual, January, June, April) and for 3 settings UG, PG and PG-N.

Furthermore, the referee writes that *"the biggest evidence of the weakness of the areal method and the centroid method, and the biggest undercut to the authors' claims of advance, is that, when it comes to application, even these authors do not use their proposed methods. See sections 4.3 and 5.5. where the authors present a new, untested method to reproduce*

*annual values across Southern Norway. The reader is left interested in the hypothetical method, but surprised that it is not used."*

It is not correct that we don't use our own method to produce the annual runoff map: We use the centroid method to produce the map in Figure 15 b. This map is compared to the results obtained by a simpler reference model (Figure 15c) where each year of data is treated separately from each other. While the cross-validation represents a simplification of a real world problem, Section 5.5 is included in order to show that the model actually is computationally feasible when considering 30 years of mean annual runoff (instead of 10) and catchments with different (and realistic) record lengths.

**4   Choice of evaluation set-up and the length of short records**

We here comment the referee's concerns about the choice of experimental set-up/evaluation. In particular the referee criticizes the choice of having a short-record of length 1 and writes: *"Indeed, there is a long history of such procedures, but I find it surprising that authors simulate a partially gauged site as one having on a single year of annual data. This is an extreme, and possibly unrealistic, case of partial gauging that will substantially affect the performance of the methods presented. While it is difficult to work with short records (e.g. 10 years of annual data), I would represent the ungauged case with three or more values."*

As stated above, the centroid and areal model are feasible for real case examples. However, the computational complexity is large when performing a cross-validation for 2 methods, 4 time-scales, 3 settings and 20 folds, i.e. we need to fit the models 480 times for our set-up. Hence, some choices had to be made in order to make a full cross-validation possible. Our choice was to fit 10 years of runoff and include a short-record of length 1. This might not be the most realistic hydrological dataset, but we think that the cross-validation still is valuable. We can look at it as mainly an experiment performed for providing an understanding of how the suggested model works. For example, the cross-validation shows how the increase in predictive performance when adding a short-record is related to the parameter values $\sigma_c$, $\sigma_x$, $\rho_x$ and $\rho_c$ (e.g. in Figure 11 and Figure 12). We are able to show how the posterior uncertainty is distributed (Figure 6, middle plot) for the methods, and how this uncertainty is affected by adding a short-record of length 1 (Figure 13).

Furthermore, it is not that unrealistic to have a short record of length 1. Considering the Norwegian data showed in Figure 15a, five of these catchments have only 1 annual observation, five catchments have 2 annual observations and 10 catchments have only 3 annual observations between 1981 and 2010. Our cross-validation shows how important a small bit of information like this can be if the weather patterns (thus also the model parameters) are similar to the Norwegian case ($\sigma_c >> \sigma_x$ and $\rho_c < \rho_x$).

Here, it is also important to note that we provide a model where it is relatively risk-free to include very short record lengths in our framework, which is also one of our main contributions. The model itself figures out if the study area is driven by a stable hydrological pattern that repeats itself every year ($\sigma_c > \sigma_x$) or more local year-dependent effects ($\sigma_c < \sigma_x$). If the latter is the case, the short record will not influence the results particularly. It will only have an influence for the particular year for which we have data. The monthly predictions demonstrate this point: In January the spatial patterns of runoff are not stable in Norway, and including a short record of length 1 (PG in Table 2) does not affect the model negatively compared to the UG case (Table 1). If, on the other hand, $\sigma_c > \sigma_x$, the suggested model uses information both in time and space, and a short record will on average influence the predictive performance positively.

This leads us to another point stated by the referee: *"I suggest dropping the sections on monthly analysis. The simulation of monthly streamflow tends to imply that one is producing monthly sequences line Jan-Feb-Mar, but this work is looking at Jan-Jan-Jan (for example). This is akin to only predicting a new statistic of streamflow and is not a novel advance of the method. While it could be expanded to provide a more robust analysis, removing it might help streamline the manuscript".* The monthly predictions are mainly included to illustrate the method for a different hydrological regime and a different set of parameters. The monthly predictions can illustrate that the model itself adjusts the two spatial effects $x_j(\boldsymbol{u})$ and $c(\boldsymbol{u})$ relatively to each other, and that the impact of including a short record when $\sigma_x > \sigma_c$ is small (however, it does not affect the model negatively either). We suggest rephrasing the monthly predictions part to emphasize why these can be informative about what we can expect from the methods for other parts of the world where other precipitation types dominate.

Furthermore, the referee writes: *"Given that the use of one point for partial gauges and two points for regression (line 12, page 24), it would seem wise to use a consistent number of points to represent partial gauging."* There is already an answer to this in our previous reply to Joris Beemster: We chose to use a record length equal to one for our methods to emphasize that 1) our methods are able to exploit a short record of length one. Linear regression requires a record

of length two or more. 2) We can provide predictions approximately equally as good or better than linear regression by using a shorter record length.

Again; it is a main contribution to provide a model where also very short records can and should be included.

**5    Suggestions to address the main concerns**

In order to address the main concerns of the referee, the authors' have the following suggestions for clarifying and improve the manuscript:

- Keep the cross-validation set-up as it is and consider it as a demonstration of the model properties (and parameter values).

- Keep the monthly predictions, but rephrase this part and emphasize that these predictions are included to illustrate the model performance for a different set of parameters, i.e. these could represent the performance for a other part of the world with a different hydrological regime. The monthly predictions show that it is safe to include short records even if the annual patterns are more unstable than the Norwegian annual patterns since the results are not affected negatively for regions where $\sigma_x > \sigma_c$.

- Edit the conceptual example in 3.2.6 in order to explain the areal model in more detail. Here, we can write out parts of the model more mathematically to show 1) how we can predict larger values than any of the observed values and 2) how we go from areal observations to point observations and the other way around. Furthermore, we can refer to Roksvåg et al. (2019) in order to document a case for which the areal model makes a difference compared to the centroid model when it comes to runoff modeling.

- Include a more realistic case example e.g. based on the data in Figure 15a, where we predict the mean annual runoff for a 30 year period for some selected fully gauged catchments (which we first treat as ungauged for the evaluation, then partially gauged with different record lengths). In this example, we can use short records from neighbouring catchments as they are (the record lengths are 1-30 years, with average 15 years), and compare the performance of the areal, centroid and TK methods for predictions of mean annual runoff (for the 30 years period as a whole). By this we aim to show that the suggested models are feasible also for a realistic case, and demonstrate the differences between the methods for a bigger dataset. This experiment can replace Section 5.5, and will also provide an assessment of the methods' performance on mean annual runoff as requested by referee Dr.Gregor Laaha.

**6    Minor comments**

Finally, we comment two of the referee's minor comments.

*"Page 11, line 2: Why would we expect the long-term spatial average runoff (c(u)) to have a zero mean".* The effect $c(\boldsymbol{u})$ has indeed zero mean. The long-term spatial average runoff, however, has mean equal to $\beta_c$, i.e. we have a distinct component/parameter to model the mean.

*Page 11, line 4: Why would expect a sequence of annual values to be independent? Is this true for monthly values?* The sequence of annual/monthly values are not modeled as independent. They are dependent through components $c(\boldsymbol{u})$ and $\beta_c$ that are common for all years. In this line, we talk about the year-specific spatial fields $x_j(u)$. These are assumed to be independent realizations, or replicates for $j = 1, ..., r$. However, the models are stationary in time. This can be supported by looking at time series of mean annual runoff: We don't see any e.g. increasing/decreasing trend or change in spatial pattern over time. Stationarity in time also makes sense when modeling monthly time series as $Jan - Jan - Jan$, and not $Jan - March - Feb$.

The other comments will be taken into account.

**References**

G. Guillot, R. Vitalis, A. le Rouzic, and M. Gautier. Detecting correlation between allele frequencies and environmental variables as a signature of selection. a fast computational approach for genome-wide studies. *Spatial Statistics*, 8:145 – 155, 2014. ISSN 2211-6753. doi: https://doi.org/10.1016/j.spasta.2013.08.001.

R. Ingebrigtsen, F. Lindgren, I. Steinsland, and S. Martino. Estimation of a non-stationary model for annual precipitation in southern Norway using replicates of the spatial field. *Spatial Statistics*, 14:338 – 364, 2015.

D. Khan and M. Warner. A bayesian spatial and temporal modeling approach to mapping geographic variation in mortality rates for subnational areas with r-inla. *Journal of data science: JDS*, 18:147–182, 01 2018.

T. Opitz, R. Huser, H. Bakka, and H. Rue. Inla goes extreme: Bayesian tail regression for the estimation of high spatio-temporal quantiles. *Extremes*, 21, 02 2018. doi: 10.1007/s10687-018-0324-x.

T. Roksvåg, I. Steinsland, and K. Engeland. A knowledge based spatial model for utilising point and nested areal observations: A case study of annual runoff predictions in the Voss area. *arXiv:1904.02519*, 2019.

H. Rue, S. Martino, and N. Chopin. Approximate Bayesian inference for latent Gaussian models using integrated nested Laplace approximations. *Journal of the Royal Statistical Society: Series B (Statistical Methodology)*, 71:319–392, 2009.

Y. Yuan, F. Bachl, F. Lindgren, D. Borchers, J. Illian, S. Buckland, H. Rue, and T. Gerrodette. Point process models for spatio-temporal distance sampling data from a large-scale survey of blue whales. *The Annals of Applied Statistics*, 11:2270–2297, 12 2017. doi: 10.1214/17-AOAS1078.

---

## Author Comment (AC4) · 5 Dec 2019

**Response to the comments on the manuscript (HESSD-2019-415) "A geostatistical framework for estimating flow indices by exploiting short records and long-term spatial averages - Application to annual and monthly runoff"**

T. Roksvåg, I.Steinsland and K.Engeland

December 5, 2019

This is the authors' answer to referee Dr. Jon Olav Skøien. We would like to thank him for several insightful comments and constructive suggestions.

His review suggests, as the other reviews, that we include an analysis of the mean annual runoff. We suggest to do this as described in our response to Anonymous referee # 2. We think that performing this analysis (and adding some more discussion) will resolve most of the referees' concerns regarding the case study and choice of record length in the case study (record length 1). The cross-validation will be a case study to illustrate the model properties of our two field model, while the assessment of mean annual runoff will be a test of the method on a more realistic hydrological dataset.

The rest of the comments of Dr. Jon Olav Skøien are mainly specific comments that can be resolved by adding some more discussion/rephrasing sentences. We comment some of them here. The remaining comments will be taken into account by adding more discussion/rephrasing.

**1 Specific comments**

*"P25 – description of Figs 11-12. It is mentioned that UG has problems predicting large values of runoff, but I'd say it is just as difficult with small values. Additionally, the negative values should be mentioned here, not only in the conclusions. It seems there are no negative values for annual runoff for UG? I think this result is partly related to the fact that the data don't follow the assumption of being normally distributed, which should be discussed. A transformation method such as logtransform could avoid this problem, although log-transformed data on the other hand don't go so well with the linear aggregation assumption, see also Clark (1998)."*
As the referee suggests we did not transform the data because it doesn't go so well with the linear aggregation assumption in the areal model. However, it could be done for the centroid model. This can be mentioned.

*"Figure 1b is a bit difficult to understand, including the reference to 0-4 catchments. These types of figures are generally difficult to make nice, so I'm only asking the authors to test if other visualizations could work better. The lines of 4 catchments is difficult to see. Maybe it would be possible to add river network on top, to understand the catchment order?"*
We will search for a better way of visualizing.

*"P27L10 I do not think it is correct that observations from nearby catchments have a larger impact on the target catchment for the centroid model when rangec > rangea. It should be the opposite, large range means that stations further away will also get a weight in the interpolation."*
Yes, this is correct and the sentence should be rephrased. The point here was that the range $\rho_c$ for the areal model is very low, lower than for the centroid model. That means that the predicted runoff for an ungauged catchments that is located far away from gauged catchments, goes towards the mean value $\beta_c$ instead of being influenced by neighboring observations. The low range $\rho_c$ for the areal method might be connected to our relatively strong constraint on the measurement uncertainty for the areal observations. The tendency of providing a low range might also be a contributing

reason for why the areal model performs poorer than Top-Kriging for the ungauged case. This can be mentioned in the discussion.

*"Then also the next sentence (L12-15) seems somewhat incorrect. If the values in Table 3 are divided by the similar values in Table 1, maybe the differences could easier be interpreted in light of the range?"* We agree. Again, the point was that a low range means that the predicted runoff for the ungauged catchments goes towards the mean $\beta_c$. We will look through this part and rephrase.

---

## Author Response (AR1)

Dear editor.

We are grateful for the insightful comments from the reviewers on the paper originally entitled "*A geostatistical framework for estimating flow indices by exploiting short records and long-term spatial averages – Application to annual and monthly runoff*". We have addressed their concerns and modified the manuscript accordingly.

In this response, we summarize the biggest changes that are done to the manuscript. First, we address your four main concerns. Next, we address comments posted by the reviewers on the HESSD forum discussion that are not already discussed. Furthermore, a marked-up version of the original manuscript is available in the end of this reply (this should be opened in Adobe reader to work properly). However, as there are substantial changes in both manuscript structure and content, we think that reading the "summary of changes" section below is the best strategy for getting an overview of the changes we have made.

We again thank the reviewers for their constructive feedback. Hopefully, our changes have clarified the paper and made it more relevant for the readers such that it can be acceptable for publication.

Best regards,
Thea Roksvåg and co-authors.
* * *
**Summary of changes**

**Editor comments**

**Editor**: (1) There are methodological concerns regarding the "areal model". The "areal model" is one of the main (claimed) innovations of the paper. The authors should provide evidence that the areal model indeed improves results. Significance testing is an option here. The paper should also dig deeper in the issue of mass conservation of the "areal model".

**Major changes:** We have now improved the explanation of the areal model. This can be found in Section 4.2.2. This section is rephrased to clarify how the areal model considers water-balance, and how we are able to predict values that are larger than any observed values. The conceptual figure (original Figure 5) is replaced by a new figure with fewer grid nodes to make the notation simpler. See the new Figure 6.

Lines 26-30 on page 12 are added to emphasize that we put constraints on the observed runoff over the gauged catchments through the observation likelihood. We have also added some sentences in Section 4.1.5 (page 15, lines 17-27) to clarify the relationship between point referenced runoff and areal referenced runoff, and the linear aggregation (according to Equation 11).

There is also a new Figure (Figure 10) that can contribute to clarify the relationship between the point predictions ($q_j(u)$) and the areal predictions ($Q_j(A)$).

As we wrote in the HESSD discussion, we did not find an example where the water-balance considerations of the areal model represented a clear benefit over the centroid model for this dataset. This is now discussed more in Section 7 (Discussion). See page 32, line 1-21. We have not searched for a new case example showing the mass-conserving properties of the areal model because it already exists in Roksvåg et al (2019). This work is referred to in the introduction, the discussion and in Section 4.2.2. We think the main focus of the analysis should be to describe the models' ability to exploit short records, and that the areal model and the centroid model only are two different versions of this framework. We have tried to emphasize this focus by e.g. removing the reference of the areal model's mass conserving properties in the abstract (page 1, line 6 in the original MS) and by line 28 on page 3 in the new MS where we state that the two models only are two *versions* of the suggested framework. In the new discussion we also try to be more objective when evaluating our two methods (i.e. we don't have a favorite method) and highlight both benefits and drawbacks by the centroid and the areal model. See Section 7.1.

**Editor**: (2) The test cases presented in the results section should be re-considered. The reviewers suggest that some cases could be omitted from the manuscript. On the other hand, reviewers suggest to extend the partially gauged case including more observations there.

**Major changes:**
We have kept most of the original case study of annual and monthly runoff, as we think these results are important for two reasons:
* We want to show that our model is able to handle a short record of length one, and illustrate that it is safe to include very short records in the model regardless of the underlying weather pattern. Short records of length one can also can contribute to a large improvement in the RMSE/CRPS for some areas/climates.
*The monthly predictions show the framework's behavior for another set of parameters. This can be used to indicate how the framework works for other climates and/or hydrological variables that are driven by more unstable hydrological processes than what we have in Norway.

However, the monthly predictions part is rephrased as requested by more than one of the referees. We have emphasized that this assessment is included to show the framework's behavior for a different climate/hydrological regime, and a different set of parameters. This can be seen e.g. in the introduction (page 3 line 23-27), in the description of the data (page 6, line 6) and in the discussion (page 33, line 5-13,).

To reduce the focus on the monthly predictions as a new flow index, we have removed the original Section 3.2.7 (extension to monthly runoff), and instead written about the monthly predictions in Section 5.1 as a part of the experimental set-up. Here, we again emphasize that we do these predictions to learn something about the framework's performance for other

climates (page 19, line 19). Furthermore, we have changed the title to remove the focus on the monthly predictions.

To reduce confusion, we write "annual time series of monthly runoff" in the revised manuscript instead of "time series of monthly runoff". See e.g. page 6, line 6. This is to emphasize that we have time series on the form: Jan-Jan-Jan and not Jan-Feb-March.

Also mark that we have added a sentence on page 7, line 8 about the number of catchments in the study area with only 1-3 annual observations (20 catchments). This is done to show that catchments with (very) short records exist, which can motivate the experimental set-up. We also discuss on page 35, line 1, that the model can be used to assess the value of collecting one annual observation. This is another motivation for having a case study with a short record of length one.

As requested by the reviewers, we have removed the original Section 5.5 and replaced it by an assessment of the framework's performance on predictions *mean* annual runoff for a 30 year period. Hence, in the revised manuscript we demonstrate the framework for both infill of missing annual observations and for mean annual runoff interpolation. Based on this, the main objectives of the paper are changed. See page 4, lines 12-18. The dataset we use for mean annual runoff is presented in Section 2, page 7 (line 1-11) and in the new Figure 5. The experimental set up for this new experiment is described in Section 5.2. We use Top-Kriging as the reference method, where we set the uncertainty based on the record length as suggested by referee Dr. Gregor Laaha. We also test the methods for different record lengths, not only 0 and 1, as requested. Hence, this experiment represents a more realistic case.

As we now have two different prediction types (infill and mean annual runoff), the notation in the new Section 5.3 for RMSE/CRPS/ANE/r^2 had to be changed to be more general. Many of the figure texts and subsection titles were also changed to clarify which experiments we are talking about (infill or mean annual runoff). The new mean annual results are presented in the new Section 6.2. We also compare the results to other studies in the discussion (7.2). This replace the original Section 5.4 and Figure 14.

Note that we removed the PG-N case that was present in the original study. This choice was made in order to make space for the important results for mean annual runoff. Hence, the original Figure 10 and Section 5.3 is removed.

**Editor:** (3) The discussion should be re-organized. Part of the discussion is a repetition of results and a more strict division the results and the discussion sections can be made. Next, it is important to extend the discussion regarding the generalization of your results. It is important to discuss how the method could work in other flow regimes and climates. The role of human intervention (e.g., dams) is also an important point to be discussed.

**Major changes:** The whole discussion is rewritten. We have included several of the topics suggested by the referees, e.g. more comparisons between the three methods (centroid vs. areal on page 32, lines 1-21 and our framework vs. Top-Kriging on page 32, line 22-30), comparison of computational speed (page 32, lines 13-22), performance for other climates

(page 33, lines 5-13), performance for other gauging densities (page 33, lines 14-19) and the role of regulated catchments (page 34, line 17). In the discussion we also have the comparison with other studies (Section 7.2) as requested by the reviewers. Please read the new discussion.

**Editor:** (4) Please consider also sharpening your conclusions.

**Major changes made:** We have rewritten the conclusion. Please read the new Section 8. The abstract is also rewritten to fit better to the conclusion and the new results/discussion.

**Other changes that should be mentioned:**
* Originally, there were 15 regulated catchments included in the dataset for which one of them significantly affected the results. We have re-run the analysis without these 15 regulated catchments. Hence, the values in all result tables (original Table 1-3) are updated. The figures in the result section are also updated, and some of the numbers and figures in the presentation of the study area (Section 2) are updated as we now have 180 catchments for cross-validation and not 195. The main conclusions remain the same with this new dataset.
The regulated catchments should have been removed from the analysis from the beginning as catchments that are significantly influenced by human activity should not be included in a model like this. Such catchments can e.g. lead to negative runoff. The impact of regulated catchments is mentioned in the discussion, page 34, line 17.
* Code for the centroid model with example data from Norway is now available on github (http://www.github.com/tjroksva/runoffinterpolation , doi: 10.5281/zenodo.3630348). This is stated on page 19, line 8.
* We separated the background chapter (original chapter 3.1) from the model developing chapter (original chapter 3.2). They are now chapter 3 and 4 respectively.

This was a summary of the major changes done to the manuscript to address the major concerns of the editor. Below we have included comments from the HESSD forum discussion that are not already covered above, and refer to changes that are made to address them.

**Comments from reviewer 1: Dr. Gregor Laaha.**

**Reviewer 1:** Study area: "This leaves 195 catchments for testing with areas ranging from 7.5 km$^2$ to 18934 km$^2$." (p4 line 16). How many of them are nested?»

**Answer:** This is now written on page 5, line 4.

**Reviewer:** «Section 3.1.4: Please make clear whether such regression methods have been used for estimating annual discharge."

**Answer:** Linear regression is mainly included as a simple reference method (base model). As it performs quite well on annual runoff in Norway, it says something about the large correlation in the study area. We don't know about an example where linear regression in its simplest form is used to estimate annual discharge.

**Reviewer 1:** "Section 3.3.1: monthly rainfall is not the scope of this paper, why not using annual runoff as an example?"

**Answer:** The example is changed as suggested.

**Reviewer 1:** «Section 3.2.1: "Likewise is c(u)a spatial effect that models the long-term spatial average of runoff, or the spatial variability caused by climatic conditions in Norway..." (p10, line 30) - I think this interpretation is not sound, it is the combined effect of climate and catchment characteristics that lead to spatial variability of runoff. (This interpretation occurs several times throughout the MS). "... while xj(u) is a year specific spatial effect that models the spatial variability due to annual discrepancy from the climate." (p10, line31f): This could also be formulated in a clearer, more meaningful way."

**Answer:** The word "climate" is used to describe both long-term weather-patterns *and* runoff generation due to catchment characteristics that are static. This is done for simplicity. The climatic spatial field captures all long-term effects. We have now emphasized this two places in the manuscript: In the introduction (page 3, line 4-5), and in the model specification (page 11, line 22-23).

**Reviewer 1:** "Firstly, the underlying approach is a GRF decomposition by a linear model (Eq. 4). It would be interesting to see the importance (magnitude) of each effect. This will be informative about whether the yearly deviation from the annual pattern is rather constant, or has a spatial structure. This can be summarized in a table and in an additional plot of maps showing the spatial variability of the annual residual (range of xj(u)) as compared to the average spatial pattern c(u)."

**Answer:** We have now included an example in Figure 10 where we show the spatial components c(u) and x_j(u) for two selected years. This plot is included in order to show that almost all the spatial variability for these years can be explained by long-term effects (c(u)), and that the range rho_c is small. This figure can also help the reader to understand the relationship between the point predictions (q_j(u)) and the areal predictions (Q_j(A), for example in Figure 7). Apart from this, the magnitude of each effect is already indicated from the parameter values in Table 2 (from sigma_c and sigma_x) and discussed several places in the results and discussion section.

**Reviewer 1:** "P11, line 19: Centroid model: "This alternative does not require preservation of water- balance and can be used for any environmental variable". Think the model "does not allow" for preservation of the water balance and is therefore not well suited for runoff and runoff-related variables, but can be applied for other environmental variables. "

**Answer:** This is rephrased as **"**This model does not consider preservation of the water-balance, but on the other hand it can be used for any point referenced environmental variable…".

**Reviewer 1:** "Several times: Hydrological stability is rather an abstract term that can be interpreted in different ways. Consider using low inter-annual variability instead."

**Answer:** We have tried to solve this by rephrasing what we mean by "hydrological stability" (now renamed as hydrological spatial stability) in the introduction. See page 3, line 17-18. It is also repeated on page 14, line 10. We have used this phrase to ease the "notation". The choice of words is difficult here, as what we mean is that the model has its benefits when the spatial variability is stable over time. The variability between years can still be large (captured by beta_j), but the spatial variability c(u) should be stable over time.

**Comments from Joris Beemster**

**JB:** It would be valuable to mention the amount of nested catchments and degree of "nestedness". Currently, figure 1b gives an indication, but no reference to this figure is made in the study area description. Adding a couple of sentences mentioning the amount of nested catchments, as well as, a reference to figure 1b, will also improve the study area description".

**Answer:** We have now referred to figure 1b and written how many of the catchments that are nested. See page 5, line 4-5.

**JB:** "Lastly, it is unclear to me why records from all over Norway are used (figure 1a), but all maps in the results section only show the results for southern Norway. It seems more consistent to limit the analysis to southern Norway or to present the results for the entire country to increase transparency".

**Answer:** This is mainly done to save space, and to make some of the figures clearer. However, Figure 10 is new and this shows northern Norway. Apart from this, we think the other figures (of southern Norway) are sufficient in order to show what we want to show.

**JB:** "p1, line 17: Please provide more than one reference if you state that "Average annual flow is often used..."

**Answer:** Rephrased to "average annual flow can be used…" and added one reference.

**JB:** "p3, lines 19-20: "A similar model has already shown promising results". Please mention how it differs.»

**Answer:** We have rephrased this part. See page 3, line 32-34. This is done to clarify our contributions relative to Roksvåg et al, 2019.

**JB:** "p9, lines 29-30: "However, it has been shown that Top-Kriging also performs well for variables that are not mass conserved, like e.g. the specific 100-year flood". Please support this statement by one or more citations.»

**Answer:** We have removed these sentences to save space as they are not directly relevant.

**JB:** "p10, lines 30-31: "Likewise is c(u) a spatial effect that models the long-term spatial average of runoff, or the spatial variability caused by climatic conditions in Norway". The spatial variability is not only caused by climatic conditions, but also by the catchment characteristics. "

**Answer:** The word "climate" is used to describe both long-term weather-patterns *and* runoff generation due to catchment characteristics that are static. This is done for simplicity. The climatic spatial field captures all long-term effects. We have now emphasized this two places in the manuscript: In the introduction (page 3, line 4-5), and in the model specification (page 11, line 23).

**JB:** "p12, lines 10-12: Petersen-Øverleir (2004) indeed shows that heteroscedasticity is a widespread problem of Norwegian gauging stations. However, he also shows that differences between gauging stations are large and that there is at least one example where the uncertainty decreases with increasing runoff. Please motivate why this value for the scaling factor was chosen. "

**Answer:** We added a sentence about the scaling factor on page 13, line 15. We could have set individual uncertainties for all catchments, but these numbers are not available for all catchments in Norway. The solution was then to use 0.025y_ij which can be considered as the prior average uncertainty over all catchments in Norway.

**JB:** "p12, lines 22-23: Please provide a source for the following statement: "This corresponds well to what we know about the measurement uncertainty for runoff in the study area.""

**Answer:** NVE (the data provider) is the source for this statement. This is now added on page 13, line 27-29.

**JB:** "Inference (3.3): In this section several simplifications are mentioned aimed at reducing the computational complexity. Could you please comment on the effect these simplifications have on the expected outcome?»

**Answer**: We have added some sentences about the accuracy of INLA/SPDE on page 18, line 30-34, and added some references.

**JB:** "p17, line 31: It would be interesting if the case of ungauged neighbors is also evaluated. Likely, large improvements will be seen in the PG-N, relative to the UG-N case for the new methods that are less apparent for Top-Kriging.

**Answer:** This was a good suggestion. As we did not prioritize to add UG-N, we instead removed PG-N to make space for the mean annual runoff results.

**JB:** "Evaluation scores (4.2): The performance of the model is mainly evaluated in terms of RMSE and CRPS, two evaluation scores that are scale-dependent. In my opinion, adding a scale independent evaluation score, such as the Nash-Sutcliffe or the Kling-Gupta efficiency, would make the comparison between averaged annual and monthly runoff more straightforward. Furthermore, this would enable the evaluation in terms of the correlation, the conditional bias and the unconditional bias (Gupta et al., 2009). "

**Answer:** We did not prioritize to change evaluation scores. However, we have results for ANE and $r^2$ in Figure 14 and 15 that are scale independent. These give the same conclusions as the RMSE and CRPS for the Norwegian data.

**JB:** "Figures 8-10: The units of the y-axis are not mentioned. Please add them. In my opinion these figures could be left out of the paper, because they are made redundant by table 1-3 and figure 7 .»

**Answer:** Units are added to the figure text. We have removed the original figure 10 (and Table 3) as the PG-N case is removed. We want to keep the original figure 8 and 9, because they show the spread in the predictions, in addition to the summary statistics in the original Table 1-3.

JB: "p24, line 11-13: "However, recall that a short-record of length 2 from the target catchment is needed in order to use this method, while our areal model performs approximately equally well with a short- record of length 1 (and observations from other neighboring catchments)." If you would also test the areal and centroid method for partially gauged catchments with a record length of two, the comparison with linear regression would be more straightforward. "

**Answer:** We want to keep the case with short-record of length one because it shows what our method is capable of. That is, safe use of very short records, and that short records of length one also can have a large effect on the predictions.

**JB:** "Minor textual suggestions…"
**Answer:** Thank you. These are taken care of.
* * *
**Comments from reviewer 2 (Anonymous).**

**Reviewer 2:** "My main concern is with the method that the authors are most excited about, the areal method. The manuscript leaves the reader with concerns about the benefits, performance and utility of this method."

**Answer:** Based on this we have tried to clarify that we mainly focus on the short records properties of the framework. The main focus has not been to give a comparison of the

centroid and areal model. These are only two versions of the framework. For example is the following sentence in the abstract removed: "*Another property, is that the model takes the nested structure of catchments into account such that the water balance is preserved for any point in the landscape*". In the modified introduction we write: "*In the following presentation, we introduce two versions of our framework, i.e. two geostatistical models*» to clarify that there are two *versions* of the methodology. In addition, we have also added more discussion around the centroid vs. the areal model in Section 7. See page 32, line 1-21. Here we are a bit less enthusiastic about the areal model to make the discussion fit better to the results actually presented in this paper. Apart from this, we refer to Roksvåg et al (2019), in order to "prove" the mass-conserving properties of the areal model.

**Reviewer 2:** "The second property proposed by the authors is that the areal method conserves mass. This is only demonstrated hypothetically. As a valuable claim, I think it important to document explicitly."

**Answer:** As the main topic should be the short record properties of the framework, we document this by referring to Roksvåg et al (2019). As already mentioned, the conceptual example in Section 3.2.6 is also changed to make the theory clearer. This is now found in Section 4.2.2.

**Reviewer 2:** "Section 3.2.4. shows how runoff would be accumulated across the drainage area, but I am concerned that this would not conserve mass because, as one example, it does not account for routing."

**Answer:** It does not account for routing since we are applying the framework for time-aggregated runoff variables for which the transport time in the river network can be neglected. This is now emphasized on page 16, line 1-4.

**Reviewer 2:** "On utility, the authors acknowledge that the areal model is computationally prohibitive for any real-world application. For example, see line 11 on page 19 and section 3.3, where the author points out that the spatial discretization of the areal method means that several substantial assumptions must be made to simplify the areal method for application. While there is certainly value in presenting a hypothetical model for discussion, this leaves the reader feeling like the areal method is only a hypothesis that cannot be tested."

**Answer:** The only simplification that is unique for the areal model is the discretization of the catchments. Otherwise, the simplifications mentioned in Section 3.3 (now Section 4.3) are used for both the areal and the centroid model. We have now emphasized this on page 18 line 9-11 by mentioning that the simplifications are done for both methods, and we have added some references to show that the simplifications are used for other studies (page 19, line 3-5). We have also added a discussion around the computational complexity of the methods in Section 7.1. See page 32, lines 13-21. We also hope that our new assessment of mean annual runoff for southern Norway (Section 6.2) shows that the areal model is feasible for a real case example (30 years of data).

**Reviewer 2:** "Line 23 claims that a few years of data could be useful for estimation. Indeed, there is a long history of such procedures, but I find it surprising that authors simulate a partially gauged site as one having on a single year of annual data. This is an extreme, and possibly unrealistic, case of partial gauging that will substantially affect the performance of the methods presented. »

**Answer:** We have chosen to keep the original case study for reasons that are already stated. We have added a sentence about the number of catchments in the dataset with short records of length 1-3 (20 catchments) on page 7, line 8, to show that the case is not that unrealistic.

Furthermore, we have added the assessment of the methods for mean annual runoff. Here, we use varying record lengths (0, 1, 3, 5, 10 out of 30 years). See Figure 13-15.

**Reviewer 2:** "The simulation of monthly streamflow tends to imply that one is producing monthly sequences line Jan-Feb-Mar, but this work is looking at Jan-Jan-Jan (for example)."

**Answer**: To reduce confusion, we write "annual time series of monthly runoff" instead of "time series of monthly runoff". See e.g. page 6, line 8. This is to emphasize that we have time series on the form: Jan-Jan-Jan and not Jan-Feb-March. We have also changed the title to reduce the focus on the monthly predictions, as mentioned in the reply to editor.

**Reviewer 2:** "Finally, I'd love to seem some additional analysis on the regional variability of performance of these methods. What seems to drive the varying levels of performance across Norway?»

**Answer**: We have added some discussion around this in Section 7. See page 32, subsection 7.1

**Reviewer 2:** «Page 6: This discussion focusses on whether or not the lines look parallel. This is highly subjective and should be quantified in some way. For example, if I changed the vertical axis to run from 0 to 100,000, all the lines would "look parallel". Please provide some quantification of correlation.»

**Answer**: After thinking about this, it is not really the correlation that matters, but the difference in runoff between two locations over time, i.e. what we describe as hydrological spatial stability in the introduction. The spatial stability can be quantified through the parameters of the GRFs, i.e. through sig_c and sig_x, but apart from this it is difficult to quantify these patterns by e.g. correlation. We decided to keep the figures as they are, as we think that they give an indication of the statistical patterns in the study area and the potential gain of including short records. We also tried to improve our explanation of what spatial pattern we refer to (hydrological spatial stability, page 3, line 17.)

**Reviewer 2:** «Page 11, line 2: Why would we expect the long-term spatial average runoff (c(u)) to have a zero mean? Page 11, line 4: Why would expect a sequence of annual values to

be independent? Is this true for monthly values?»

**Answer**: The long-term spatial average has zero mean as the mean in the model is captured by beta_c. The sequence of annual values are not independent. They are dependent through c(u). However, the components x_j(u) for j=1,..,r are regarded as independent realizations of the underlying GRF. This is the year specific part of the model.

**Reviewer 2:** "Page 22: The coloring of figures like Figure 7 make it difficult to see the variability of performance. The results appear highly skewed, resulting in almost all RMSEs being brown; an alternative scale might distinguish performance better. "

**Answer:** We have tried with different color scales, but they all give one color for eastern Norway. It is difficult to find a color palette that is suitable because of the large spatial variability in Norway. A solution can be to take the log of the results. However, this makes the results more difficult to interpret. We have chosen to keep the color scale. As we now have new results with a dataset (without regulated catchments), the scale limits for the RMSE are more narrow in the revised manuscript and the results are easier to see. This partially solves the problem.
* * *
**Comments from reviewer 3 (Dr. Jon Olav Skøien).**

**Reviewer 3:** Spatio-temporal kriging will most likely not do much better than the spatial interpolation, as it will use the model based covariance rather than the observed covariance also for the PG case, but I think it should be mentioned. It might be a good alternative for a time series with a few (maybe non-consecutive) missing observations.

**Answer:** The reason for not trying/using a spatio-temporal model with a time trend is that the time dependency in the data is low for annual runoff (considering one location). What happens this year don't affect next year (apart from the underlying *constant* climatic effect). We have mentioned this briefly on page 16, line 1-4 in the revised MS.

**Reviewer 3:** P3L3 It is referred to how the model is developed for annual datasets, but might be used for indicators with other temporal supports (such as monthly). However, it is never explained why there is a difference between monthly or annual data, except for different correlation lengths etc. I guess this is related to the comments on P3L15, were it is referred to the water balance being "close to preserved . . . with some uncertainty". However, the "almost preserved" is never explained. The same description is used on P11L13. Could the method also be used for daily data? If not, why?

**Answer:** We have rephrased the article such that we now present the method as a method for annual runoff. We have also chosen a new manuscript title that reflects this. However, we also state that "*The framework we suggest is flexible can be used for any hydrological variable.*

*However, its benefits are linked to exploiting long-term spatial trends in the data, and in order to work better than other interpolation methods, the hydrological variable of interest should be driven by weather patterns that are repeated over time. For this reason, we develop our methodology for annual runoff."* This is added in the introduction, page 3 line 13-16. By this we mean that either the centroid or areal model can be used for the variable of interest, but that the framework only has benefits if there is some underlying long-term pattern.

Regarding the mass-conserving properties, the example in Section 3.2.6 is rephrased to explain this part better. The uncertainty in the predictions is now linked to the strict priors. See page 17, line 1-10.

**Reviewer 3:** P25 – description of Figs 11-12. It is mentioned that UG has problems predicting large values of runoff, but I'd say it is just as difficult with small values. Additionally, the negative values should be mentioned here, not only in the conclusions. It seems there are no negative values for annual runoff for UG? I think this result is partly related to the fact that the data don't follow the assumption of being normally distributed, which should be discussed. A transformation method such as logtransform could avoid this problem, although log-transformed data on the other hand don't go so well with the linear aggregation assumption, see also Clark (1998).

**Answer:** We removed the sentence that said that UG has problems predicting large values of runoff. As you say, we see it also for smaller values.  We have also added that it is possible to avoid negative values by using a log transform, but that this only works for the centroid model due to the linear aggregation assumption. See page 34, line 11-13. Regarding negative values, we have also added that negative values almost never appear for mean annual predictions. See page 34, line 22-23.

**Reviewer 3:** Figure 1b is a bit difficult to understand, including the reference to 0-4 catchments. These types of figures are generally difficult to make nice, so I'm only asking the authors to test if other visualizations could work better.

**Answer:** We have made a better version of the figure. There are less nested catchments in the new dataset (as regulated catchments are removed). Hopefully, the Figure is a bit easier to understand (river networks were not available). We also added a sentence about how many of the catchments that were nested. See page 5, line 4-5.

**Reviewer 3:** "The discussion at the end of P31 should also include some thoughts around gauging density. If the density is high, it is more likely that catchments are nested. Can the centroid model be expected to be as good as the areal model for non-nested observations? And when mentioning other environmental variables, it should maybe be stated that these are point values?»

**Answer:** We now write «any point referenced environmental variable» instead of «any environmental variable". We have added discussion around the gauging density (page 33, line

14-19) and a comparison of the centroid and the areal model  (page 32, line 1-21.).

**Reviewer 3:** "P1L5 "The climatic GRF . . ."- I think this could be rephrased. If I understand correct, the GRF learns the spatial pattern from a limited number of years, and can use this information to improve predictions for years without observations."

**Answer:** For simplicity we call the component the climate. If we have for example 30 years of data, it will give a good approximation of the climate. If we have only 10 years, it will be a poorer approximation of the climate.

**Reviewer 3:** «P10L30 "in Norway" – could maybe be generalized to something like "that models the average runoff over the study area (Norway)"? I think it is only some of the priors that are particular for Norway, the rest of the framework should be general."

**Answer:** Good point. This is changed. We have also added that the priors are specific for the study area (page 13, line 4-5).

**Reviewer 3:** P12 Eq8 It is not clear where 0.025 comes from, and I also think the value should have a unit.

**Answer:** Added a sentence about this on page 13, line 15. It's unit is given by y_ij.

**Reviewer 3:** "P17L3 The default covariance function -> this model was also fitted?"

**Answer:** Replaced "the default covariance function was used" by "the default covariance function was fitted".

**Reviewer 3:** "P17L10 might influence or will influence?"

**Answer:** Might because we don't know whether the climatic spatial field c(u) is larger than 0 yet. However, we replaced "might" by "can" in this sentence to make it less confusing.

**Reviewer 3:** "P18L21 I like the idea of using CRPS for kriging predictions, but as this is (so far) rather uncommon, maybe also clarify here what the predictive cumulative distribution from kriging is in this context? "

**Answer:** We use the Gaussian cumulative distribution for all experiments. This is now written on page 22, line 13-14.

**Reviewer 3:** Specific comments about sentences/rephrasing/grammar.

**Answer:** Thank you. Most of these are fixed/removed/rephrased.

[Figure]

[Figure]
* * *

[revised manuscript text omitted]

---

## Author Response (AR2)

Dear Editor.

We have now revised the paper according to the new referee comments. We first comment the three major criticisms that were raised in this second revision round. Detailed answers to the remaining referee comments are found below.

Once again we would like to thank the referees for many good suggestions. They have all done a very good job reviewing this paper and their work has certainly contributed to improving the manuscript. Hopefully our revisions have made the paper acceptable for publication.

Kind regards,
Thea Roksvåg and co-authors.
* * *
**Editor comments / Main concerns**

1) One of the referees remains concerned about our claims regarding the areal model and writes:

"However, the main claim remains: That the areal method conserves mass. This remains undemonstrated. The authors have claimed it is proven in a companion arXiv article, but arXiv is self-administered repository that is not peer-reviewed."

Again, we have chosen not to demonstrate the mass-conservation properties of the areal model by experiments in the revised manuscript. There are several reasons for our choice: That the areal method is able to conserve mass is a mathematical property that is given by Equation (7) and (11). It is a direct consequence of the model specification that we explain mathematically/theoretically in Section 4.2.2. Hence, from math/theory we think that our claims regarding the mass-conserving properties of the model are valid and don't need additional documentation. However, that the mass conserving properties actually have a *practical* importance is another question, and this remains undemonstrated in this particular paper. However, we mention it in discussion section where we state that the practical differences between the areal and centroid model is low in terms of posterior mean for the Norwegian dataset, possibly because of the low percentage of nested catchments. That a point referenced model and an areal referenced model often produces similar results is shown by other studies as well. We have added two references on this (on page 33, line 1-2) as suggested by referee Dr. Jon Olav Skøien: (Farmer, 2016 and Skøien et al., 2014).

That the Norwegian dataset was not suitable for demonstrating
that the areal model conserves mass, means that we need a new case/dataset to demonstrate the mass conserving properties. We think that introducing an additional

example will make our presentation confusing (and it is already a relatively long paper). In practice, it would mean repeating the example from the arXiv article. The arXiv article demonstrates the mass-conserving properties of the areal model for a practical example, but is at the moment only available on this self-administrated repository, as the Anonymous referee states. However, the paper is in review in a statistics journal. It was resubmitted in February after the first revision round. Hopefully it will published soon. We think that citing the arXiv-article is supported by the HESS guidelines:

"Informal or so-called "grey" literature may only be referred to if there is no alternative from the formal literature. Works cited in a manuscript should be accepted for publication or published already. In addition to literature, data and software used should be referenced (citations should appear in the body of the article with a corresponding reference in the reference list). These references have to be listed alphabetically at the end of the manuscript under the first author's name. Works "submitted to", "in preparation", "in review", or only available as preprint should also be included in the reference list. Please do not use bold or italic writing for in-text citations or in the reference list."

Finally, we don't think that discussing the areal vs. centroid model should be the main point of this paper. Our discussion on page 32 line 1-8, page 33 and line 1-21 should be sufficient. Here benefits and drawbacks of both methods are mentioned, and we don't hide that the centroid model and the areal model perform similarly for our test dataset. Claiming that the areal model conserves mass is valid from a mathematically point of view.

2) The second major criticism of the referees was related to the significance of the results. This criticism is met by performing a paired Wilcoxon Signed-Rank test on the RMSE/CRPS results. See Table 1 in the manuscript, the bottom on page 22, and page 23, line 1-5. According to this test, our conclusions remain the same as before, and the results are significant.

3) The third major criticism was related to the linear regression results, where we do regression based on two data points. The Anonymous referee writes:
«The result is zero in the denominator. The linear regression used for comparison must use more than two data points."

It is not correct that our results are not mathematically valid: We do a linear regression with two parameters (sigma^2 and beta1). We don't include the intercept (beta0) in the model. Two unknown variables and two observed values makes linear regression model with uncertainty mathematically feasible. We have clarified this in the revised manuscript emphasizing that we have a model without intercept. See page 21, line 18-21.

A model without intercept means that we in practice force beta0 to be zero. For this type of model, linear regression does not give a straight line between the two observations.

The results show that regression with two points actually gives good predictions for the Norwegian annual data (Table 1, RMSE for PG annual) We think that this is a good

illustration of the behavior of the Norwegian annual runoff: The spatial pattern of annual runoff is very strong over time, and you can actually get quite good results by computing the ratio in runoff between to catchments and using this ratio for prediction (see Figure 2b for motivation). The linear regression results can contribute to explaining why the areal and centroid model work so well for the annual data, and we have kept the results in the paper.

───────────────────────────────────────────────

**Dr. Gregor Laaha**

**GL**: "Please describe and plot only the catchments used in the evaluations, not others that are not used. In detail:
The second sentence (It consists of … 450 catchments … is misleading, as this data set has not been used in the study. Please delete.»

**Reply**: This is fixed by removing/modifying the lines were we mention 450 catchments.

**GL:** "p5, L4: "In total 53 of 180 catchments used for cross-validation were nested" – Add: "(30%)" as the degree of nestedness is quite important."

**Reply**: This is added. We have also added the number of nested catchments for the dataset in Figure 5. See page 5, line 10-11 and page 7, line 18.

**GL:** "P7: There are two data sets mentioned, consisting of 260 and 83 catchments, respectively. This is a gain quite confusing. If my guess is right, the 83 have finally be used – so please delete the part describing 260, and add the number and ratio of nested catchments instead.»

**Reply:** We use runoff observations from all 260 catchments. However, we do the cross-validation predictions for the 83 fully gauged catchments: For these catchments we know the true mean annual runoff. The partially gauged catchments are used as observations. We have clarified this part on page 7, line 16-18. We have also updated Figure 5 to make it clearer which catchments are fully gauged (black borders) and partially gauged (no borders). Also see page 22, line 7-13.

**GL:** "Figure 6 and text: The problem here is that the example gives no valid estimation problem for Top-kriging, so it cannot be used for demonstrating the superiority of the areal method. Note that Top-kriging is defined to estimate discharges at river sites, and is not defined for disaggregating these discharges into sub-catchments. When reformulating this setting into a valid estimation problem for Top-kriging, there would be one gauge at the outlet of u1+u2+u3, and a gauge at the outlet of the entire catchment u1+u2+u3+u4+u5. As kriging is an exact interpolator, the discharges at the gauges are exactly predicted, and the derived flow difference from the predictions would be 2500 mm/year. Top-kriging implicitly conserves the water balance without requiring additional discharge constraints.
p17, L12 is therefore not valid for Top-kriging."

**Reply:** I don't think it is correct that Top-Kriging is able to conserve the water balance in this example, i.e. I don't think Top-Kriging is able to predict 2500 mm/year when area $A\_4$ is unobserved, and the observed values are 2000 mm/year ($A\_1$), 2000 mm/year ($A\_3$) and 1000 mm/year ($A\_2$) as in the original Figure 6. As I understand Top-Kriging, the runoff observations are considered as areal referenced when computing the covariance between catchments. This influences the model to weight subcatchments more than non-overlapping nearby catchments. However, the sum of the Kriging weights (the lambda's) are still restricted to be lower than 1 (this is a consequence of requiring an unbiased estimator). Hence, it is not possible to predict larger values than any of the observed values (except for a uncertainty sigma, that usually is relatively small). It is also stated in the Top-Kriging paper that the mass balance is not necessarily conserved: *"In fact, although Top-kriging is based on linear aggregation it does not necessarily reproduce the mass-balance of the variable of interest (Sauquet et al., 2000)."* I suppose reviewer Jon Olav Skøien can correct me here if I am wrong.

Furthermore, the purpose of the example is not to propose a valid estimation problem for Top-Kriging or demonstrate the superiority of the areal method. The purpose is to demonstrate how the areal method works (and state that this is different from Top-Kriging), not to claim that one approach is better than the other.

Based on this reviewer comment, and a comment by Dr. Jon Olav Skøien, we think that Figure 6 possibly has been a bit confusing and unclear. To clarify, we have made a new, simpler figure with 3 catchments instead of 4 catchments. We have also added the direction of the river, and the location of the river outlets/gauges. The new figure text also explains which nodes belong to which catchments. Hopefully, this will make the example simpler to follow.

In addition, we have added that the Kriging methods require that the sum of lambdas is 1. See page 10, line 20-22.

**GL:** "I guess that the climatic GRF c(u) was updated in the cross-validation at each turn, when one catchment / one fold of catchments was left out? Pls. Specify.»

**Reply:** Yes, you are correct. This is now specified on page 21, line 31-32.

**GL:** P22, Eq. 15: The r² is an unusual, not recommended measure as it is seemingly (or often confounded with) the coefficient of determination R², but subtracts any biases and is therefore misleading in case of systematic errors. Suggest that you report the coefficient of determination (R²) instead.

 Please use the coefficient of determination ($R^2$) instead to cover possible biases (see my previous comment).

**Reply:** We use $r^2$ because this score is used in the "Runoff Prediction in Ungauged Basins" book by Blöschl et al, 2013. The purpose of including $r^2$ is hence to make it possible to compare our results to comparable studies in this book. This was also stated on page 23, line 10-12. We have used the other evaluation scores (ANE, RMSE and CRPS) to cover possible biases.

**GL:** p32 L4: Please be exact: For data set 1, 70% of catchments are not nested (rather than "more than 50%). How is I for data set 2?

**Reply:** This is added. See page 33, line 4-6.
47 % of the catchments are not nested for dataset 2.
* * *
**Dr. Jon Olav Skøien**

**JOS:** The authors have updated the text with the number of catchments with observations length 1-3, showing that these are not as rare as one could expect. As an additional possible use case, the manuscript could be used as motivation for installing new (maybe temporary) stations, as they can improve long-term estimates only a year after installation.

**Reply:** Yes, this is a good motivation for the framework. It can also be used to decide whether a gauging station can be shut down. We have added this on page 36, line 10-13. We have also added it to the conclusion, page 36, line 30-31.

**JOS:** The text on P23 refers to how the methods fail for some catchments, based on the RMSE. However, as the runoff values are much higher in the catchments in western Norway, the RMSE might not be the best measure for capturing the prediction performance. This could at least be mentioned.

**Reply:** This is now mentioned on page 24, 1-3.

**JOS:** The authors now refers to lognormal being an option, but not compatible with the linear aggregation assumption, what I mentioned in the previous review. I am fine with the authors not testing this. However, one should be careful dismissing a method only for violating one of the assumptions, when they are most likely already violated to some degree in other ways, probably similar to a large number of interpolation use cases found in the literature. The suggested framework can handle heteroscedasticity better than TK, as this is already a violation of the assumptions behind kriging. However, it is never tested if the assumptions of runoff data is actually Gaussian, and I doubt that a test would confirm it. Runoff is truncated at zero, which is the reason

for the problem with negative estimates. A lognormal transformation could solve this violation, but instead create a new.

**Reply:** This is a good point. Yes, you are correct that the data are not Gaussian. This can be seen from the histograms in Figure 1 and Figure 3. However, as you say, when specifying a model, there are always choices regarding which assumptions that can/should be included in the model or not. In this case: 1) Do we allow the violation of the water-balance or 2) do we allow negative values (and the modeling of non-Gaussian data as Gaussian). We have added a bit more discussion around this topic on page 35, line 6-3 and 26-29.

**JOS:** The difference between the centroid model and the areal model is small for most cases and catchments. It can be mentioned that this is somewhat expected, as it has also been shown for ordinary kriging vs top-kriging, by e.g. Farmer (2016) and Skøien et al. (2014).

The discussion around transferability of observations from neighbouring catchments and gauging density could maybe have a reference to Patil and Stieglitz (2012).

**Reply:** Thank you for good references. These are added in the discussion. See page 33, line 1-2, and on page 34, line 17-18.

**JOS:** It could also be mentioned in the discussion that it would be possible to use Top-kriging on residuals in the partially gauged case, which might improve the performances of the method. It is fair not to include this in the comparison, as there are different possible ways to implement such a procedure, and that is out of scope for this study.

**Reply:** This is a good suggestion. We did not include this, but we suggest in the discussion that the areal/centroid model can be used as a pre-processing step for the partially gauged catchments, before doing interpolation with Top-Kriging for ungauged catchments. See page 36, line 1-6.

**JOS:** Minor edits:

**Reply:** These are taken care of.
* * *
**Anonymous referee**

**Referee:** In the previous version, almost all reviewers expressed concerns about the claims of the areal method. The revision and response do much to soften the claims of the areal method. However, the main claim remains: That the areal method conserves mass. This remains undemonstrated. The authors have claimed it is proven in a companion arXiv article, but arXiv is self-administered repository that is not peer-reviewed. This advantage of the areal model, if it is important, should be demonstrated in peer-reviewed literature. For more questions, see my

comments on an earlier version.

The authors have failed to respond to the editor's and reviewers' comments on the significance of performance differences. In my review, I discussed how tables 1, 2 and 3 show only average performance and do not interpret the significance (or some proxy thereof) of the differences across methods. The editor raised this as a concern as well. The revision continues to ignore the question of significance (whether formal significance or some proxy). Without this information, we evidence of differences in performance is weak.

**Reply**: See the above reply to the Editor, point 1 (areal method) and 2 (significance).

**Referee:** The authors have done a better job of acknowledging that their proposed method does not improve over standard methods in the UG case. They appropriately highlight the marked improvement in performance with the introduction of a single observations. This is a very interesting finding and worthy of publication. It shows how their method can better leverage sparse information. However, in order to strengthen this case, **the authors should consider how to demonstrate that the increases (though large) are a result of improvements and not random selection of years (i.e., an approach to testing significance is needed here as well).**

**Reply:** We think that we already have demonstrated that the increases are actual improvements and not due to a beneficial selection of years. The reason is that we have demonstrated the method for four different datasets: Annual, January, April and June, and the method performs as expected for all datasets.

Furthermore, we also test the approach for *mean* annual runoff on a different dataset than in the first experiment and for four different record lengths (PG1, PG3, PG5 and PG10). For each of these configurations (PG1, PG3, PG5 and PG10), a new sample of years is drawn randomly for each catchment. Hence, in total we have done 8 experiments (PG Annual, PG January, PG April, PG June, PG1, PG3, PG5 and PG10) and the results are as expected. This supports our claim.

**Referee:** When considering the PG case, the authors continue to use a regression based on two data points. I raised this concern in my previous comments, and the authors did not respond. It is wholly inappropriate to build a regression on two data points. The result would just be a straight line, from which it would be impossible to conduct any statistical inference (e.g., intervals, significance, etc.). (This can be showing by considering the denominator of most of the OLS formula, which have n (number of observations; here, 2) – k (number of slopes; here, 1) – 1 (for the intercept term). The result is zero in the denominator. The linear regression used for comparison must use more than two data points.

**Reply:** See the above reply to the Editor, point 3.

**Referee:** Many of the kriging applications in hydrology have shown how average variograms (rather than using year-specific variograms) can improve regional performance; this could be relevant in that so-called "average" variograms can absorb partial record information. While probably beyond the scope of this work at the time, it seems like it should be acknowledged that others have proposed different methodologies for incorporating partial records (I'm thinking of the entire field of record augmentation, including MOVE methods and the like).

**Reply:** We have added some more references on record augmentation and MOVE methods. See page 11, line 8-10, and page 2, line 33-34. We have also changed the name of subsection 3.4 (record augmentation techniques in parenthesis).

─────────────────────────────────────────────────────────

[revised manuscript text omitted]

---

## Author Response (AR3)

**Response letter**

Dear Editor.

We have now revised the paper according to the referee comments. Our responses to the referee comments are found below together with a marked-up version of the paper. We would like to thank the reviewers for nice discussions through these revision rounds and for patiently revising several versions of this paper.

Kind regards,
Thea Roksvåg and co-authors.

**Referee 1: Dr. Gregor Laaha**

**GL:** *"I would, however, be careful with the statement that (Top-)kriging cannot produce larger values than the observed values because of the unbiasedness condition (kriging weights sum up to one). I have often read about negative kriging weights, which can make the estimates to fall outside the range of observed data (e.g. Wackernagel, 2003)."*

**Response:** Thank you for pointing this out. We have updated this part. See page 18, L11-16 and L25-32. We also provide some more discussion around the differences and similarities between our method and Top-Kriging.

**Referee 2: Anonymous reviewer**

**A:** *"I remain concerned with the use of regression for partially gauged sites. I've raised this before, so I'm sure the authors are not happy to hear it again, but I do want to log my concern. I suspect that the authors will disagree, which is fine, and, as other may feel similarly, it might be a point for discussion. In their experiment.to replicate partially gauged sites, the authors compute a regression with two observations of paired years at the target and donor site. My concern is that this is far two few points to build a meaningful regression. Most statisticians will tell you that, as a rule of thumb, you would want roughly ten data points per predictor variable. [...] However, this application of regression is only used for a point of comparison. I think the lack of data provides a bit of a disadvantage to LR, but I still believe that the novel method would outperform the best use of LR. As that is my belief, it may not be useful to change the methodology, but rather to provide a discussion of the implications of a limited use of LR".*

**Response:** We agree that a linear regression with two observations is not optimal. As stated in our previous answer, we can think of linear regression as a method that can tell us something about the target variable and the study area. If linear regression performs well in spite of having only two observations, it means that the runoff in the study area is very correlated. This also tells us something about the proposed method: In such areas short records have a potentially high value and should be exploited. This was seen for the annual runoff data, where linear regression actually performed very well. To clarify that the linear regression results are not optimal, we have added some sentences on page 21, L31-36. Here, we suggest how the

linear regression results can be understood.

**Referee 3: Dr. Jon Olav Skøien**

***JOS:*** *"The issue of mass conservation was discussed by the other reviewers, and also discussed in relationship with top-kriging. There is maybe a confusion here – whereas TK is based on the assumption that the variable is linearly aggregated and mass-conserved, it does not have constraints for preserving the mass balance between subcatchments and the basin. This is something that was done in a slightly different approach by Sauquet et al. (2000), introducing the mass balance as extra constraints in the kriging equation. On the other hand, the authors answer that TK cannot predict outside the observation range because the sum of the weights is equal to 1, and have also added this to the text. This is not correct and should be corrected (P10 L20-22 and P18 L13-15). Kriging methods can interpolate outside the range as long as negative weights are allowed. The sum of the weights is one, but the sum of the norm of the weights can be above 1. Another confusion in the understanding of the figure 6 might also be that the areal method, as far as I understand, is trying to predict at the grid nodes, whereas TK only uses the grid nodes to establish the statistical relationships between subcatchments, hence no prediction for the individual grid points."*

**Response:** Thank you for correcting our misinterpretation of the Top-Kriging approach. We have removed our claim that Top-Kriging cannot predict higher values than the observed values. Based on your comments, we have also added some more discussion around the differences and similarities between our method and Top-Kriging, and linked the methods to the Sauquet method. See page 18, L11-16 and L25-32. Also thank you for pointing out the mistake in the calculations on page 18. It should be 2667 mm/year and not 2500 mm/year. This is now updated.

***JOS:*** *"It should be clearer from the text what MOVE methods are."*

**Response:** We have added a couple of sentences about MOVE methods on page 11, L9-14.

[revised manuscript text omitted]